# AN EVOLVED UNIVERSAL TRANSFORMER MEMORY

**Edoardo Cetin, Qi Sun, Tianyu Zhao, Yujin Tang**
Sakana AI, Japan
{edo,qisun,tianyu,yujintang}@sakana.ai

## ABSTRACT

Prior methods propose to offset the escalating costs of modern foundation models by dropping specific parts of their contexts with hand-designed rules, while attempting to preserve their original performance. We overcome this trade-off with Neural Attention Memory Models (NAMMs), introducing a learned network for memory management that improves *both* the performance and efficiency of transformers. We *evolve* NAMMs atop pre-trained transformers to provide different latent contexts focusing on the most relevant information for individual layers and attention heads. NAMMs are universally applicable to any model using self-attention as they condition exclusively on the values in the produced attention matrices. Learning NAMMs on a small set of problems, we achieve substantial performance improvements across multiple long-context benchmarks while cutting the model's input contexts up to a fraction of the original sizes. We show the generality of our conditioning enables zero-shot transfer of NAMMs trained *only* on language to entirely new transformer architectures even across input modalities, with their benefits carrying over to vision and reinforcement learning. Our source code is available at https://github.com/SakanaAI/evo-memory.

## 1 INTRODUCTION

Transformer architectures have become the golden standard in deep learning, with ubiquitous applications in the design of modern foundation models, exhibiting exceptional performance and scalability (Achiam et al., 2023; Das et al., 2023; Team et al., 2023; Dosovitskiy et al., 2020; Chen et al., 2021a; Brohan et al., 2023; Gur et al., 2023). The outputs of a transformer are exclusively conditioned on a recent context of input tokens, which for language models (LMs) generally correspond to a window of preceding words. Thus, addressing the challenge of extending this context window is critical to enable tackling long-range tasks and is currently a focal area of re-

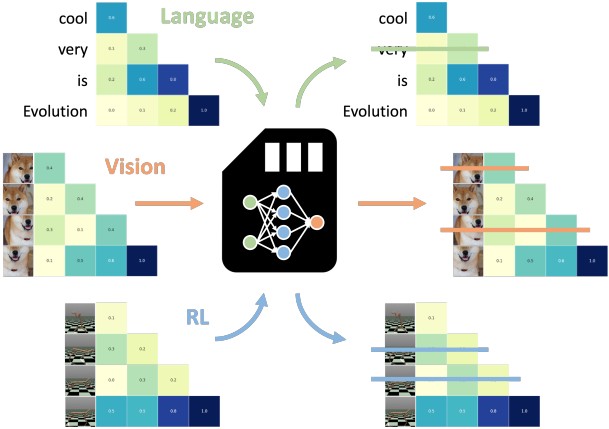

Figure 1: NAMMs use evolution to optimize the performance of LMs by pruning their KV cache memory. Evolved NAMMs can be zero-shot transferred to other transformers, even across input modalities and task domains.

search (Huang et al., 2023). However, long contexts also immediately impact training and inference costs, with modern foundation models being increasingly resource-hungry and expensive. Many recent methods proposed to partially offset these costs by studying how to heuristically quantify the importance of each token stored in the model's *latent memory*, i.e., stored in its *Key-Value (KV) cache*. Then, by simply *evicting* the least important tokens with hand-designed strategies, they have shown early success at reducing memory size while limiting performance losses (Luohe et al., 2024).

Our research aims to go beyond these hand-designed strategies as we hypothesize that shaping the latent memory KV cache of transformers entails new opportunities to *improve* their capabilities in

downstream tasks. One widely evidenced example in support of our hypothesis is the effectiveness of hand-crafted input context modifications through prompt engineering (Liu et al., 2023), even allowing foundation models to learn *in-context* entirely new skills at test time (Brown et al., 2020). Furthermore, unlike prompt engineering, directly managing the memory of transformers enables the provisioning of distinct contexts to each latent level independently, such that individual layers and attention heads can focus on the most relevant information for their specific needs.

Motivated by these considerations, we propose Neural Attention Memory Models (NAMMs), introducing a new class of networks trained with evolution to learn an efficient memory system that maximizes the downstream performance of pre-trained transformers. Evolution inherently overcomes the non-differentiability of memory management operations with binary outcomes (selecting tokens to preserve/discard) which renders gradient-based optimization incompatible. Our efforts are inspired by the key role that natural evolution played in shaping human memory, which analogously appears to selectively incorporate and actively prune information based on its lifelong usefulness (Sherry & Schacter, 1987; Nairne & Pandeirada, 2010; Frankland & Bontempi, 2005).

Our NAMMs are conditioned on features entirely constructed from the attention matrix, making them universally applicable to any transformer-based architecture. Learning NAMMs atop a pre-trained Llama 3 8B model (Dubey et al., 2024), we not only obtain efficiency benefits, with substantial reductions in the number of retained tokens in the KV cache, but also *exceed* the performance of the full-context model with notable margins. We validate these findings across 36 different tasks from LongBench (Bai

Table 1: Summarized NAMMs performance in language modeling (top) and zero-shot transfer settings (bottom)

| Model/Eval | LongBench | | InfiniteBench | | ChouBun | |
|---|---|---|---|---|---|---|
| | Performance | Cache size | Performance | Cache size | Performance | Cache size |
| Base model | 28.86 (1.00) | 32768 (1.00) | 1.05 (1.00) | 32747 (1.00) | 21.21 (1.00) | 12099 (1.00) |
| H2O | 28.37 (0.99) | 8192 (0.25) | 1.05 (1.00) | 8193 (0.25) | 19.86 (0.94) | 8292 (0.69) |
| L2 | 27.42 (1.00) | 8192 (0.25) | 1.63 (1.55) | 8193 (0.25) | 18.93 (0.89) | 8292 (0.69) |
| FastGen | 27.88 (0.95) | 9538 (0.94) | 1.42 (1.49) | 23016 (0.70) | 18.93 (0.89) | 8616 (0.71) |
| NAMMs | 29.33 (1.11) | 8155 (0.25) | 11.00 (10.45) | 13192 (0.40) | 24.44 (1.15) | 9895 (0.82) |

| Model/Eval | Llama 3 70B | | Computer Vision | | Reinforcement Learning | |
|---|---|---|---|---|---|---|
| | Performance | Cache size | Performance | Cache size | Performance | Cache size |
| Base model | 35.22 (1.00) | 10107 (1.00) | 43.84 (1.00) | 7039 (1.00) | 29.04 (1.00) | 3000 (1.00) |
| H2O | 34.17 (0.97) | 6662 (0.66) | 41.97 (0.96) | 4479 (0.64) | 28.70 (0.99) | 2048 (0.68) |
| L2 | 33.50 (0.95) | 6662 (0.66) | 41.45 (0.95) | 4479 (0.64) | 27.91 (0.96) | 2048 (0.68) |
| NAMMs | 34.70 (0.99) | 8365 (0.83) | 44.38 (1.01) | 5100 (0.72) | 31.73 (1.09) | 2434 (0.81) |

et al., 2023), InfiniteBench (Zhang et al., 2024a), and *ChouBun*[1], a new Japanese benchmark designed to assess long-context capabilities beyond the common English and Chinese. These results mark a clear contrast with the aforementioned hand-designed strategies that appear to inevitably trade off efficiency for performance, in line with their stated purpose.

Furthermore, we show that the generality of our parameterization enables *zero-shot transfer* of NAMMs trained on three natural language tasks to entirely new transformer models. In particular, we obtain further performance and efficiency improvements not only when using the evolved NAMMs with other LMs of increased size, but also transformers with entirely different architectures concerned with new input modalities, for problems such as vision and reinforcement learning. In a nutshell, our main technical contributions can be summarized as the following:

- We introduce NAMMs, a novel memory evolution framework that adds a new dimension to optimizing transformer models without altering their powerful architectures.

- We design and successfully train NAMMs on top of pre-trained transformer models, obtaining both performance and efficiency gains on several long context language tasks.

- We show NAMMs, trained only on language tasks, can be transferred zero-shot to any other transformers, retaining benefits across different input modalities and task domains.

## 2 BACKGROUND AND PRELIMINARIES

**Attention and transformers.** Transformers are neural network architectures designed specifically for efficiently processing input sequences. These models take as input a stream of tokens (e.g., embeddings of words, image patches, robotic states, etc.) and, produce a set of latents with the same length within their layers. Multi-headed dot product attention (Vaswani et al., 2017), or simply *self-attention*, characterizes modern transformers, facilitating effective information sharing across

---

[1]ChouBun is the pronunciation of "長文", literally translating to "long text" in Japanese.

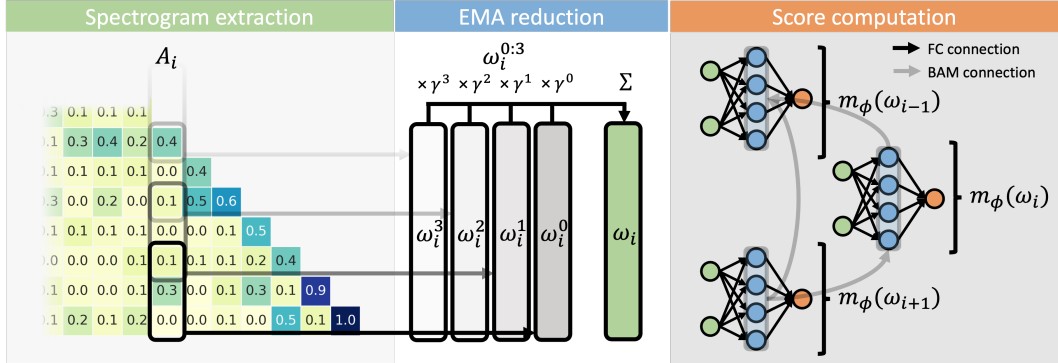

Figure 2: Schematic depiction of our Neural Attention Memory Model design. We extract features from a spectrogram over the attention values of the KV cache tokens (left), which we reduce via an element-wise exponential moving average (EMA) operation (center). These features are fed to our memory model's networks with fully connected (FC) and cross-token BAM connections (right).

token representations. The attention layer conducts a set of parallel computations, each known as an attention head, mapping tokens to query, key, and value vectors $\in \mathbb{R}^d$. These vectors are organized along the sequence dimension in the matrices $Q$, $K$, and $V$, and the layer's output is computed as:

$$attention_M(Q, K, V) = AV, \quad \text{where,} \quad A = softmax\left(M \times \frac{QK^T}{\sqrt{d}}\right). \tag{1}$$

Here, $M$ represents an optional mask multiplying the *attention matrix* $A$, usually enforcing an *auto-regressive conditioning* such that each token cannot attend to its future. An interpretation of the attention layer comes from the elements of the attention matrix $A_i^j$, i.e., the dot products between each key $i$ and query $j$ normalized along the column dimension. Intuitively, each of these values can be understood as the relative *importance* of token $i$ in processing the input representation of token $j$.

**Frequency-based feature extraction.** An established canonical technique to pre-process one-dimensional non-stationary signals is the Short-Time Fourier Transform (STFT) (Allen & Rabiner, 1977). This technique has seen plenty of applications for feature extraction concerning audio, biomedical, seismic, and many more kinds of modalities. The STFT performs a time-convolution of a signal, shifting each convolutional window to the frequency domain through a discrete Fourier transform, producing a *spectrogram* representation of the original input. We use $\omega^t \in \mathbb{R}^{N+1}$ to denote the fixed-sized vector produced at each timestep $t$, where the $N$ frequencies span from zero up to the Nyquist frequency (half the original sampling rate). Mathematically, the $n$-th frequency from an STFT for time $t$ is extracted from an input vector $v \in \mathbb{R}^T$ as:

$$\omega^t[n] = \sum_{t'=0}^{T} v[t']w[t - t']e^{\frac{-n\pi t}{N}}. \tag{2}$$

Here, the convolutional filter of the SFTF is defined by the product of a finite-length *window function* $w$ with each exponential term in the Fourier transform. A popular choice for $w$ is the Hann window (Oppenheim, 1999), employing a smooth decay at its edges which helps minimize the overestimation of the magnitudes of the higher frequencies in $\omega$ due to *spectral leakage* (Harris, 1978).

## 3    NEURAL ATTENTION MEMORY MODELS

An immediate limitation of transformers is the quadratic costs associated with computing the attention matrix $A$. To partially address this issue, during auto-regressive generation, the latents for the keys and values of the tokens generated at the previous steps are usually stored in what is referred to as the KV cache. This object can be regarded as being analogous to the *memory* of the transformer, which now, at each step, only needs to compute the query, key, and value of the latest token and perform attention over a horizontal vector by exploiting causal ordering. In this section, we describe the feature extraction, architecture, and optimization of NAMMs, which have been designed to act on the KV cache to improve both the performance and practicality of this powerful class of models.

### 3.1 ATTENTION SPECTROGRAMS FOR MODEL-AGNOSTIC FEATURE EXTRACTION

The feature extraction framework of NAMMs is designed to be agnostic to the parameterization of the base transformer they are applied for. In particular, we build a representation for each token in the current KV cache memory directly from its corresponding unmodified column vector in the attention matrix $A_i$. To meaningfully compress this unbounded vector signal, we process it via an STFT with a fixed-sized Hann window (Figure 2, left). This operation produces a spectrogram representation of the attention columns $\omega_i^t$, representing the frequencies with how the queries attend to each of the stored key tokens (indexed by $i$) on a compressed time-axis (indexed by $t$). Thus, this representation exposes precisely the knowledge of how each token's relative importance varies across all past queries in a compact form factor, discarding all other information specific to the learned transformer weights.

As NAMMs rely only on the attention values for their input, they are universally applicable to any layer producing an attention matrix. This property is crucial, enabling us to avoid learning individual memory models for the different layers of a transformer, thus, greatly limiting the number of total optimized parameters. Furthermore, it also allows efficient training on top of smaller foundation models for targeted problems, and later transferring the resulting models zero-shot at test-time to larger architectures and arbitrary applications.

### 3.2 MEMORY MODEL DESIGN AND CROSS-TOKEN COMMUNICATION

NAMMs parameterize a small neural network $m_\phi$ to output a scalar *selection score* $s_i = m_\phi(\omega_i^{1:T})$ for each $i^{th}$ token in the KV cache. First, to obtain a consistent input dimension, we reduce the attention spectrogram into a smaller feature vector $\omega_i$ by compressing the time-axis via an element-wise exponentially moving average (EMA: $\omega_i = \sum_t \gamma^t \omega_i^t$; Figure 2, center). We then append positional encodings and feed the vector $\omega_i$ to the memory model's network $m_\phi$ to produce the score $s_i$. Finally, we evict from the KV cache memory all latent tokens with $s_i < 0$, effectively treating the problem as a binary classification task. We repeat this process with a fixed interval, every set number of new input tokens, $n_{\text{up}}$.

**Backward attention memory models (BAM).** For the design of $m_\phi$, we posit that sharing information from all tokens in memory could be key for assessing their importance. A particularly motivating scenario in LMs arises when considering the case of repeated words or sentences, where learning a diversity measure that compares different tokens would allow preventing redundancies in the KV cache. Corroborating this intuition, even from a biological perspective, memory formation and retention appear to adhere to models of neuronal competition (Han et al., 2007).

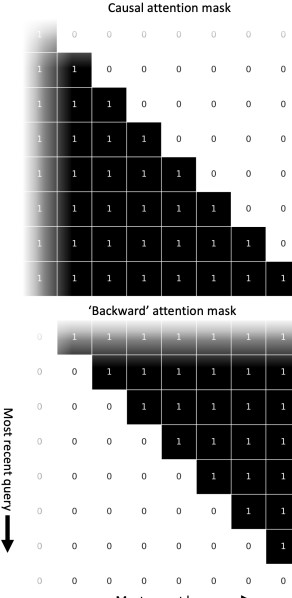

Figure 3: Our backward mask makes each token attend exclusively to its future relatives in the KV cache.

Based on these considerations, we design the backward attention memory architecture (BAM) for parameter-efficient sharing of information while making use of the powerful inductive biases enabled by the masked self-attention operation. In particular, we implement $m_\phi$ via an initial self-attention layer with a *counter-causal* mask $\hat{M}$, which we refer to as *backward* (Figure 3). This design serves to introduce a purposeful asymmetric relationship, allowing to distinguish between older and newer tokens. We then output $s_i$ from a final linear operation:

$$o_i = attention_{\hat{M}}(K_\Omega, V_\Omega, Q_\Omega), \quad s_i = linear(o_i), \tag{3}$$

where $K_\Omega, V_\Omega, Q_\Omega$ are the key, value, and query matrices from all feature vectors $\omega_i$ in memory. Using BAM to tackle the previous motivating scenario, only the representation for older tokens would be potentially affected by the presence of newer duplicates. Thus, just by learning a simple diversity metric within self-attention, backward masking would provide the memory model with the potential to preserve only the most informed occurrence of each token without risking discarding any information in its entirety (since the score for the latest instance of each repeated token would be independent of its past).

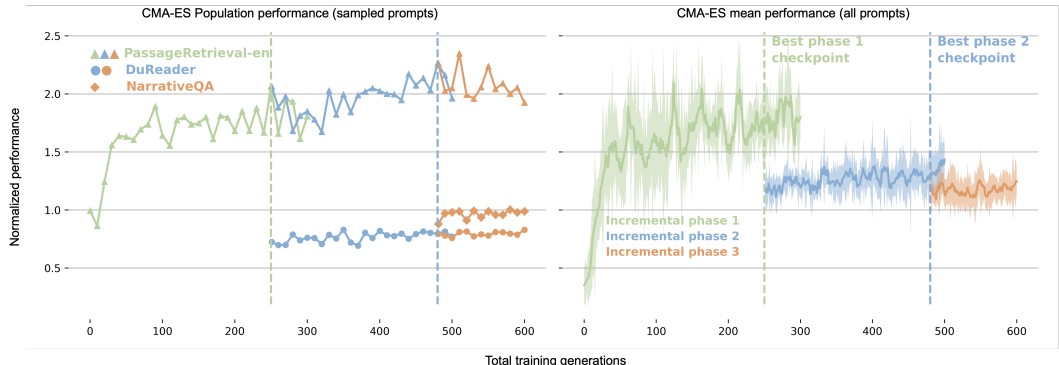

Figure 4: Mean and standard deviation over the CMA-ES population batch performance (left), together with the performance of the learned mean parameter on each task (right).

In practice, when applying NAMMs, we only affect the KV cache of the base model with a fixed frequency, once every $n_{up}$ steps. When feeding longer prompts to our model, we simply split the tokens into $n_{up}$-sized chunks. We summarize the full execution pipeline of NAMMs in Algorithm 1. We refer to Appendix A and our shared code for additional implementation details and discussion.

---

**Algorithm 1** NAMMs

**Input:** KV cache, Iteration $k$, Stride size $s_w$, Update interval $n_{up}$, Attention matrix $A$ (latest $n_{up}$ queries), Past STFT $\omega'$

1: **if** $k \% n_{up} == 0$ **then**  $\triangleright$ use NAMMs every $n_{up}$ steps
2:    **for** each $i^{th}$ token in the KV cache **do**  $\triangleright$ or column $A_i$ in $A$
3:        STFT $\omega_i^t$ for $t = 0, \ldots, n_T$  $\triangleright$ Eq. 2, $n_T = n_{up}/s_w$
4:        Reduce $\omega_i = (\sum_t \gamma^t \omega_i^t) + \gamma^{n_T} \omega'$  $\triangleright$ EMA reduction
5:    $s_i = m_\phi(\omega_i)$  $\triangleright$ apply NAMMs, Eq. 3
    **Output:** KV cache where $s_i > 0$  $\triangleright$ update the KV cache

---

### 3.3 INCREMENTAL EVOLUTION

We evolve the network weights of our NAMMs to directly optimize the performance on a subset of long-context language modeling tasks from LongBench (Bai et al., 2023). As we share a single $m_\phi$ across all layers, even with our largest NAMM we only evolve about 4000 total parameters. We use the seminal CMA-ES optimization algorithm (Hansen, 2006) and apply NAMM atop a Llama 3 8B base model (Dubey et al., 2024) with a context extended from 8192 to 32768 tokens via NTK-aware positional interpolation (bloc97, 2023). Due to the inference costs of LMs with long inputs, we sample a subset of different prompts from each task in each generation and propose training in an *incremental* fashion: starting from a single task, and adding additional tasks at later training stages. Empirically, we found both these choices to provide effective regularization, improving generalization (see Appendix C). The performance of modern LMs on LongBench varies considerably across tasks, and even across different task prompts. Hence, instead of using the raw scores, we opt to maximize normalized performance relative to the vanilla base model's stored evaluation performance on each same subset of prompts, retaining all tokens in its KV cache memory. Using evolution, we note that our training loop simply corresponds to running inference NAMMs atop the base, requiring no expensive backpropagation or dedicated hardware.

We choose three tasks from different LongBench categories across both English and Chinese where the Llama 3 base model seems to particularly struggle: PassageRetrieval-en, DuReader, and NarrativeQA; optimizing the normalized exact match, ROUGE-L, and F1 metrics, respectively. We evolve our NAMM for 300 generations in its first incremental phase, 250 in its second, and 120 in its third. We diminish the number of generations to counteract the increasing costs with each additional phase and make more efficient use of computational resources. At the end of each phase, we resume from the best previous checkpoint. We provide training curves of our main backward-attention model in Figure 4, showing the average and standard deviation of the normalized batch performance across the population (left), together with the normalized per-task and average performance on all samples of the optimized mean from CMA-ES (right). We refer to Appendix A for additional architectural and optimization details, together with the set of hyper-parameters. We also provide additional statistics and training curves for other memory model designs in Appendix C.

### 4 EXPERIMENTAL RESULTS

In this section, we evaluate and analyze evolved NAMMs as compared to full-context transformers and three recent hand-designed methods for KV cache management: H2O (Zhang et al., 2024c)

Table 2: NAMMs evaluation on LongBench. The normalized performance (in brackets) is calculated using the base model with full cache. The tasks used for NAMM's training are highlighted in gray.

| Model/Task id | Single-Doc QA | | | | Multi-Doc QA | | | | Summarization | | | |
|---|---|---|---|---|---|---|---|---|---|---|---|---|
| | 1-1 | 1-2 | 1-3 | 1-4 | 2-1 | 2-2 | 2-3 | 2-4 | 3-1 | 3-2 | 3-3 | 3-4 |
| Base model | 10.38 (1.00) | 12.79 (1.00) | **22.60** (1.00) | 21.31 (1.00) | **10.41** (1.00) | **12.67** (1.00) | 7.54 (1.00) | 25.86 (1.00) | **29.34** (1.00) | 23.93 (1.00) | 0.92 (1.00) | 2.66 (1.00) |
| H2O | 8.75 (0.84) | 13.07 (1.02) | 22.11 (0.98) | 21.62 (1.01) | 10.28 (0.99) | 12.40 (0.98) | 7.20 (0.95) | **26.58** (1.03) | 28.56 (0.97) | 23.98 (1.00) | 0.88 (0.96) | 2.25 (0.85) |
| L2 | 8.83 (0.85) | **13.13** (1.03) | 22.22 (0.98) | **21.79** (1.02) | 9.97 (0.96) | 12.15 (0.96) | 5.88 (0.78) | 24.96 (0.97) | 28.05 (0.96) | 23.28 (0.97) | **1.15** (1.25) | 1.52 (0.57) |
| FastGen | **10.49** (1.01) | 13.09 (1.02) | 21.85 (0.97) | 21.57 (1.01) | 10.28 (0.99) | 11.60 (0.92) | 6.77 (0.90) | 17.04 (0.66) | 28.98 (0.99) | 23.53 (0.98) | 0.85 (0.92) | 3.02 (1.14) |
| NAMM (Ours) | 9.14 (0.88) | 12.63 (0.99) | 21.94 (0.97) | 21.34 (1.00) | 9.71 (0.93) | 11.63 (0.92) | 6.98 (0.93) | 20.58 (0.80) | 28.78 (0.98) | **24.39** (1.02) | 1.04 (1.13) | **3.63** (1.36) |

| Model/Task id | Few-shot Learning | | | | Synthetic | | | Code | | Overall | | |
|---|---|---|---|---|---|---|---|---|---|---|---|---|
| | 4-1 | 4-2 | 4-3 | 4-4 | 5-1 | 5-2 | 5-3 | 6-1 | 6-2 | All tasks | Test tasks | Cache size |
| Base model | **73.00** (1.00) | 89.45 (1.00) | **46.54** (1.00) | **40.00** (1.00) | 1.48 (1.00) | 12.18 (1.00) | **28.80** (1.00) | 69.09 (1.00) | 65.17 (1.00) | 28.86 (1.00) | N/A | 10107 (1.00) |
| H2O | **73.00** (1.00) | **90.03** (1.01) | 46.48 (1.00) | 34.00 (0.85) | 2.18 (1.47) | 9.93 (0.82) | 27.76 (0.96) | 69.37 (1.00) | 65.44 (1.00) | 28.37 (0.99) | N/A | 6662 (0.66) |
| L2 | 66.41 (0.91) | 84.92 (0.95) | 45.78 (0.98) | 34.38 (0.86) | 3.13 (2.11) | 11.00 (0.90) | 28.68 (1.00) | **73.45** (1.06) | 55.20 (0.85) | 27.42 (1.00) | N/A | 6662 (0.66) |
| FastGen | **73.00** (1.00) | 88.76 (0.99) | 46.40 (1.00) | 36.00 (0.90) | 1.15 (0.78) | 10.23 (0.84) | 27.01 (0.94) | 69.34 (1.00) | 64.50 (0.99) | 27.88 (0.95) | N/A | 9538 (0.94) |
| NAMM (Ours) | **73.00** (1.00) | 89.81 (1.00) | 46.35 (1.00) | **40.00** (1.00) | 3.04 (2.05) | **27.55** (2.26) | 28.60 (0.99) | 69.53 (1.01) | **66.35** (1.02) | **29.33** (1.11) | **1.07** | 8409 (0.83) |

and L2 (Devoto et al., 2024), and FastGen (Ge et al., 2024). We compare each method in terms of absolute and normalized performance and also provide the resulting average cache size recorded at the end of each prompt. We first consider three long-context language modeling benchmarks spanning 36 diverse tasks in three languages, using the same Llama 3 8B base transformer from training. Then, we evaluate the capabilities of zero-shot transferring NAMMs to other *unseen* transformers and task domains. In particular, we not only consider transfer to larger LMs, but also transformers with tokens constructed from modalities other than language. Across all these settings, we also compare BAM with a simpler 2-layer MLP architecture and provide summarized results after every stage of incremental evolutions. We refer to Appendix C additional evaluations (e.g., transferring NAMMs to a Mistral LM), ablation studies (e.g., comparing different architectures and input features), and all learning curves. Lastly, we conclude the Section with a targeted qualitative analysis, aimed at understanding the behavior of our new memory framework.

## 4.1 LONG-CONTEXT LANGUAGE UNDERSTANDING

**Longbench.** In Table 2, we provide results across all LongBench tasks (Bai et al., 2023) and in Figure 5 we provide a summarized comparison varying the maximum cache size of H2O and L2 (we provide a similar analysis for FastGen in Figure 9). Our NAMM yields concrete improvements to the Llama 3 8B transformer both when considering the full set or exclusively the held-out set of *test* tasks that were not used for evolution, with improvements of 11% and 7% respectively. At the same time, our NAMM also yields efficiency side benefits, notably reducing the context-extended KV cache size. Instead, H2O, L2, and Fastgen all come with performance costs

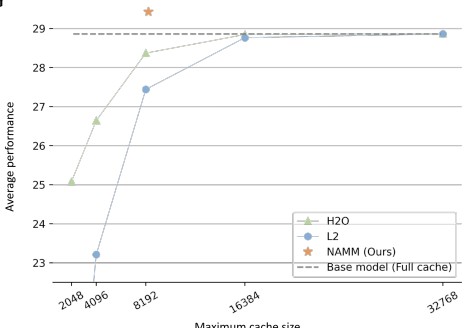

Figure 5: Comparing NAMM with H2O and L2 while varying the cache size.

which notably grow the smaller their cache sizes - in line with their stated objective of *retaining* rather than *improving* the original full-context performance. These results emphasize the inevitable tradeoff induced by prior hand-designed methods, able to obtain efficiency gains but at increasing performance costs due to their lossy heuristics. On the other hand, we find NAMMs successfully provide a paradigm shift, yielding consistent improvements from the base model across both performance and efficiency axes by learning to discard unhelpful information, highlighting how end-to-end evolutionary optimization can open new orthogonal directions beyond what is feasible with manually-designed heuristics.

**InfiniteBench.** In Table 3, we provide results across the InfiniteBench tasks (Zhang et al., 2024a). In this benchmark, the average prompt length is close to 200K tokens making it extremely challenging, especially for LMs that were not expensively finetuned for very long context understanding. In fact, as reported by Zhang et al. (2024a), even GPT4 (Achiam et al., 2023) cannot exceed a performance of 1% on some of its problems. In line with these results, the full-context Llama 3 together with H2O and L2 obtain near-zero performance on most tasks. Instead, our NAMM provides outstanding

Table 3: NAMMs evaluation on InfiniteBench. The normalized overall performance (in brackets) is calculated using the average performance of the base model with full cache.

| Model/Task name | Retrieval | | | Dialogue | Novel | | | | Math | Code | | Overall | |
|---|---|---|---|---|---|---|---|---|---|---|---|---|---|
| | Ret.PassKey | Ret.Number | Ret.KV | En.Dia | En.Sum | En.MC | En.QA | ZH.QA | Math.Find | Code.Run | Code.Debug | All tasks | Cache size |
| Base model | 0.00 | 0.00 | 0.00 | 1.00 | 7.73 | 0.00 | 1.05 | 1.79 | 0.00 | 0.00 | 0.00 | 1.05 (1.00) | 32747 (1.00) |
| H2O | 0.00 | 0.00 | 0.00 | **1.50** | 5.38 | 0.00 | 1.01 | 1.71 | 1.71 | 0.25 | 0.00 | 1.05 (1.00) | 8193 (0.25) |
| L2 | 0.00 | 0.00 | 0.00 | 1.00 | 5.41 | 0.44 | 0.83 | 2.59 | 7.43 | 0.25 | 0.00 | 1.63 (1.55) | 8193 (0.25) |
| FastGen | 0.00 | 0.00 | 0.00 | 1.00 | 5.82 | 1.31 | 1.25 | 1.38 | 5.43 | 0.00 | 0.25 | 1.42 (1.49) | 23016 (0.70) |
| NAMM (Ours) | **11.86** | **11.86** | **1.80** | 1.00 | **14.91** | **36.24** | **8.78** | **17.67** | **10.57** | **1.75** | **4.57** | **11.00** (10.45) | 13192 (0.40) |

Table 5: NAMMs evaluation on LongBench with a Llama 3 70B model. The normalized performance (in brackets) is calculated using the base model with full cache.

| Model/Task id | Single-Doc QA | | | | Multi-Doc QA | | | | Summarization | | | |
|---|---|---|---|---|---|---|---|---|---|---|---|---|
| | 1-1 | 1-2 | 1-3 | 1-4 | 2-1 | 2-2 | 2-3 | 2-4 | 3-1 | 3-2 | 3-3 | 3-4 |
| Base model | **9.38** (1.00) | **13.84** (1.00) | 24.99 (1.00) | 17.78 (1.00) | 11.73 (1.00) | 14.26 (1.00) | 8.11 (1.00) | **26.43** (1.00) | 13.13 (1.00) | **24.55** (1.00) | 23.20 (1.00) | **10.08** (1.00) |
| H2O | 8.80 (0.94) | 13.48 (0.97) | **25.02** (1.00) | **18.44** (1.04) | 12.36 (1.05) | **14.32** (1.00) | 8.15 (1.01) | 26.22 (0.99) | **13.37** (1.02) | 24.50 (1.00) | 23.20 (1.00) | 9.22 (0.91) |
| L2 | 8.57 (0.91) | 13.40 (0.97) | 24.70 (0.99) | 17.94 (1.01) | **12.77** (1.09) | 13.85 (0.97) | 7.13 (0.88) | 25.74 (0.97) | 12.78 (0.97) | 23.21 (0.95) | 23.35 (1.01) | 8.45 (0.84) |
| NAMM (Ours) | 9.13 (0.97) | 13.53 (0.98) | 24.25 (0.97) | 17.82 (1.00) | 11.45 (0.98) | 13.76 (0.96) | **8.34** (1.03) | 21.79 (0.82) | 12.66 (0.96) | 24.21 (0.99) | **23.56** (1.02) | 8.62 (0.86) |

| Model/Task id | Few-shot Learning | | | | Synthetic | | | Code | | Overall | | |
|---|---|---|---|---|---|---|---|---|---|---|---|---|
| | 4-1 | 4-2 | 4-3 | 4-4 | 5-1 | 5-2 | 5-3 | 6-1 | 6-2 | All tasks | Test tasks | Cache size |
| Base model | 78.00 (1.00) | 92.43 (1.00) | **48.67** (1.00) | **45.50** (1.00) | **22.50** (1.00) | **75.37** (1.00) | 33.89 (1.00) | 74.60 (1.00) | 71.19 (1.00) | **35.22** (1.00) | N/A | 10107 (1.00) |
| H2O | 77.50 (0.99) | 92.43 (1.00) | 48.33 (0.99) | 39.75 (0.87) | 18.12 (0.81) | 64.69 (0.86) | 33.89 (1.00) | 74.61 (1.00) | 71.09 (1.00) | 34.17 (0.97) | N/A | 6662 (0.66) |
| L2 | 76.50 (0.98) | **93.22** (1.01) | 46.15 (0.95) | 36.25 (0.80) | 16.98 (0.75) | 64.34 (0.85) | **36.28** (1.07) | 74.38 (1.00) | 67.43 (0.95) | 33.50 (0.95) | N/A | 6662 (0.66) |
| NAMM (Ours) | **78.50** (1.01) | 92.36 (1.00) | 48.49 (1.00) | **45.50** (1.00) | 19.07 (0.85) | 74.19 (0.98) | 34.28 (1.01) | **74.71** (1.00) | **72.42** (1.02) | 34.70 (0.99) | **0.99** | 8365 (0.83) |

improvements, bringing overall benchmark performance from 1.05% to 11%. We also observe that while our NAMM's memory size is larger than for LongBench, it is considerably lower in relation to the base model's (now only 40%). This result suggests that NAMMs emergently learned a scalable memory strategy, forgetting redundant and detrimental information at an increasing rate with longer contexts without requiring the hand-designed hard cache limits enforced by L2 and H2O.

**ChouBun.** Our new benchmark focuses on tasks designed exclusively in Japanese, a novel language unseen during NAMMs training. We hope this benchmark might itself be a valuable contribution

Table 4: NAMMs evaluation on ChouBun.

| Model/Task | Extractive QA | | | Summarization | Overall | |
|---|---|---|---|---|---|---|
| | JA.WikiQA | JA.EdinetQA | JA.CorpSecQA | JA.CorpSecSum | All tasks | Cache size |
| Base model | 22.91 (1.00) | 28.34 (1.00) | 11.83 (1.00) | 21.75 (1.00) | 21.21 (1.00) | 12099 (1.00) |
| H2O | 20.76 (0.91) | 26.39 (0.93) | 10.42 (0.88) | 21.87 (1.01) | 19.86 (0.94) | 8292 (0.69) |
| L2 | 19.60 (0.86) | 24.06 (0.85) | 8.23 (0.70) | 23.83 (1.10) | 18.93 (0.89) | 8292 (0.69) |
| FastGen | **23.83** (1.04) | 8.23 (0.29) | **19.60** (1.66) | 24.06 (1.11) | 18.93 (0.89) | 8616 (0.71) |
| NAMM (Ours) | 21.34 (0.93) | **28.61** (1.01) | 14.64 (1.24) | **33.15** (1.52) | **24.44** (1.15) | 9895 (0.82) |

to the research community, allowing the assessment of long-context capabilities in multilingual LLMs beyond the already-ubiquitous English and Chinese. We provide further benchmark statistics, details about task composition, together with evaluation metrics for a wider range of popular LLMs in Appendix B.1. In Table 4, we report our results evaluating NAMMs. Once again, we observe a clear contrast with prior hand-designed methods. While integrating either H2O or L2 leads to notable performance drops, NAMMs provides substantial improvements, with overall performance up by 15% from the full-context Llama 3 8B base model.

## 4.2 ZERO-SHOT TRANSFER ACROSS ARCHITECTURES AND MODALITIES

**Cross-scale adaptation.** In Table 5, we provide results zero-shot transferring our NAMM from the Llama 3 8B to the Llama 3 70B model on LongBench. Across all tasks, we find performance to be very close to the full-context baseline with an overall gap of less than 1% even for the

Table 6: Evaluation on the LongVideoBench and MLVU benchmarks with Llava Next Video 7B.

| Model/Task name | LongVideoBench | MLVU | All tasks | Cache size |
|---|---|---|---|---|
| Base model | 43.45 (1.00) | **44.23** (1.00) | 43.84 (1.00) | 7039 (1.00) |
| H2O | 40.91 (0.94) | 43.03 (0.97) | 41.97 (0.96) | 4479 (0.64) |
| L2 | 40.84 (0.94) | 42.07 (0.95) | 41.45 (0.95) | 4479 (0.64) |
| NAMM (Ours) | **44.58** (1.03) | 44.18 (1.00) | **44.38** (1.01) | 5100 (0.72) |

test subset. While NAMMs are not able to improve the overall full-context performance in this first transfer setting outside specific task categories (e.g., coding and few-shot learning), they still outperform both H2O and L2 baselines and retain a similar efficiency as with their original training transformer.

Table 7: Evaluation on D4RL with a Decision Transformer. The normalized performance (in brackets) is calculated using the base model with full cache.

| Model/Task name | Hopper-v3 | | | Walker2d-v3 | | | HalfCheetah-v3 | | | Overall | |
|---|---|---|---|---|---|---|---|---|---|---|---|
| | *Medium* | *Med-Replay* | *Expert* | *Medium* | *Med-Replay* | *Expert* | *Medium* | *Med-Replay* | *Expert* | *All tasks* | *Cache size* |
| Base model | 33.36 (1.00) | 18.37 (1.00) | 44.62 (1.00) | 68.21 (1.00) | 7.18 (1.00) | 38.98 (1.00) | **34.91** (1.00) | 5.06 (1.00) | 10.64 (1.00) | 29.04 (1.00) | 3000 (1.00) |
| H2O | 33.19 (1.00) | 17.86 (0.97) | 49.10 (1.10) | 67.63 (0.99) | **7.59** (1.06) | 40.03 (1.03) | 26.73 (0.77) | 4.46 (0.88) | 11.74 (1.10) | 28.70 (0.99) | 2048 (0.68) |
| L2 | 32.85 (0.98) | 17.96 (0.98) | 43.75 (0.98) | 65.47 (0.96) | 7.18 (1.00) | 40.64 (1.04) | 30.10 (0.86) | 4.76 (0.94) | 8.52 (0.80) | 27.91 (0.96) | 2048 (0.68) |
| NAMM (Ours) | **36.10** (1.08) | **18.86** (1.03) | **49.39** (1.11) | **70.87** (1.04) | 7.53 (1.05) | **50.02** (1.28) | 34.56 (0.99) | **5.90** (1.17) | **12.34** (1.16) | **31.73** (1.09) | 2434 (0.81) |

**Vision Language Understanding.** In Table 6, we provide results zero-shot transferring to the computer vision domain, evaluating NAMMs with a Llava Next Video 7B model (Zhang et al., 2024b) on LongVideoBench (Wu et al., 2024) and Multi-Task Long Video Understanding (MLVU) (Zhou et al., 2024). As when evaluated atop Llama 8B, our NAMM is the only method recording gains over the full-context base transformer in both benchmarks. Furthermore, we find that NAMMs learns to forget almost exclusively parts of redundant video frames rather than the language tokens describing the final prompt, even though they were never faced with such modality during training. This result validates that our NAMM recovered a domain-agnostic memory management strategy, further highlighting their flexibility.

**Reinforcement learning.** In Table 7, we provide our zero-shot transfer results for the offline reinforcement learning setting, where we apply NAMMs atop a decision transformer (Chen et al., 2021b) using the open-sourced models from Beeching & Simonini (2022) pre-trained on the canonical the continuous-control tasks from D4RL (Fu et al., 2020). We find our NAMM improves the base transformer quite considerably in this domain across eight out of nine offline tasks with over 9% overall gains, opposing the performance loss of the other efficient baselines. We posit that since the nature of the decision transformer optimization is closely tied to behavior cloning, the ability to discard part of the context is likely to allow NAMMs to *forget* and avoid imitating previous mistakes autoregressively. In support of this hypothesis, we observed slightly higher average rewards in the transitions for the retained tokens (by 1.4%, 0.8%, and 12.3% for the Hopper, Walker2d, and HalfCheetah environments, respectively).

**NAMMs comparison.** In Table 8, we provide summarized results comparing NAMMs with either BAM or the simpler MLP architecture at the end of each stage of incremental evolution. First, we note that even the MLP NAMM after stage 1 impressively improves performance across all language benchmarks. Additionally, performance sees near-monotonic improvements with each additional stage of incremental evolution in both language and zero-shot transfer settings. Comparing our implementations, the performance benefits from the memory models with BAM appear consistently superior to the MLP. Moreover, on ChouBun. we observe that the performance with BAM sees a notable upswing after the second stage of incremental training, which might be associated with the introduction of another ideogram-based language in the training set.[2] The same improvement not occurring with the MLP-based NAMMs might be further evidence of architectural performance saturation, highlighting the effectiveness of our main implementation.

Table 8: Summarized comparison of different NAMMs in language modeling (top) and zero-shot transfer (bottom)

| Model/Eval | LongBench | | InfiniteBench | | ChouBun | |
|---|---|---|---|---|---|---|
| | Performance | Cache size | Performance | Cache size | Performance | Cache size |
| Base model | 28.86 (1.00) | 32768 (1.00) | 1.05 (1.00) | 32747 (1.00) | 21.21 (1.00) | 12099 (1.00) |
| NAMM (MLP, s1) | 28.83 (1.05) | 7639 (0.23) | 3.08 (2.93) | 11329 (0.35) | 22.09 (1.04) | 9525 (0.79) |
| NAMM (MLP, s2) | 29.22 (1.07) | 8475 (0.26) | 4.00 (3.80) | 13031 (0.40) | 22.06 (1.04) | 9815 (0.81) |
| NAMM (BAM, s1) | 28.91 (1.05) | 7951 (0.24) | **10.14** (9.63) | 11173 (0.34) | 22.73 (1.07) | 9569 (0.79) |
| NAMM (BAM, s2) | 29.25 (1.07) | 8267 (0.25) | 9.78 (9.29) | 12789 (0.39) | 24.05 (1.13) | 9867 (0.82) |
| NAMM (BAM, s3) | **29.33** (1.11) | 8155 (0.25) | **11.00** (10.45) | 13192 (0.40) | 24.44 (1.15) | 9895 (0.82) |

| Model/Eval | Llama 3 70B | | Computer Vision | | Reinforcement Learning | |
|---|---|---|---|---|---|---|
| | Performance | Cache size | Performance | Cache size | Performance | Cache size |
| Base model | 35.22 (1.00) | 10107 (1.00) | 43.84 (1.00) | 7039 (1.00) | 29.04 (1.00) | 3000 (1.00) |
| NAMM (MLP, s1) | 34.11 (0.97) | 7930 (0.78) | 40.44 (0.92) | 584 (0.08) | 29.30 (1.01) | 1993 (0.66) |
| NAMM (MLP, s2) | 34.29 (0.97) | 8445 (0.84) | 40.39 (0.92) | 713 (0.10) | 29.58 (1.02) | 2834 (0.94) |
| NAMM (BAM, s1) | 34.11 (0.97) | 7947 (0.79) | 41.52 (0.95) | 723 (0.10) | 30.44 (1.05) | 2009 (0.67) |
| NAMM (BAM, s2) | 25.20 (0.72) | 8276 (0.82) | **44.63** (1.02) | 4948 (0.70) | 31.53 (1.09) | 2534 (0.84) |
| NAMM (BAM, s3) | **34.70** (0.99) | 8365 (0.83) | 44.38 (1.01) | 5100 (0.72) | **31.73** (1.09) | 2434 (0.81) |

## 4.3 UNDERSTANDING NEURAL ATTENTION MEMORY MODELS

**Influence of layer depth.** We begin analyzing NAMMs by focusing on the final amount of retained tokens and their oldness[3]. At the top of Figure 6, we provide these normalized metrics as a function

---

[2]The DuReader task, used in the second stage of incremental training, uses the Chinese language.

[3]We define *oldness* of a retained token as the number of new queries since its introduction in the KV cache.

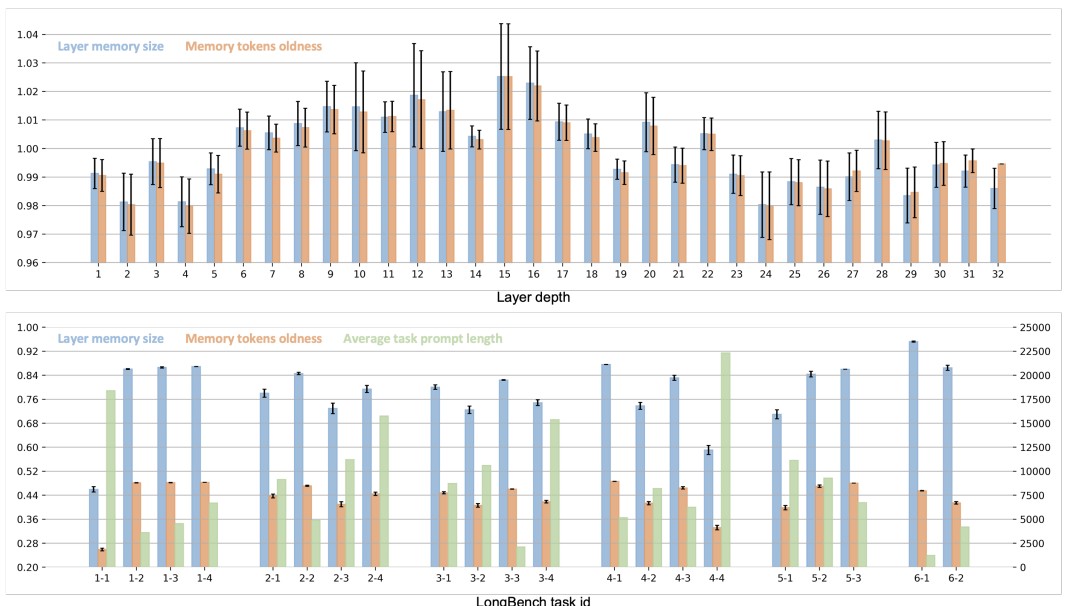

Figure 6: Memory size and token oldness as recorded for each layer in the base model (top) and for each task in LongBench (bottom). We normalize these statistics per task using either their average across all task prompts (top) or the mean sample length (bottom).

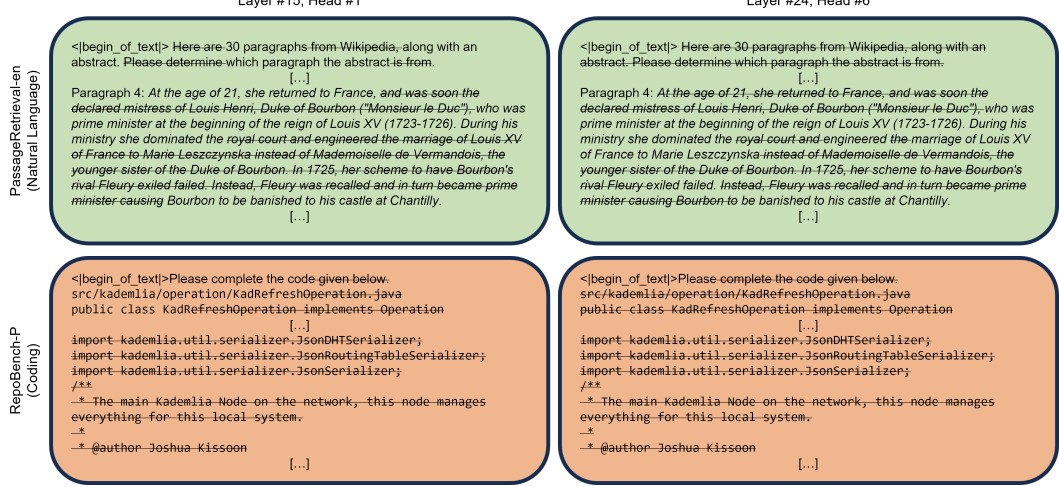

Figure 7: Qualitative inspection of the text from decoding the retained tokens in the KV cache after applying NAMMs. We compare the behavior of NAMMs across the layers with the highest (left) and lowest average retained tokens (right), for either a natural language (top) or coding task (bottom).

of layer depth. Interestingly, our learned NAMM does not appear to affect the KV cache uniformly, retaining visibly more and older tokens for some of the early-middle layers of the base transformer. One possible interpretation of our results, complementing recent analysis (Wendler et al., 2024), is that these layers might be particularly important for processing and aggregating information over longer contexts, thus requiring larger memories than the rest.

**Influence of task structure.** At the bottom of Figure 6, we instead provide these metrics while varying the source task, this time normalized by the average prompt lengths shown in green. Our results illustrate an inverse correlation between normalized memory size and prompt length (with a Pearson coefficient of -0.84), further confirming our earlier observations of sub-linear memory growth and favorable scaling to longer contexts. Additionally, we observe that in the code completion tasks (with task id 6-1 and 6-2) NAMMs learn to preserve visibly more tokens relative to their average prompt lengths. This result appears intuitively consistent with the higher information density in code, leaving room for less redundancy as opposed to natural language.

**Selected qualitative examples.** We qualitatively find these analyzed trends by inspecting the text corresponding to the forgotten tokens for a few selected prompts. In particular, we consider the layers with the highest and lowest average retained tokens (15 and 24), for tokens from either a natural language or coding task (PassageRetrieval-en, id 5-1, and RepoBench-P, id 6-2). As shown in Figure 7, for early-middle layers, NAMMs tend to focus on retaining global information such as the task preamble and key words throughout the text. Instead, for later layers, NAMMs seem to forget many of these tokens, whose information has likely been already incorporated in the previous layers, allowing the transformer to focus more on tokens with more detailed local information. Furthermore, in coding tasks, we find that the pruned tokens are mostly contiguous, corresponding to whitespace, comments, and whole segments of boilerplate code. This is in contrast to natural language tasks, where NAMMs appear trying to exploit some of the grammatical redundancies of the English syntax often dropping specific tokens mid-sentences.

**Additional analysis.** We provide additional analytic results in Appendix D. For instance, we analyze how the presence of each token in memory affects the scores of the other tokens, we compare the generated responses before and after the introduction of NAMMs in a very long context task, and show the sensitivities of the token scores for each input feature. These results show that NAMMs learn mechanisms for 'cross-token' competition relying on high-frequency components of the attention matrices and illustrate how they learn to overcome different failure modes of long context LMs, further evidencing the need to go beyond simple strategies and the potential of end-to-end learning for token-level memory systems.

## 5 RELATED WORKS

Devoto et al. (2024) and Yao et al. (2024) try to identify the least important tokens to evict using heuristics such as L2 magnitude and entropy to improve efficiency. Alternative strategies include considering simple statistics from the attention matrix (Liu et al., 2024b; Oren et al., 2024; Zhang et al., 2024c). Ge et al. (2024) and Li et al. (2024b) build on these ideas by applying multiple strategies based on matching specific attention patterns. Motivated by similar considerations, Nawrot et al. (2024) proposed directly fine-tuning the original base transformer to compress the KV cache while minimizing a regularization loss to preserve the original base model's behavior and limit performance degradation. Complementary to our method, MQA (Shazeer, 2019) and GQA (Ainslie et al., 2023) propose merging attention heads during training to improve inference throughput. Similarly, also KV cache quantization is another orthogonal area where different hand-designed strategies have been proposed (Hooper et al., 2024; Dong et al., 2024a;b), with even recent work empirically showing their direct compatibility with token eviction methods (Liu et al., 2024a). We note that, unlike this prior work, our approach uniquely learns a black-box model to *maximize performance* through token-level memory management and shows potential for providing improvements to both the effectiveness and efficiency of transformers. We refer to App. E for references and to the wider literature, including efficient architectures, memory, and evolution.

## 6 DISCUSSION AND FUTURE WORK

This work introduced Neural Attention Memory Models, providing a new framework to enhance the performance of transformers while significantly reducing memory footprint. By evolving NAMMs on top of pre-trained LMs, we demonstrated their effectiveness across diverse long-context tasks in three languages, significantly surpassing previous hand-designed KV cache eviction frequently hindering performance, and the original model relying on costly full-context conditioning. Our carefully designed approach also enabled NAMMs, trained solely on language tasks, to achieve zero-shot transferability across architectures, input modalities, and task domains. While NAMMs do appear to provide benefits beyond what achieved with hand-designed strategies, we believe there is much room for improvement (e.g., see Limitations F). This work has only begun to explore the design space of our memory models, which we anticipate might offer many new opportunities to advance future generations of transformers. In this regard, we believe NAMMs should not be viewed as a replacement for gradient-based optimization, but rather an orthogonal framework that could be combined and alternated with parameter fine-tuning. Such an extension has the potential to unlock efficient long-context training, drawing parallels to the iterative process of learning and evolution that shaped human memory.

## 7 AUTHOR CONTRIBUTIONS

Edoardo Cetin initiated the project, led the design and implementation of NAMMs, and provided major contributions to writing. Qi Sun designed and implemented the zero-shot transfer experiments with Llama 3 70B and Llava Next Video 7B, and provided contributions and feedback to writing. Tianyu Zhao devised and implemented ChouBun, and provided contributions and feedback to writing. Yujin Tang coordinated the project, gave key advice for the design of NAMMs, and provided major contributions to writing.

## ACKNOWLEDGEMENTS

The authors would like to thank David Ha and Llion Jones for providing valuable discussions during the early stages and feedback while drafting the text. This paper is based on results obtained from a project, JPNP20017, subsidized by the New Energy and Industrial Technology Development Organization (NEDO).

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

Table 9: NAMMs hyper-parameters used for training and evaluation. The omitted CMA-ES hyper-parameters can be obtained by following the recommended default calculation by Hansen (2006).

| NAMMs hyperparameters | |
| --- | --- |
| Spectrogram window size $n_w$ | 32 |
| Spectrogram window stride $s_w$ | 16 |
| Spectrogram window type | Hann |
| Spectrogram EMA reduction coefficient $\gamma$ | $0.99^{16}$ |
| Positional features | 8 |
| NAMMs execution delay | 512 |
| NAMMs non-linearity | ReLU |
| **Optimization hyperparameters**, notation from Hansen (2006) | |
| Evolution algorithm | CMA-ES |
| Elite ratio | 0.5 |
| Mean coefficient $c_m$ | 1 |
| Initial step size $\sigma$ | 0.65 |
| Samples batch size per-task | 64 |
| Population size | 32 |
| Task for incremental stage 1 | PassageRetrieval-en |
| Task for incremental stage 2 | DuReader |
| Task for incremental stage 3 | NarrativeQA |
| **BAM-specific** | |
| Hidden dimensions | 16 |
| Use bias | True |
| Masking strategy | counter-causal |
| Number of attention layers | 1 |
| Number of final linear layers | 1 |
| Use residual connections | True |
| Use multiplicative interactions | True |
| **MLP-specific** | |
| Hidden dimension | 25 |
| Number of hidden layers | 2 |
| Use residual connections | True |

## A  IMPLEMENTATION DETAILS

### A.1  MODEL SPECIFICS AND NAMMS EXECUTION

We evolve our Neural Attention Memory Models on top of a context-extended Llama 3 8B (Dubey et al., 2024) base model. In particular, we employ the NTK-aware positional interpolation strategy (bloc97, 2023) to extend the context by four times from 8192 to 32768. Unlike prior strategies that require further gradient fine-tuning to avoid performance collapse (Chen et al., 2023), NTK-aware positional interpolation has been shown to produce sensible results even when applied zero-shot. In case the length of a task prompt still exceeds 32768 we perform mid-sentence cropping (Xiao et al., 2023; Jin et al., 2024), as standard in long-context LM evaluation (Bai et al., 2023; Zhang et al., 2024a).

When applying NAMMs, we only affect the execution of the base model with a fixed frequency, once every $n_{up} = 512$ steps. When feeding longer prompts to our model, we simply split the tokens into $n_{up}$-sized chunks. We note that due to modern frameworks being bound primarily by memory constraints, input-splitting in itself has minimal effects on running time, with similar approaches being already performed under the hood by established kernel procedures (Dao et al., 2022).

### A.2  FEATURE EXTRACTION AND ARCHITECTURE DETAILS

Our new feature extraction framework is a key component for enabling the transfer properties of NAMMs. In practice, we extract the attention spectrogram from the real-valued attention matrix using a Hann window of size $n_w = 32$, resulting in just seventeen complex-values frequencies that

we convert to real numbers by simply taking their magnitude, yielding each $\omega_i^t \in \mathbb{R}^{17}$. We use a stride of half the window size $s_w = 16$, producing $n_T = n_{up}/s_w = 32$ frequency representations over the time axis of the attention matrix from the latest chunk of $n_{up}$ queries, $\omega_i^1, \ldots, \omega_i^{n_T}$. Thus, we reduce these frequency representations over the time axis via an element-wise exponentially moving average operation. We note that our EMA does not only consider the $n_T$ representations computed for the frequency of each token in the $n_{up}$-sized chunk of the latest queries, but also the discounted EMA at the *previous execution step* or our memory for each retained token, denoted $\omega_i'$. Thus, each of our reduced spectrogram representations reflects the full history of previous attention values:

$$\omega_i = \left(\sum_{t=1}^{n_T} \gamma^{t-1}\omega_i^t\right) + \gamma^{n_T}\omega_i', \tag{4}$$

where we use $\gamma$ to denote the EMA's discount factor. To expedite learning the weights of our architecture, we ensure all spectrogram features have unit variance at initialization across our training data, using the statistics of the base Llama 3 model computed on the first task employed in incremental learning (PassageRetrieval). Finally, we also concatenate a small eight-dimensional sinusoidal positional embedding using the *oldness* of each token, i.e., the amounts of new queries observed since its introduction in the KV cache. We provide an extended summarized pseudocode description of the execution pipeline in Algortihm 2 (to complement Algorithm 1 in the main text).

---

**Algorithm 2** NAMMs

    **Input:** Attention matrix $\mathbf{A}$, Hann window size $n_w$, Stride size $s_w$, Update interval $n_{up}$
    **Output:** Memory score for each token

1: Initialize token score array $S$
2: Split input tokens into chunks of size $n_{up}$
3: **for** each input chunk **do**
4:     Extract attention matrix $\mathbf{A}_k$ from the latest $n_{up}$ queries
5:     **for** column $i$ in $\mathbf{A}_k$ **do**
6:         Apply STFT of window size $n_w$ and stride $s_w$ to $\mathbf{A}_k[:, i]$
7:         Compute spectrograms $\omega_i^t \in \mathbb{C}^{17}$ for $t = 1, \ldots, n_T$         $\triangleright n_T = n_{up}/s_w$
8:         Calculate $\omega_i$ with Equation 4
9:         Update memory $\omega_i' \leftarrow \omega_i$
10:         Normalize spectrogram features $\omega_i$
11:         Concatenate positional embedding to $\omega_i$
12:         Update $S$ with BAM predicted score $m_\phi(\omega_i)$
    **return** $S$

---

Our backward-attention memory network processes these representations by directly first applying the self-attention layer employing the counter-autoregressive backward masking introduced in Section 3, designed to facilitate asymmetric interactions between tokens in memory. The output of self-attention is then fed to a single final linear layer to obtain the final score. We employed a few important additional design choices following some preliminary testing. First, motivated by efficiency

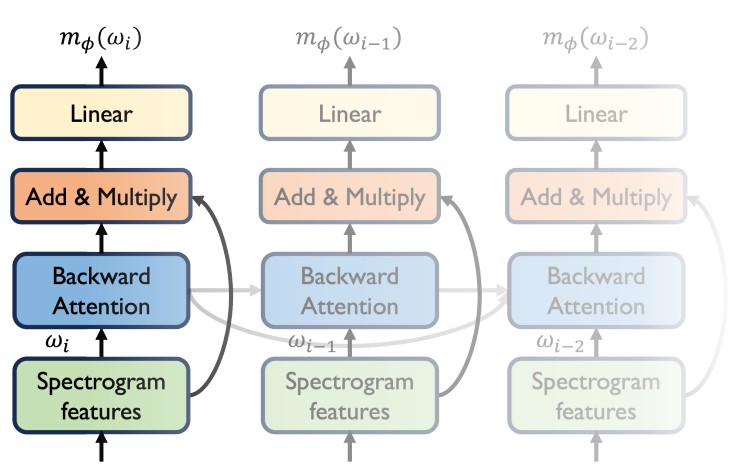

Figure 8: Schematic depiction of the components of our Neural Attention Memory Models, denoted $m_\phi$, parameterized with our BAM architecture. The spectrogram representation of each token, denoted $\omega_i$, is processed by an attention layer followed by a simple linear operation to output its relative score. Backward masking introduces asymmetry, ensuring that each token can only attend to its future relatives.

considerations, we use a single head within our attention mechanism and no layer normalization. Second, our attention layer produces outputs that are twice the dimensionality of the spectrogram features. These outputs are integrated back into the main network before the final linear layer via both residual and multiplicative interactions. We provide a schematic depiction of our minimal architecture in Figure 8. Through our minimalist design choices, our full network comprises only just over four thousand learnable parameters, a negligible amount, orders of magnitudes lower than even a single layer in modern transformers.

### A.3 ZERO-SHOT TRANSFER

For our zero-shot transfer experiments, we consider a Llama 3 transformer with 70B parameters (Dubey et al., 2024), a Llava Next Video transformer with 7B parameters (Zhang et al., 2024b), and a decision transformer (Chen et al., 2021b) with about 1M parameters. For our 70B experiments, we follow the exact same setup as when evaluating our 7B Llama model used in training. For our video-language model, we extract $12 \times 12$ image tokens from 48 uniformly sampled frames, 6912 in total. We also slightly shift the selection score threshold by 5, to counteract the lower number of total tokens and get a comparable average cache size to the L2 and H2O baselines. We adapt the code and follow the standardized experimental setup from Li et al. (2024a). For the reinforcement learning experiments, we encode each state, action, and return-to-go into separate tokens and do not apply any restrictions or modifications to our standard NAMM LM setup. We average the performance collected over 20 random seeds to account for the stochasticity of the initial state in the Gym Mujoco environments (Brockman et al., 2016). Rather than re-training a decision transformer from scratch, our RL experiments adapt the open-sourced checkpoints and implementation provided by Beeching & Simonini (2022). We would like that note that on some task-dataset combinations of D4RL, these checkpoints appear to yield lower performance than what was reported in the original decision transformer paper (e.g., Walker pre-trained on medium-expert data) Chen et al. (2021b). However, we do not believe these differences should affect our conclusions as we used the same base model for all our memory management baselines.

### A.4 EVOLUTIONARY OPTIMIZATION

As described in Section 3, we optimize NAMMs with the Covariance Matrix Adaptation Evolution Strategy (CMA-ES) (Hansen, 2006). Being an evolutionary algorithm, CMA-ES does not require any gradient information and can directly optimize black-box undifferentiable metrics. This property allows us to both optimize for the non-differentiable token selection task of our NAMMs and also maximize non-differentiable task performance metrics directly. In the case of the Long-Bench (Bai et al., 2023) tasks considered for training, these metrics correspond to exact match accuracy (PassageRetrieval-en), ROUGE-L score (DuReader), and F1 score (NarrativeQA).

On a high level, given a neural network with $P$ parameters, CMA-ES maintains a mean vector $\mu \in \mathbb{R}^P$ and a covariance matrix $\Sigma \in \mathbb{R}^{P \times P}$. Then, it repeats the following steps:

1. **Sampling.** CMA-ES generates a population of neural networks, sampling their parameters from the multivariate normal $N(\mu, \Sigma)$ distribution.

2. **Evaluation.** Each population candidate is evaluated to the objective function for the objective function used.

3. **Updating.** By both selecting a subset of the population candidates and also weighting them based on their overall ranking the mean and covariance are updated towards higher-performing regions of the search space.

We provide the main hyper-parameters in Table 9 and refer to either the work by Hansen (2006) or our shared code for the full implementation details.

### A.5 FASTGEN IMPLEMENTATION AND TUNING

We re-implemented the recent FastGen method proposed by Ge et al. (2024), which proposes to adopt a hand-designed combination of different strategies targeted to retain tokens with high attention values, belonging to recent words, or encoded from particular grammatical features (i.e., punctuation, 'special tokens'). In particular, after observing the input prompt, FastGen performs a 'profiling step' where the strategy able to evict the most amount of tokens is selected such that:

$$|A - \hat{A}|_2 < 1 - \mathrm{T}. \tag{5}$$

Here, $A$ is the full-cache attention matrix, $\hat{A}$ is the 'reconstructed' attention matrix re-calculated after performing a softmax between each layer's queries and keys with masked-out entries for the keys evicted by the individual strategies. Furthermore, T is the main threshold hyper-parameter, determining how aggressively FastGen is allowed to prune tokens even if resulting in degradation to the attention-reconstruction heuristic.

We note that, unlike our other baselines, FastGen is only directly compatible with language modeling tasks. This is because one of the main ways it differs from H2O is by preserving particular grammar-based tokens in some of its strategies (e.g., punctuation, special words, etc.). Thus, as this baseline was specifically designed for LMs rather than arbitrary transformers, we did not consider applying it in the 0-shot transfer settings, and only focused on Llama 3 8B.

We note that as we are dealing with much longer prompts (sometimes far beyond tens/hundreds of thousand tokens), for efficiency consideration, we performed the profiling steps in our re-implementation after the first 4096 tokens any prompt exceeds this length. We also found to avoid losing too much performance over the base model on longer context tasks we had to retune its main 'threshold.' We selected T=0.999, as this choice allowed FastGen to retain over 95% normalized performance while still discarding a non-trivial portion of tokens on all LongBench, as shown in Figure 9. Other than the main threshold for attention reconstruction, FastGen has two other main hyperparameters: the 'recency ratio,' the 'attention ratio' determining the portion of most recent tokens or with the highest attention values to retain in its individual strategies. We set these hyper-parameters to 0.3, following the paper's recommendation.

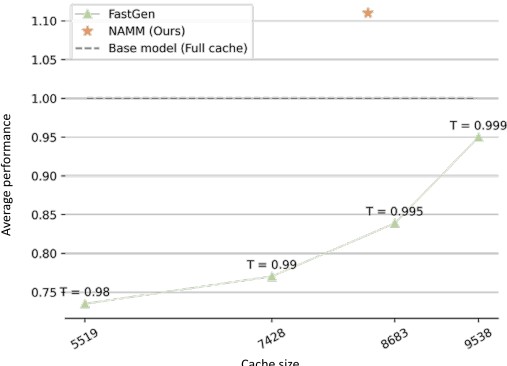

Figure 9: Averaged normalized performance and cache size of FastGen as compared to NAMM over LongBench when varying the threshold parameter T.

Table 10: Statistics of ChouBun. Lengths are counted by tokens produced by Llama 3 tokenizer.

| Statistics | Extractive QA | | | Summarization | Overall |
|---|---|---|---|---|---|
| | *JA.WikiQA* | *JA.EdinetQA* | *JA.CorpSecQA* | *JA.CorpSecSum* | **All tasks** |
| Number of documents | 20 | 20 | 30 | 30 | 70 |
| Number of QA pairs | 200 | 390 | 150 | 30 | 770 |
| Number of reference answers | 1 | 1 | 1 | 5 | 1 or 5 |
| Document length max. | 13027 | 10152 | 85981 | 85981 | 85981 |
| Document length mean | 10131 | 8994 | 26220 | 26220 | 13317 |
| Document length min. | 8196 | 6825 | 5640 | 5640 | 5640 |
| Answer length max. | 40 | 208 | 30 | 140 | 208 |
| Answer length mean | 7 | 11 | 8 | 80 | 21 |
| Answer length min. | 1 | 1 | 1 | 55 | 1 |

Table 11: Performance of a wider range of LLMs on the ChouBun benchmark.

| Model/Task name | Extractive QA | | | Summarization | Overall | |
|---|---|---|---|---|---|---|
| | *JA.WikiQA* | *JA.EdinetQA* | *JA.CorpSecQA* | *JA.CorpSecSum* | **All tasks** | **Max. length** |
| `mistralai/Mistral-7B-v0.1` | 8.68 | 8.34 | 16.25 | 10.50 | 10.94 | 32768 |
| `rinna/llama-3-youko-8b` | 16.68 | 12.23 | 17.03 | 22.27 | 17.05 | 8192 |
| `meta-llama/Meta-Llama-3-8B` | 14.58 | 14.77 | 16.86 | 22.84 | 17.27 | 8192 |
| `meta-llama/Llama-2-7b-hf` | 16.77 | 9.92 | 20.86 | 21.97 | 17.38 | 2048 |
| `01-ai/yi-6b-200k` | 30.36 | 23.64 | 38.09 | 21.11 | 28.30 | 200000 |
| `elyza/Llama-3-ELYZA-JP-8B` | 20.77 | 21.45 | 35.59 | 40.21 | 29.50 | 8192 |

## B    BENCHMARK DESCRIPTIONS

### B.1    CHOUBUN DETAILS

The ChouBun benchmark is created to assess the generalization ability of NAMMs to a new language (Japanese), but we hope it will also serve as a standard benchmark for Japanese LLMs. The benchmark is composed of two task categories — extractive QA and abstractive summarization — and four tasks as follows.

- *JA.WikiQA* is an extractive QA task about 20 randomly sampled articles from the 20240429 dump of Japanese Wikipedia[4]. Each article corresponds to 10 QA pairs, and there are 200 QA pairs in total.

- *JA.EdinetQA* is an extractive QA task based on 20 security reports from EDINET[5]. The EDINET security reports are in CSV format, which makes them less human-readable. Nevertheless, we choose not to convert the format because the conversion process per se is non-trivial, and using a CSV-style text input helps us evaluate a model's capability of understanding structured data. The total number of QA pairs in *JA.EdinetQA* is 390.

- *JA.CorpSecQA* is another extractive QA task based on 30 security reports downloaded from three corporation websites (MUFG[6], NTT[7], and Toyota[8]). We extract texts from original file in PDF format. There are 150 QA pairs in total.

- *JA.CorpSecSum* is an abstractive summarization task based on the same data of *JA.CorpSecQA*. Each document corresponds to one data point, and we collect 5 reference summaries for each data point.

---

[4]https://dumps.wikimedia.org/other/cirrussearch/
[5]https://disclosure2.edinet-fsa.go.jp/
[6]https://www.mufg.jp/ir/report/security_report/
[7]https://group.ntt/jp/ir/library/results/
[8]https://global.toyota/jp/ir/library/securities-report/

**Prompt for extractive QA**

抽出型の長文QAモデルのトレーニングデータを作成しています。
コンテキストとして長い文書を提供します。
文書を注意深く読み、分析し、20個の質問と回答のペアを生成してください。
生成されたQAペアの要件は以下の通りです。\n
1. 回答は文書からのテキストの一部でなければなりません。\n
2. 回答は短く簡潔なテキストの一部であるべきです。\n
3. 質問は多様で、文書の異なる側面をカバーすべきです。\n
4. 回答は、直接文章の内容を引用してください。余計な情報は含めないでください。\n
以下の形式で20個の質問と回答のペアを直接回答してください：\n
### 質問 1 ###\n
{question_1}\n
### 回答 1 ###\n
{answer_1}\n
...\n
### 質問 20 ###\n
{question_20}\n
### 回答 20 ###\n
{answer_20}\n
文書は以下の通りです：\n
### 文書 ###\n
{doc_text}

**Prompt for abstractive summarization**

抽象的な長文要約モデルのトレーニングデータを作成しています。
コンテキストとして長い文書を提供します。
文書を注意深く読み、分析し、5個の要約例を生成してください。
以下は生成される要約の要件です。\n
1. 各要約は短く簡潔であるべきです。\n
2. 各要約は文書の一般的なアイデア、トレンド、洞察を網羅すべきです。\n
3. すべての要約は内容が同一でありながら、表現が多様であるべきです。\n
以下の形式で5個の要約を直接返信してください：\n
### 要約 1 ###\n
{summary_1}\n
...\n
### 要約 5 ###\n
{summary_5}\n
文書は以下の通りです：\n
### 文書 ###\n
{doc_text}

Figure 10: LLM prompts for generating synthetic QA pairs and summaries in ChouBun.

Collecting human annotations for long-text tasks is challenging, therefore we use synthetic QA pairs and summaries. In particular, we prompt various LLMs[9] to generate multiple question-answer pairs or summaries for each document. Different instructions are designed for the two tasks and they are shown in Figure 10. To improve the reliability of the synthetic data, we ensure that every answer in extractive QA tasks is a text span presented in its corresponding source document. In Table 10, we provide the statistics of the benchmark.

We use F1 score and ROUGE score for evaluation in the extractive QA tasks and summarization task, respectively. Reference text and hypothesis text are pre-tokenized by the MeCab tokenizer[10]. A wider range of LLMs' performance on the ChouBun benchmark is presented in Table 11.

---

[9]`gpt-4o-2024-05-13`, `gpt-4o-mini-2024-07-18`, `gpt-4-turbo-2024-04-09`, and `claude-3-5-sonnet-20240620`

[10]`https://github.com/polm/fugashi`

## B.2 BENCHMARKS SUMMARY

We provide a summary of the types of tasks and domains of the other benchmarks we considered for our experiments. We refer interested readers to the relative referenced papers for full details.

**LongBench (Bai et al., 2023).** This benchmark comprises 21 different tasks targeted to evaluate the long-context capabilities of LMs. These tasks include both English and Chinese and come from either modified/subsampled versions of existing datasets or synthetic generation. The authors divided them in 6 categories, numbered with the prefixes 1 to 6: single-document QA, multi-document QA, summarization, few-shot learning, synthetic, and code. The tasks have a reported average length of 6711 English words and 13386 Chinese characters.

**InfiniteBench (Zhang et al., 2024a).** This benchmark comprises 12 different tasks designed to go beyond the existing benchmarks and push the limits in long-context LMs. In fact, while popular prior long context benchmarks, including LongBench, focus on prompts of around 10K tokens InfiniteBench considers tasks with contexts beyond 100K tokens. These tasks again include both English and Chinese and come from either modified/subsampled versions of existing datasets or synthetic generation. The authors divided them into 5 categories: retrieval, dialogue, novel, math, and code. We note some of these tasks are considered extremely difficult, with even powerful proprietary LMs such as GPT4 not able to get above a performance of 1%.

**LongVideoBench (Wu et al., 2024).** This benchmark comprises 3763 curated long videos with subtitles. These videos are coupled with 6678 human-annotated questions focusing on 17 different categories. The benchmark is focused on what the authors refer to as frame-specific 'reasoning' style questions. In particular, for these kinds of questions, video language models are tasked to respond to 'referred queries' targeting particular parts of the whole video context.

**Multi-task Long Video Understanding Benchmark (Zhou et al., 2024).** This benchmark focuses on evaluating long-video understanding performance. It includes videos averaging 12 minutes in length up to 2 hours. The videos span different genres such as movies, documentaries, surveillance videos, ego-centric videos, games, and cartoons. In total, this benchmark comprises 2593 evaluation problems divided into 9 categories: topic reasoning, anomaly recognition, video summarization, needle question answering, ego reasoning, plot question answering, sub-scene captioning, action count, and action order. These problems are quite diverse including both multi-choice and generation-style questions for video language models.

**D4RL (Fu et al., 2020).** This benchmark focuses on evaluating offline reinforcement learning agents (Lange et al., 2012). In particular, it provides pre-training datasets for different reinforcement learning tasks simulated through Mujoco based on OpenAI gym (Brockman et al., 2016). The datasets are named based on the displayed agent skills (e.g., expert medium), and based on their inclusion of 'replay data' from the demonstrator agent's own prior learning experiences. Evaluation is then performed after pre-training by running the learned agents online in the respective environments. We focus on the most popular subset of this benchmark, involving continuous-control tasks with three different agents: Hopper, HalfCheetah, and Walker-2d, evaluating the agent after pre-training on Expert, Medium, and Medium Replay data. Rather than re-training from scratch, we use the open-sourced checkpoints from Chen et al. (2021b) and focus on the evaluation aspect of the benchmark.

Table 12: NAMMs evaluation on LongBench (Bai et al., 2023). The normalized performance (in brackets) is calculated using the base model with full cache. The aggregate test task performance of NAMMs models is taken by averaging the normalized scores on the tasks not used for incremental evolution.

| Model/Task id | Single-Doc QA | | | | Multi-Doc QA | | | | Summarization | | | |
|---|---|---|---|---|---|---|---|---|---|---|---|---|
| | 1-1 | 1-2 | 1-3 | 1-4 | 2-1 | 2-2 | 2-3 | 2-4 | 3-1 | 3-2 | 3-3 | 3-4 |
| Base model | 10.38 (1.00) | 12.79 (1.00) | 22.60 (1.00) | 21.31 (1.00) | 10.41 (1.00) | 12.67 (1.00) | 7.54 (1.00) | 25.86 (1.00) | 29.34 (1.00) | 23.93 (1.00) | 0.92 (1.00) | 2.66 (1.00) |
| NAMM (MLP, s1) | 7.60 (0.73) | 12.74 (1.00) | 22.74 (1.01) | 21.08 (0.99) | 9.58 (0.92) | 12.24 (0.97) | 6.48 (0.86) | 19.41 (0.75) | 27.76 (0.95) | 23.61 (0.99) | 0.95 (1.03) | 3.44 (1.29) |
| NAMM (MLP, s2) | 6.76 (0.65) | 12.77 (1.00) | 23.74 (1.05) | 20.56 (0.96) | 9.69 (0.93) | 12.21 (0.96) | 6.93 (0.92) | 22.40 (0.87) | 27.30 (0.93) | 24.20 (1.01) | 1.72 (1.87) | 2.78 (1.05) |
| NAMM (BAM, s1) | 5.77 (0.56) | 12.76 (1.00) | 22.94 (1.02) | 21.55 (1.01) | 9.47 (0.91) | 12.21 (0.96) | 6.51 (0.86) | 18.73 (0.72) | 28.06 (0.96) | 23.97 (1.00) | 1.01 (1.10) | 4.00 (1.50) |
| NAMM (BAM, s2) | 7.08 (0.68) | 12.70 (0.99) | 22.21 (0.98) | 21.50 (1.01) | 9.94 (0.95) | 12.21 (0.96) | 7.13 (0.95) | 20.34 (0.79) | 28.87 (0.98) | 23.84 (1.00) | 0.92 (1.00) | 3.94 (1.48) |
| NAMM (BAM, s3) | 9.14 (0.88) | 12.63 (0.99) | 21.94 (0.97) | 21.34 (1.00) | 9.71 (0.93) | 11.63 (0.92) | 6.98 (0.93) | 20.58 (0.80) | 28.78 (0.98) | 24.39 (1.02) | 1.04 (1.13) | 3.63 (1.36) |

| Model/Task id | Few-shot Learning | | | | Synthetic | | | Code | | Overall | | |
|---|---|---|---|---|---|---|---|---|---|---|---|---|
| | 4-1 | 4-2 | 4-3 | 4-4 | 5-1 | 5-2 | 5-3 | 6-1 | 6-2 | All tasks | Test tasks | Cache size |
| Base model | 73.00 (1.00) | 89.45 (1.00) | 46.54 (1.00) | 40.00 (1.00) | 1.48 (1.00) | 12.18 (1.00) | 28.80 (1.00) | 69.09 (1.00) | 65.17 (1.00) | 28.86 (1.00) | N/A | 32768 (1.00) |
| NAMM (MLP, s1) | 73.00 (1.00) | 89.48 (1.00) | 46.80 (1.01) | 37.50 (0.94) | 2.46 (1.66) | 23.98 (1.97) | 28.46 (0.99) | 69.75 (1.01) | 66.40 (1.02) | 28.83 (1.05) | 1.01 | 7639 (0.23) |
| NAMM (MLP, s2) | 74.00 (1.01) | 88.64 (0.99) | 46.04 (0.99) | 41.50 (1.04) | 1.53 (1.03) | 25.94 (2.13) | 29.78 (1.03) | 69.80 (1.01) | 65.23 (1.00) | 29.22 (1.07) | 1.02 | 8475 (0.26) |
| NAMM (BAM, s1) | 73.00 (1.00) | 89.81 (1.00) | 46.70 (1.00) | 38.75 (0.97) | 2.19 (1.48) | 25.14 (2.06) | 28.51 (0.99) | 69.50 (1.01) | 66.51 (1.02) | 28.91 (1.05) | 1.00 | 7951 (0.24) |
| NAMM (BAM, s2) | 73.00 (1.00) | 90.03 (1.01) | 46.85 (1.01) | 42.00 (1.05) | 2.35 (1.59) | 24.69 (2.03) | 28.46 (0.99) | 69.65 (1.01) | 66.57 (1.02) | 29.25 (1.07) | 1.04 | 8267 (0.25) |
| NAMM (BAM, s3) | 73.00 (1.00) | 89.81 (1.00) | 46.35 (1.00) | 40.00 (1.00) | 3.04 (2.05) | 27.55 (2.26) | 28.60 (0.99) | 69.53 (1.01) | 66.35 (1.02) | 29.33 (1.11) | 1.07 | 8155 (0.25) |

Table 13: NAMMs evaluation on InfiniteBench (Zhang et al., 2024a). The normalized overall performance (in brackets) is calculated using the average performance of the base model with full cache.

| Model/Task name | Retrieval | | | Dialogue | Novel | | | | Math | Code | | Overall | |
|---|---|---|---|---|---|---|---|---|---|---|---|---|---|
| | Ret.PassKey | Ret.Number | Ret.KV | En.Dia | En.Sum | En.MC | En.QA | ZH.QA | Math.Find | Code.Run | Code.Debug | All tasks | Cache size |
| Base model | 0.00 | 0.00 | 0.00 | 1.00 | 7.73 | 0.00 | 1.05 | 1.79 | 0.00 | 0.00 | 0.00 | 1.05 (1.00) | 32747 (1.00) |
| NAMM (MLP, s1) | 0.00 | 10.00 | 0.00 | 3.00 | 7.27 | 3.93 | 1.57 | 4.26 | 0.57 | 0.00 | 3.30 | 3.08 (2.93) | 11329 (0.35) |
| NAMM (MLP, s2) | 10.17 | 11.86 | 0.00 | 2.50 | 7.48 | 3.06 | 1.58 | 4.10 | 1.71 | 0.00 | 1.52 | 4.00 (3.80) | 13031 (0.40) |
| NAMM (BAM, s1) | 9.49 | 9.83 | 1.80 | 0.50 | 14.36 | 37.12 | 8.95 | 16.20 | 5.71 | 1.50 | 6.09 | 10.14 (9.63) | 11173 (0.34) |
| NAMM (BAM, s2) | 11.86 | 11.86 | 1.80 | 1.00 | 14.62 | 35.37 | 8.96 | 15.45 | 0.57 | 1.75 | 4.31 | 9.78 (9.29) | 12789 (0.39) |
| NAMM (BAM, s3) | 11.86 | 11.86 | 1.80 | 1.00 | 14.91 | 36.24 | 8.78 | 17.67 | 10.57 | 1.75 | 4.57 | 11.00 (10.45) | 13192 (0.40) |

## C ADDITIONAL RESULTS

### C.1 PERFORMANCE ACROSS INCREMENTAL STAGES AND ARCHITECTURES

We provide additional results and analysis to the summarized one, complementing Section 4, with the detailed performance across different NAMMs, evaluating the best checkpoints after each stage of incremental training stage, and ablating the BAM architecture with an MLP.

**Extended language modeling results.** We report our results for LongBench, InfiniteBench, and ChouBun in Tables 12, 13, 14. First, we note that even training on a single task with our simple MLP architecture impressively improves performance across all benchmarks. Additionally, performance across benchmarks sees near-monotonic further improvements with each stage of our incremental evolution recipe. Comparing our implementations, we note that the performance benefits from the memory models with backward attention are consistently superior to the fully connected variant in both initial stages of incremental training, empirically validating our hypothesis about the importance of global KV cache information for determining the importance of each token. Lastly, on ChouBun. we observe that the performance with BAM sees a notable upswing after the second stage of incremental training, which might be associated with the introduction of another ideogram-based language in the training set.[11] The same improvement not occurring with the MLP-based NAMMs might be further evidence of architectural performance saturation, highlighting once again the effectiveness of our main implementation design.

**Extended zero-shot transfer results.** We report our extended zero-shot transfer results for the 70B model and the offline RL setting in Tables 15, 16, and 17. We see the benefits from NAMMs again increase as we incorporate backward attention, and with each stage of incremental training to a similar extent as with the language modeling tasks. These results further highlight the potential benefits of scaling up the architecture of our memory model and increasing the number of incremental stages.

---

[11]The DuReader task, used in the second stage of incremental training, uses the Chinese language.

Table 14: NAMMs evaluation on the new ChouBun benchmark. The normalized performance (in brackets) is calculated using the base model with full cache.

| Model/Task name | Extractive QA | | | Summarization | Overall | |
|---|---|---|---|---|---|---|
| | *JA.WikiQA* | *JA.EdinetQA* | *JA.CorpSecQA* | *JA.CorpSecSum* | All tasks | Cache size |
| Base model | **22.91** (1.00) | 28.34 (1.00) | 11.83 (1.00) | 21.75 (1.00) | 21.21 (1.00) | 12099 (1.00) |
| NAMM (MLP, s1) | 21.60 (0.94) | 26.81 (0.95) | 10.34 (0.87) | 29.60 (1.36) | 22.09 (1.04) | 9525 (0.79) |
| NAMM (MLP, s2) | 20.76 (0.91) | 26.30 (0.93) | 11.86 (1.00) | 29.32 (1.35) | 22.06 (1.04) | 9815 (0.81) |
| NAMM (BAM, s1) | 19.19 (0.84) | **28.85** (1.02) | 14.36 (1.21) | 28.51 (1.31) | 22.73 (1.07) | 9569 (0.79) |
| NAMM (BAM, s2) | 20.75 (0.91) | 28.46 (1.00) | 14.55 (1.23) | 32.45 (1.49) | 24.05 (1.13) | 9867 (0.82) |
| NAMM (BAM, s3) | 21.34 (0.93) | 28.61 (1.01) | **14.64** (1.24) | **33.15** (1.52) | **24.44** (1.15) | 9895 (0.82) |

Table 15: NAMMs evaluation on LongBench (Bai et al., 2023) with a Llama 3 70B model. The normalized performance (in brackets) is calculated using the base model with full cache. The aggregate test task performance of NAMMs models is taken by averaging the normalized scores on the tasks not used for incremental evolution. The tasks on which NAMMs are trained are highlighted with a gray background.

| Model/Task id | Single-Doc QA | | | | Multi-Doc QA | | | | Summarization | | | |
|---|---|---|---|---|---|---|---|---|---|---|---|---|
| | 1-1 | 1-2 | 1-3 | 1-4 | 2-1 | 2-2 | 2-3 | 2-4 | 3-1 | 3-2 | 3-3 | 3-4 |
| Base model | **9.38** (1.00) | 13.84 (1.00) | **24.99** (1.00) | 17.78 (1.00) | **11.73** (1.00) | **14.26** (1.00) | 8.11 (1.00) | **26.43** (1.00) | 13.13 (1.00) | **24.55** (1.00) | 23.20 (1.00) | **10.08** (1.00) |
| NAMM (MLP, s1) | 6.94 (0.74) | 13.82 (1.00) | 24.27 (0.97) | 17.60 (0.99) | 10.83 (0.92) | 14.17 (0.99) | 7.89 (0.97) | 20.81 (0.79) | 13.09 (1.00) | 23.30 (0.95) | 23.28 (1.00) | 8.66 (0.86) |
| NAMM (MLP, s2) | 7.88 (0.84) | 13.71 (0.99) | 23.27 (0.93) | 18.18 (1.02) | 11.41 (0.97) | 14.11 (0.99) | 8.07 (0.99) | 21.75 (0.82) | **14.28** (1.09) | 24.48 (1.00) | 22.00 (0.95) | 8.99 (0.89) |
| NAMM (BAM, s1) | 7.31 (0.78) | 13.75 (0.99) | 24.51 (0.98) | 17.78 (1.00) | 10.82 (0.92) | 14.08 (0.99) | 7.59 (0.94) | 19.27 (0.73) | 13.89 (1.06) | 23.71 (0.97) | 23.41 (1.01) | 8.87 (0.88) |
| NAMM (BAM, s2) | 3.57 (0.38) | **13.86** (1.00) | 23.02 (0.92) | **18.71** (1.05) | 4.94 (0.42) | 13.32 (0.93) | 1.90 (0.23) | 17.74 (0.67) | 10.39 (0.79) | 20.45 (0.83) | 23.18 (1.00) | 8.13 (0.81) |
| NAMM (BAM, s3) | 9.13 (0.97) | 13.53 (0.98) | 24.25 (0.97) | 17.82 (1.00) | 11.45 (0.98) | 13.76 (0.96) | **8.34** (1.03) | 21.79 (0.82) | 12.66 (0.96) | 24.21 (0.99) | **23.56** (1.02) | 8.62 (0.86) |

| Model/Task id | Few-shot Learning | | | | Synthetic | | | Code | | Overall | | |
|---|---|---|---|---|---|---|---|---|---|---|---|---|
| | 4-1 | 4-2 | 4-3 | 4-4 | 5-1 | 5-2 | 5-3 | 6-1 | 6-2 | All tasks | Test tasks | Cache size |
| Base model | 78.00 (1.00) | 92.43 (1.00) | **48.67** (1.00) | **45.50** (1.00) | 22.50 (1.00) | **75.37** (1.00) | 33.89 (1.00) | 74.60 (1.00) | 71.19 (1.00) | **35.22** (1.00) | N/A | 10107 (1.00) |
| NAMM (MLP, s1) | 78.00 (1.00) | 92.28 (1.00) | 48.37 (0.99) | 43.50 (0.96) | 20.76 (0.92) | 68.66 (0.91) | 33.89 (1.00) | 74.58 (1.00) | 71.68 (1.01) | 34.11 (0.97) | **0.99** | 7930 (0.78) |
| NAMM (MLP, s2) | 77.00 (0.99) | 91.93 (0.99) | 48.60 (1.00) | 44.75 (0.98) | 17.17 (0.76) | 70.21 (0.93) | **36.18** (1.07) | **74.72** (1.00) | 71.30 (1.00) | 34.29 (0.97) | **0.99** | 8445 (0.84) |
| NAMM (BAM, s1) | 77.50 (0.99) | **92.46** (1.00) | 48.24 (0.99) | 45.00 (0.99) | 17.32 (0.77) | 69.87 (0.93) | 33.89 (1.00) | 74.58 (1.00) | 72.40 (1.02) | 34.11 (0.97) | **0.99** | 7947 (0.79) |
| NAMM (BAM, s2) | 74.50 (0.96) | 51.45 (0.56) | 39.73 (0.82) | 15.00 (0.33) | 5.86 (0.26) | 13.35 (0.18) | 34.29 (1.01) | 73.81 (0.99) | 61.91 (0.87) | 25.20 (0.72) | 0.79 | 8276 (0.82) |
| NAMM (BAM, s3) | **78.50** (1.01) | 92.36 (1.00) | 48.49 (1.00) | **45.50** (1.00) | 19.07 (0.85) | 74.19 (0.98) | 34.28 (1.01) | 74.71 (1.00) | **72.42** (1.02) | 34.70 (0.99) | **0.99** | 8365 (0.83) |

To this end, given the generality of our parameterization, an interesting unexplored approach could be to incorporate different base models and input modalities during evolutionary training, something that would substantially increase problem diversity to obtain an even more robust transfer behavior.

## C.2 TRAINING CURVES WITH FULLY-CONNECTED NAMMS

In Figure 11, we provide training curves of our Neural Attention Memory Model using a simple MLP architecture rather than backward attention, evaluated in Section 4. In the left sub-plot, we show the average and standard deviation of the normalized batch performance across the population, while in the right sub-plot, we show the normalized per-task and average performance on all samples of the optimized mean from CMA-ES. When compared with the BAM training curve from Figure 4, we note a few interesting differences, although its evaluation performance on the full Long-Bench benchmark is lower across both incremental phases (see Table 2), both its population batch performance and the CMA-ES full-task performance on the training sets are either comparable or slightly higher than BAM's. This dichotomy appears to indicate that cross-token interactions might provide a better inductive bias, mitigating the overfitting potential of NAMMs.

## C.3 EVOLUTION OF MEMORY SIZE DURING TRAINING

In Figure 12, we provide training curves for the evolution of the memory size collected at the end of each task prompt of our NAMMs. On the left and right subplots, we provide results for the BAM and MLP implementations, respectively. For both architectures, we find that the memory size generally increases with training. This result suggests that NAMMs might learn to recognize additional valuable tokens as training progresses, enabling the corresponding performance improvements on the training tasks. Hence, they might indicate that there is some degree of a trade-off between the efficiency and performance of NAMMs. However, we note that both models are trained only for

Table 16: Evaluation on the LongVideoBench and MLVU benchmarks with Llava Next Video 7B. The normalized performance (in brackets) is calculated using the base model with full cache.

| Model/Task name | LongVideoBench | MLVU | All tasks | Cache size |
|---|---|---|---|---|
| Base model | 43.45 (1.00) | **44.23** (1.00) | 43.84 (1.00) | 7039 (1.00) |
| NAMM (MLP, s1) | 39.64 (0.91) | 41.24 (0.93) | 40.44 (0.92) | 584 (0.08) |
| NAMM (MLP, s2) | 41.06 (0.94) | 39.72 (0.90) | 40.39 (0.92) | 713 (0.10) |
| NAMM (BAM, s1) | 41.06 (0.94) | 41.98 (0.95) | 41.52 (0.95) | 723 (0.10) |
| NAMM (BAM, s2) | **45.03** (1.04) | **44.23** (1.00) | **44.63** (1.02) | 4948 (0.70) |
| NAMM (BAM, s3) | 44.58 (1.03) | 44.18 (1.00) | 44.38 (1.01) | 5100 (0.72) |

Table 17: NAMMs evaluation on D4RL (Fu et al., 2020) using a Decision Transformer model (Chen et al., 2021b; Beeching & Simonini, 2022). The normalized overall performance (in brackets) is calculated using the average performance of the base model with full cache.

| Model/Task name | Hopper-v3 | | | Walker2d-v3 | | | HalfCheetah-v3 | | | Overall | |
|---|---|---|---|---|---|---|---|---|---|---|---|
| | Medium | Med-Replay | Expert | Medium | Med-Replay | Expert | Medium | Med-Replay | Expert | All tasks | Cache size |
| Base model | 33.36 (1.00) | 18.37 (1.00) | 44.62 (1.00) | 68.21 (1.00) | 7.18 (1.00) | 38.98 (1.00) | 34.91 (1.00) | 5.06 (1.00) | 10.64 (1.00) | 29.04 (1.00) | 3000 (1.00) |
| NAMM (MLP, s1) | 33.01 (0.99) | 18.39 (1.00) | 38.09 (0.85) | 70.82 (1.04) | 7.25 (1.01) | 44.61 (1.14) | 35.64 (1.02) | 5.05 (1.00) | 10.87 (1.02) | 29.30 (1.01) | 1993 (0.66) |
| NAMM (MLP, s2) | 33.48 (1.00) | **19.24** (1.05) | 30.07 (0.67) | **73.22** (1.07) | **7.95** (1.11) | 48.21 (1.24) | 33.59 (0.96) | 5.81 (1.15) | **14.67** (1.38) | 29.58 (1.02) | 2834 (0.94) |
| NAMM (BAM, s1) | 35.02 (1.05) | 18.24 (0.99) | 45.95 (1.03) | 69.33 (1.02) | 7.91 (1.10) | 44.45 (1.14) | 34.25 (0.98) | 5.12 (1.01) | 13.68 (1.29) | 30.44 (1.05) | 2009 (0.67) |
| NAMM (BAM, s2) | 35.18 (1.05) | 18.79 (1.02) | 48.08 (1.08) | 71.97 (1.06) | 7.70 (1.07) | 49.74 (1.28) | **35.67** (1.02) | 5.78 (1.14) | 10.82 (1.02) | 31.53 (1.09) | 2534 (0.84) |
| NAMM (BAM, s3) | **36.10** (1.08) | 18.86 (1.03) | **49.39** (1.11) | 70.87 (1.04) | 7.53 (1.05) | **50.02** (1.28) | 34.56 (0.99) | **5.90** (1.17) | 12.34 (1.16) | **31.73** (1.09) | 2434 (0.81) |

performance maximization, without any incentive to be more conservative. To this end, exploring regularization strategies to make NAMMs aware of deployment costs is an interesting direction for future work to obtain tailored sweet spots to cater to instance-specific resource constraints.

## C.4 INCREMENTAL TRAINING ABLATION

We provide a full set of ablations results for our incremental training strategy, training a Neural Attention Memory Model with the BAM architecture from scratch on both the PassageRetrieval-en and DuReader tasks, as employed during the second stage of incremental learning. We evolve this Neural Attention Memory Model for 360 consecutive generations and provide training curves in Figure 13. In the left sub-plot, we show the average and standard deviation of the normalized batch performance across the population, in the center sub-plot, we show the normalized per-task and average performance on all samples of the optimized mean from CMA-ES, and on the right subplot we show the corresponding memory size. Furthermore, in Table 18, we provide the full LongBench evaluation results for this baseline, also showing our original incremental model's performance for ease of comparison. Interestingly, the non-incremental NAMM obtained a notably higher score on the training tasks with a normalized performance of 1.57, in contrast to the normalized performance of 1.41 achieved by the best checkpoint from the second incremental training stage. Yet, outside the PassageRetrieval-en and DuReader tasks, its performance is notably inferior and very close to the original performance of the base model. These results appear to indicate that the usefulness of incremental training goes beyond the faster evolution provided by reducing the number of evaluation prompts to assess performance and that this strategy plays an important role in regularizing evolution and making Neural Attention Memory Models effectively generalize to new tasks.

## C.5 RUNNING TIMES AND MEMORY SAVINGS

We provide details about the efficiency and costs of NAMMs on top of the Llama 3 8B base model used for training. For our main experimental setup, we used rented cloud instances with Nvidia H100 GPUs, Intel Xeon Platinum 8481C CPUs, and 1932GB of RAM. We performed model inference for each prompt on a single GPU, with batch size 1. During training, we used a single node with 8 GPUs, distributing the evaluation of our population across 8 processes. However, we like to remark that since training NAMMs does not require any gradient computation, we were not restricted by any kind of hardware during training. In this regard, using inference-specialized resources beyond GPUs might provide considerable speedups and lower costs to ones employed in this work.

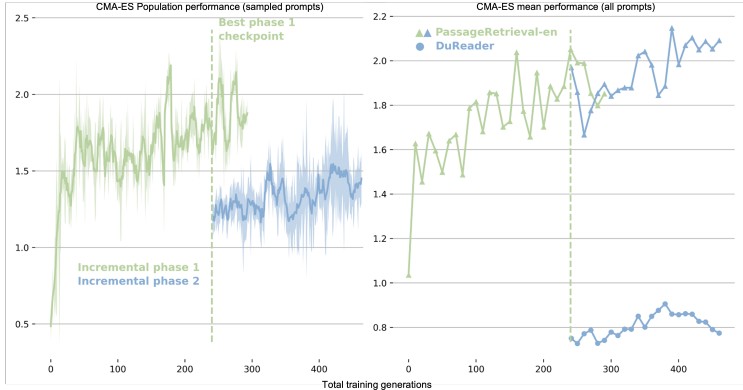

Figure 11: Mean and standard deviation over the CMA-ES population batch performance (left), together with the performance of the learned mean parameter on each task (right) for the training of the MLP NAMM.

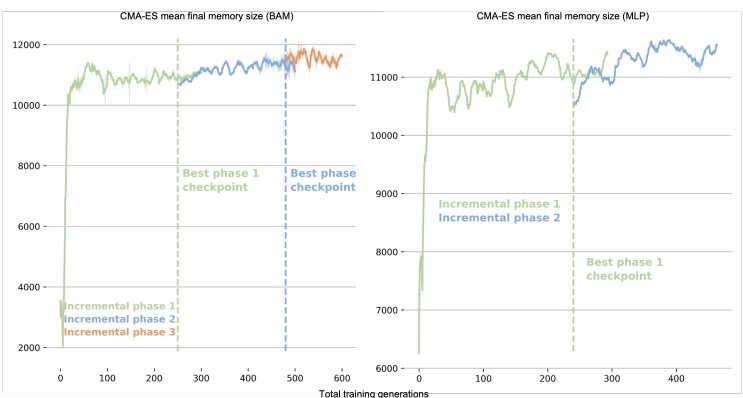

Figure 12: Final memory size of NAMM parameterized by the learned mean of CMA-ES for both the BAM (left) and the MLP implementations (right).

**Training.** We collected the training time for each generation of NAMMs. As detailed in Section 3 and Appendix A, with the employed hyper-parameters, each generation consisted of running the base model for population size × task samples $= 32 \times 64 = 2048$ prompts for each task. Thus, each incremental phase got

Table 20: Training times per generation of our final NAMMs using our available distributed hardware.

| Incremental phase | Time per generation (s) |
|---|---|
| Phase 1 (+PassageRetrieval-en) | 1597.62 |
| Phase 2 (+DuReader) | 4027.33 |
| Phase 3 (+NarrativeQA) | 9848.31 |

linearly more expensive, with up to $2048 \times 3 = 6144$ NAMM evaluation in the final phase. These prompts were distributed across our 8 processes balancing the number of tokens evaluated in each. We also note that the average prompt length and the nature of each task (e.g., exact match, summarization, etc.) varied quite significantly in LongBench, making their evaluation costs non-uniform.

**Inference.** We collected running times of NAMMs and our baselines in different settings. In particular, these include both: 1. Using samples from the full LongBench benchmark with an average length of 12099 2. Using only samples from LongBench selected to exceed the base transformer maximum length with an average length of 32641. Finally, we also record the running time of an ablated version of our NAMM run on top of the base transformer that does not modify its KV cache, in order to disentangle the gains from the reduced memory and analyze the pure overheads from our model's execution.

As shown in Table 19, the running time overhead of our NAMM ablation that does not evict tokens is small when compared to the base model. Instead, the running time of NAMMs and the baselines

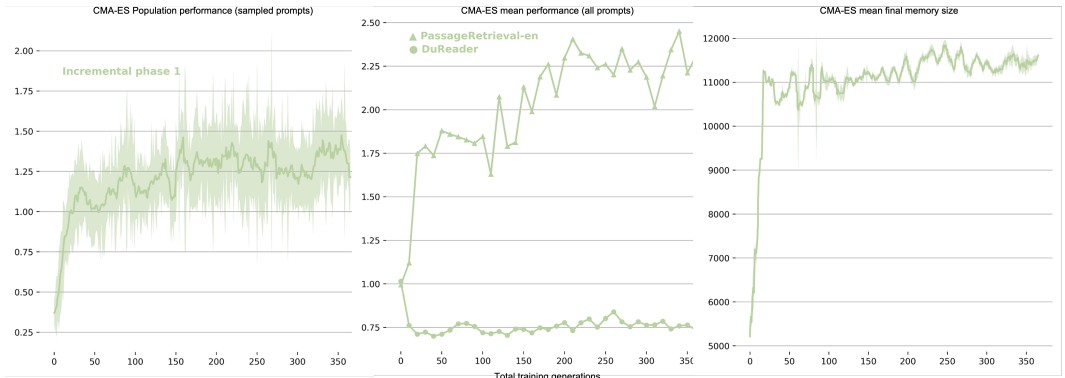

Figure 13: Mean and standard deviation over the CMA-ES population batch performance (left), together with the performance of the learned mean parameter on each task (center) and its final memory size for the NAMM trained without incremental evolution.

Table 18: NAMMs incremental learning (IL) ablation evaluation on LongBench (Bai et al., 2023). The *No IL* baseline is trained from scratch on both the PassageRetrieval-en and DuReader tasks, the same employed during the second stage of incremental learning.

| Model/Task id | Single-Doc QA | | | | Multi-Doc QA | | | | Summarization | | | |
|---|---|---|---|---|---|---|---|---|---|---|---|---|
| | 1-1 | 1-2 | 1-3 | 1-4 | 2-1 | 2-2 | 2-3 | 2-4 | 3-1 | 3-2 | 3-3 | 3-4 |
| Base model | 10.38 (1.00) | 12.79 (1.00) | 22.60 (1.00) | 21.31 (1.00) | 10.41 (1.00) | 12.67 (1.00) | 7.54 (1.00) | 25.86 (1.00) | 29.34 (1.00) | 23.93 (1.00) | 0.92 (1.00) | 2.66 (1.00) |
| NAMM (BAM, s1) | 5.77 (0.56) | 12.76 (1.00) | 22.94 (1.02) | 21.55 (1.01) | 9.47 (0.91) | 12.21 (0.96) | 6.51 (0.86) | 18.73 (0.72) | 28.06 (0.96) | 23.97 (1.00) | 1.01 (1.10) | 4.00 (1.50) |
| NAMM (BAM, s2) | 7.08 (0.68) | 12.70 (0.99) | 22.21 (0.98) | 21.50 (1.01) | 9.94 (0.95) | 12.21 (0.96) | 7.13 (0.95) | 20.34 (0.79) | 28.87 (0.98) | 23.84 (1.00) | 0.92 (1.00) | 3.94 (1.48) |
| NAMM (BAM, s3) | 9.14 (0.88) | 12.63 (0.99) | 21.94 (0.97) | 21.34 (1.00) | 9.71 (0.93) | 11.63 (0.92) | 6.98 (0.93) | 20.58 (0.80) | 28.78 (0.98) | 24.39 (1.02) | 1.04 (1.13) | 3.63 (1.36) |
| NAMM (BAM, no IL) | 6.46 (0.62) | 12.72 (0.99) | 22.87 (1.01) | 21.22 (1.00) | 9.91 (0.95) | 11.77 (0.93) | 5.61 (0.74) | 18.94 (0.73) | 27.63 (0.94) | 22.60 (0.94) | 0.91 (0.99) | 1.75 (0.66) |

| Model/Task id | Few-shot Learning | | | | Synthetic | | | Code | | Overall | | |
|---|---|---|---|---|---|---|---|---|---|---|---|---|
| | 4-1 | 4-2 | 4-3 | 4-4 | 5-1 | 5-2 | 5-3 | 6-1 | 6-2 | All tasks | Test tasks | Cache size |
| Base model | 73.00 (1.00) | 89.45 (1.00) | 46.54 (1.00) | 40.00 (1.00) | 1.48 (1.00) | 12.18 (1.00) | 28.80 (1.00) | 69.09 (1.00) | 65.17 (1.00) | 28.86 (1.00) | N/A | 10107 |
| NAMM (BAM, s1) | 73.00 (1.00) | 89.81 (1.00) | 46.70 (1.00) | 38.75 (0.97) | 2.19 (1.48) | 25.14 (2.06) | 28.51 (0.99) | 69.50 (1.01) | 66.51 (1.02) | 28.91 (1.05) | 1.00 | 8205 |
| NAMM (BAM, s2) | 73.00 (1.00) | 90.03 (1.01) | 46.85 (1.01) | 42.00 (1.05) | 2.35 (1.59) | 24.69 (2.03) | 28.46 (0.99) | 69.65 (1.01) | 66.57 (1.02) | 29.25 (1.07) | 1.04 | 8521 |
| NAMM (BAM, s3) | 73.00 (1.00) | 89.81 (1.00) | 46.35 (1.00) | 40.00 (1.00) | 3.04 (2.05) | 27.55 (2.26) | 28.60 (0.99) | 69.53 (1.01) | 66.35 (1.02) | 29.33 (1.11) | 1.07 | 8409 |
| NAMM (BAM, no IL) | 73.00 (1.00) | 89.28 (1.00) | 46.43 (1.00) | 38.75 (0.97) | 2.49 (1.68) | 29.28 (2.40) | 28.46 (0.99) | 69.80 (1.01) | 64.77 (0.99) | 28.79 (1.03) | 0.98 | 8457 |

while evicting tokens is always inferior to the base model in all settings, and scales positively with longer prompts.

**Memory.** Furthermore, in Table 21, we also reported estimated effects in peak GPU memory consumption, which were calculated from the peak KV cache sizes, together with the sizes of additional information (e.g., attention matrix) and models used by each method (again recorded on

Table 21: Calculated peak storage cost and savings (MB) of NAMMs, together with the H2O and L2 baselines with default hyperparameters.

| Model | KV cache storage | Additional storage overheads | Total savings |
|---|---|---|---|
| Base model (full cache) | 65282 | 0 | 0 |
| H2O | 17408 | 5659 | 42215 |
| L2 | 17408 | 0 | 47874 |
| NAMM | 27236 | 5668 | 32378 |

LongBench). We would like to note, however, that as the main objective of our work was to provide performance benefits we did not particularly optimize our code for memory efficiency or speed. Thus, actual empirical savings with our shared implementation might differ from these calculated estimates. For instance, both our NAMMs and H2O baseline do not employ specialized kernels to replace FlashAttention (Dao et al., 2022).

## C.6 MISTRAL BASE MODEL AND FINETUNING NAMMS

We also analyzed an additional 0-shot transfer setting, this time applying NAMMs on top of the Mistral 7B base model (Jiang et al., 2023). We considered 2 different setups: 1. Zero-shot application, taking our best NAMM model trained with the Llama 8B context-extended model. 2. Post

Table 19: Running times from using NAMMs on top of a Llama 3 8B base model, together with the running time of the H2O and L2 baselines. NAMMs (full cache) indicates our NAMMs used without pruning the KV cache memory in order to show its total overhead without the gains from reducing the KV cache size.

| Longbench (12099 average tokens per sample) | |
|---|---|
| **Method** | **Time per task sample** |
| Base model (full cache) | 4.0 (1.00) |
| NAMM (full cache) | 4.54 (1.13) |
| H2O | 3.78 (0.94) |
| L2 | 3.74 (0.93) |
| NAMM (Ours) | 3.97 (0.99) |
| **Selected max length samples (32641 average tokens per sample)** | |
| **Method** | **Time per task sample** |
| Base model (full cache) | 15.52 (1.00) |
| NAMM (full cache) | 17.6 (1.13) |
| H2O | 11.79 (0.76) |
| L2 | 10.5 (0.68) |
| NAMM (Ours) | 13.61 (0.88) |

Table 22: NAMMs evaluation on LongBench using Mistral 7B v0.3 as base model. The normalized performance (in brackets) is calculated using the base model with full cache. The tasks used for NAMM's training are highlighted in gray.

| Model/Task id | Single-Doc QA | | | | Multi-Doc QA | | | | Summarization | | | |
|---|---|---|---|---|---|---|---|---|---|---|---|---|
| | 1-1 | 1-2 | 1-3 | 1-4 | 2-1 | 2-2 | 2-3 | 2-4 | 3-1 | 3-2 | 3-3 | 3-4 |
| Base model | 15.85 (1.00) | 8.25 (1.00) | 28.45 (1.00) | 16.85 (1.00) | 12.85 (1.00) | 11.80 (1.00) | 8.28 (1.00) | 11.31 (1.00) | 28.62 (1.00) | 22.48 (1.00) | 25.47 (1.00) | 12.41 (1.00) |
| H2O | 7.96 (0.50) | 7.67 (0.93) | 28.98 (1.02) | 16.47 (0.98) | 11.83 (0.92) | 11.06 (0.94) | 7.94 (0.96) | 19.19 (1.70) | 25.54 (0.89) | 20.55 (0.91) | 25.38 (1.00) | 12.19 (0.98) |
| L2 | 9.77 (0.62) | 8.51 (1.03) | **30.20** (1.06) | **17.72** (1.05) | **32.46** (2.53) | **18.00** (1.53) | **15.77** (1.91) | 16.23 (1.43) | 11.68 (0.41) | 19.49 (0.87) | 24.57 (0.96) | 6.17 (0.50) |
| NAMM (0-shot) | 13.01 (0.82) | 10.30 (1.25) | **28.98** (1.02) | 17.10 (1.01) | 11.34 (0.88) | **12.24** (1.04) | 7.36 (0.89) | 14.19 (1.25) | 25.22 (0.88) | 20.23 (0.90) | **27.79** (1.09) | 11.97 (0.96) |
| NAMM (Finetune) | 9.82 (0.62) | 10.36 (1.26) | 28.40 (1.00) | 17.38 (1.03) | 11.72 (0.91) | 12.11 (1.03) | 7.28 (0.88) | 19.04 (1.68) | 25.77 (0.90) | 21.74 (0.97) | 27.72 (1.09) | **12.52** (1.01) |

| Model/Task id | Few-shot Learning | | | | Synthetic | | | Code | | Overall | | |
|---|---|---|---|---|---|---|---|---|---|---|---|---|
| | 4-1 | 4-2 | 4-3 | 4-4 | 5-1 | 5-2 | 5-3 | 6-1 | 6-2 | All tasks | Test tasks | Cache size |
| Base model | 76.50 (1.00) | 90.50 (1.00) | 36.55 (1.00) | 41.50 (1.00) | 1.00 (1.00) | **32.17** (1.00) | 27.50 (1.00) | 65.91 (1.00) | 62.71 (1.00) | 30.33 (1.00) | N/A | 10107 (1.00) |
| H2O | 76.00 (0.99) | 90.14 (1.00) | 38.54 (1.05) | 31.00 (0.75) | 1.50 (1.50) | 6.88 (0.21) | 24.13 (0.88) | 66.25 (1.01) | 64.46 (1.03) | 28.27 (0.96) | N/A | 6662 (0.66) |
| L2 | 72.50 (0.95) | 75.65 (0.84) | 33.91 (0.93) | 13.50 (0.33) | 0.50 (0.50) | 6.00 (0.19) | 24.50 (0.89) | 64.46 (0.98) | 50.11 (0.80) | 26.27 (0.97) | N/A | 6662 (0.66) |
| NAMM (0-shot) | 75.50 (0.99) | 90.83 (1.00) | 40.84 (1.12) | 36.25 (0.87) | 1.73 (1.73) | 26.04 (0.81) | 27.50 (1.00) | 69.45 (1.05) | **64.60** (1.03) | 30.12 (1.03) | 1.04 | 8518 (0.84) |
| NAMM (Finetune) | **76.50** (1.00) | **91.25** (1.01) | 40.63 (1.11) | **42.75** (1.03) | 2.18 (2.18) | 22.54 (0.70) | 28.00 (1.02) | 69.23 (1.05) | 63.96 (1.02) | **30.52** (1.07) | **1.08** | 8654 (0.86) |

cross-model fine-tuning, running a small amount of additional evolutionary optimization comprising 20 generations using CMA-ES and the same 3 training tasks used for Llama. We provide our full results and analysis in Table 22.

Analogously to our other zero-shot transfer results provided in Section 4, we find that NAMMs yield considerable benefits also when transferred to the Mistral model, overcoming the efficiency-performance tradeoff of hand-designed baselines. Furthermore, this analysis also shows that performance could be further improved by a few finetuning generations after transferring to a different base models. While we did not investigate finetuning with our other transformers (e.g., Llama 70B), we believe these results highlight the potential of cheaply improving NAMMs ' already-remarkable zero-shot benefits, which we hope will be further explored in future work.

## C.7 ATTENTION SPECTROGRAM FEATURES ABLATION STUDY

We examined ablating the attention spectrogram features produced by the STFT procedure and re-training our NAMMs with two different alternatives:

Table 23: NAMMs evaluation on LongBench ablating the STFT features. The normalized performance (in brackets) is calculated using the base model with full cache. The tasks used for NAMM's training are highlighted in gray.

| Model/Task id | Single-Doc QA | | | | Multi-Doc QA | | | | Summarization | | | |
|---|---|---|---|---|---|---|---|---|---|---|---|---|
| | 1-1 | 1-2 | 1-3 | 1-4 | 2-1 | 2-2 | 2-3 | 2-4 | 3-1 | 3-2 | 3-3 | 3-4 |
| Base model | **10.38** (1.00) | 12.79 (1.00) | **22.60** (1.00) | 21.31 (1.00) | **10.41** (1.00) | **12.67** (1.00) | **7.54** (1.00) | **25.86** (1.00) | **29.34** (1.00) | 23.93 (1.00) | 0.92 (1.00) | 2.66 (1.00) |
| NAMM (BAM, s2) | 7.08 (0.68) | 12.70 (0.99) | 22.21 (0.98) | 21.50 (1.01) | 9.94 (0.95) | 12.21 (0.96) | 7.13 (0.95) | 20.34 (0.79) | 28.87 (0.98) | 23.84 (1.00) | 0.92 (1.00) | **3.94** (1.48) |
| NAMM (BAM, s3) | 9.14 (0.88) | 12.63 (0.99) | 21.94 (0.97) | 21.34 (1.00) | 9.71 (0.93) | 11.63 (0.92) | 6.98 (0.93) | 20.58 (0.80) | 28.78 (0.98) | 24.39 (1.02) | **1.04** (1.13) | 3.63 (1.36) |
| NAMM (RAD features) | 10.09 (0.97) | **12.93** (1.01) | 21.35 (0.94) | **21.56** (1.01) | 9.65 (0.93) | 12.28 (0.97) | 5.26 (0.70) | 18.09 (0.70) | 28.52 (0.97) | **24.49** (1.02) | 0.88 (0.96) | 3.43 (1.29) |
| NAMM (Raw attention) | 9.98 (0.96) | 12.77 (1.00) | 22.04 (0.98) | 21.48 (1.01) | 9.72 (0.93) | 12.08 (0.95) | 6.31 (0.84) | 22.61 (0.87) | 28.68 (0.98) | 23.76 (0.99) | 0.87 (0.95) | 3.75 (1.41) |

| Model/Task id | Few-shot Learning | | | | Synthetic | | | Code | | Overall | | |
|---|---|---|---|---|---|---|---|---|---|---|---|---|
| | 4-1 | 4-2 | 4-3 | 4-4 | 5-1 | 5-2 | 5-3 | 6-1 | 6-2 | All tasks | Test tasks | Cache size |
| Base model | **73.00** (1.00) | 89.45 (1.00) | 46.54 (1.00) | 40.00 (1.00) | 1.48 (1.00) | 12.18 (1.00) | **28.80** (1.00) | 69.09 (1.00) | 65.17 (1.00) | 28.86 (1.00) | N/A | 10107 (1.00) |
| NAMM (BAM, s2) | **73.00** (1.00) | **90.03** (1.01) | **46.85** (1.01) | **42.00** (1.05) | 2.35 (1.59) | 24.69 (2.03) | 28.46 (0.99) | 69.65 (1.01) | 66.57 (1.02) | 29.25 (1.07) | 1.04 | 8521 (0.84) |
| NAMM (BAM, s3) | **73.00** (1.00) | 89.81 (1.00) | 46.35 (1.00) | 40.00 (1.00) | **3.04** (2.05) | **27.55** (2.26) | 28.60 (0.99) | 69.53 (1.01) | 66.35 (1.02) | **29.33** (1.11) | 1.07 | 8409 (0.83) |
| NAMM (RAD features) | **73.00** (1.00) | 89.48 (1.00) | 46.64 (1.00) | 38.50 (0.96) | 1.93 (1.30) | 21.15 (1.74) | 27.76 (0.96) | 69.68 (1.01) | 66.31 (1.02) | 28.71 (1.02) | 1.00 | 8000 (0.79) |
| NAMM (Raw attention) | **73.00** (1.00) | 89.48 (1.00) | 46.66 (1.00) | 39.50 (0.99) | 1.48 (1.00) | 21.86 (1.79) | 28.46 (0.99) | **69.73** (1.01) | **66.77** (1.02) | 29.10 (1.03) | 0.95 | 8523 (0.84) |

1. The *naive* approach of using the raw attention values directly (cropped to a fixed length) as input to NAMMs.

2. Substituting the STFT features by constructing a 'handcrafted' feature representation that simply includes three values: i. The sum of the attention values of each token. ii) The recency of each token. iii. The diversity of each token (computed by concatenating the keys and values to represent each token and averaging the L2 distance to all other tokens). We refer to this baseline as *RAD*.

We trained these baselines only for two incremental phases on the PassageRetrieval-en and Dureader tasks (thus, we also compared them with our original NAMM model after phase 2). Please refer to Table 23 for our results. Overall, we find our baselines yield quite different behaviors, both underperforming our original NAMM design.

First, we find our naive baseline, taking as input the cropped attention value, is not able to improve over the full cache model when evaluated on the whole of LongBench. However, we note that its performance on the training task is significantly beyond the base model. Thus, we find this is strongly suggestive of the occurrence of overfitting, which we believe is to be expected as our memory model now only conditions on very high-frequency information that only considers the latest attention values.

Second, we find that our 'handcrafted' Recency-Attention-Diversity baseline is instead able to improve over the original model, but its improvements are only marginal. We find these results consistent with section D.2 of the extended analysis, which suggests that the behavior of NAMMs is considerably influenced by a combination of different frequencies in the attention spectrogram which are lost by this approach.

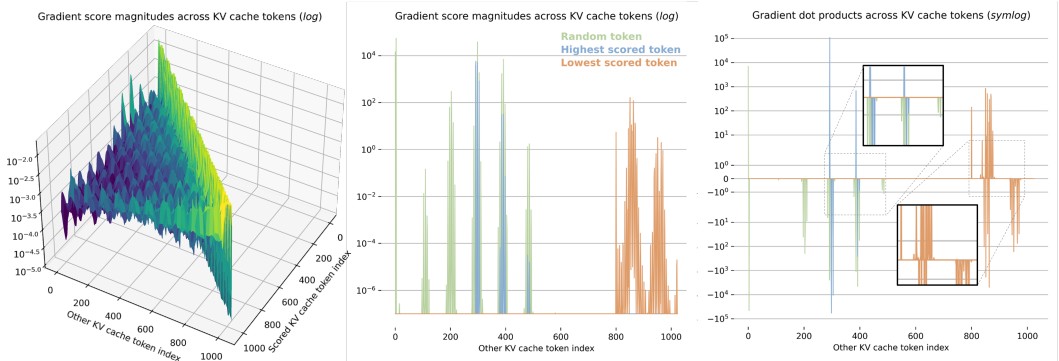

Figure 14: Density plot of gradient magnitudes for each score with respect to all memory tokens (left), together with a qualitative analysis extracting slices from three tokens (center) and computing the dot products of the gradients with the scored-token's feature vector (right).

## D ADDITIONAL ANALYSIS

### D.1 BACKWARD ATTENTION CROSS-TOKEN INTERACTIONS

We analyze the cross-token interactions learned through our BAM architecture by recording the gradients of each token score $s_i$ with respect to all input features $v_j$ for *all tokens in memory* after storing 1024 tokens, i.e., for $j = 1, 2, \ldots, 1024$. We denote these quantities as:

$$\nabla g_i^j = \frac{\partial s_i}{\partial v_j}. \tag{6}$$

We provide a qualitative visualization of our results on the PassageRetrieval-en task for a randomly selected layer and prompt in Figure 14. On the left subplot, we provide a visualization of the squared magnitudes $(\nabla g_i^j)^T \nabla g_i^j$ for each combination of tokens (either scored or attended upon in BAM, i.e., indexed by $i$ or $j$). Here, the effects of the backward mask are clearly visible, allowing tokens to exclusively attend to later ones, where $i > j$. Predictably, these magnitudes mostly peak on the subplot's diagonal, indicating the self-influence that each token's features have on its corresponding output score. However, there are also notable exceptions, as shown in the center subplot, where we overlap three slices from our left surface plot corresponding to the gradients of the first, together with the highest and lowest-scored tokens in memory (respectively indexed by $i =$0, 292, and 800). We provide additional directional information of each gradient vector from these slices in the right subplot, where we take its dot product with the scored token's own feature vector $(\nabla g_i^j)^T v_i$. After the first notable spike, at $i = j$, most other dot-product spikes with the largest magnitudes consistently have negative values. Hence we can logically deduce that the scores of these tokens would benefit from pushing the representations of future tokens away from their own. This result appears to validate the hypothesis that BAM learns a mechanism for cross-token competition, incentivizing diversity and promoting tokens covering unique frequencies in the attention spectrogram.

### D.2 SENSITIVITY TO ATTENTION FREQUENCIES AND POSITIONAL ENCODINGS

We analyze the magnitudes of the gradients of the token scores $s_i$ with respect to each dimension in the token feature vectors. This procedure quantifies how varying each dimension in our attention spectrogram representation locally affects the output score of NAMMs, thus, providing a heuristic measure of its relevance (since scores determine which tokens get discarded). In Figure 15, we plot the distribution of magnitudes for all the seventeen features up to the Nyquist frequency (0 to 16) in the attention spectrogram. All frequency distributions seem to cover a wide range of values, with each mean being close to the global mean, seemingly indicating NAMMs learn to make use of all available spectrogram information for at least some of the tokens. Additionally, we note that many of the higher frequencies have distributions with higher means and larger tails than the 'ground frequency' at dimension 0. Furthermore, as shown in the rightmost-lower subplot, NAMMs appear visibly less sensitive to recency information provided by the concatenated positional embeddings, with a lower total influence than frequency information on token scores. Overall, these observations seem to further validate the importance of going beyond simple hand-designed methods solely based

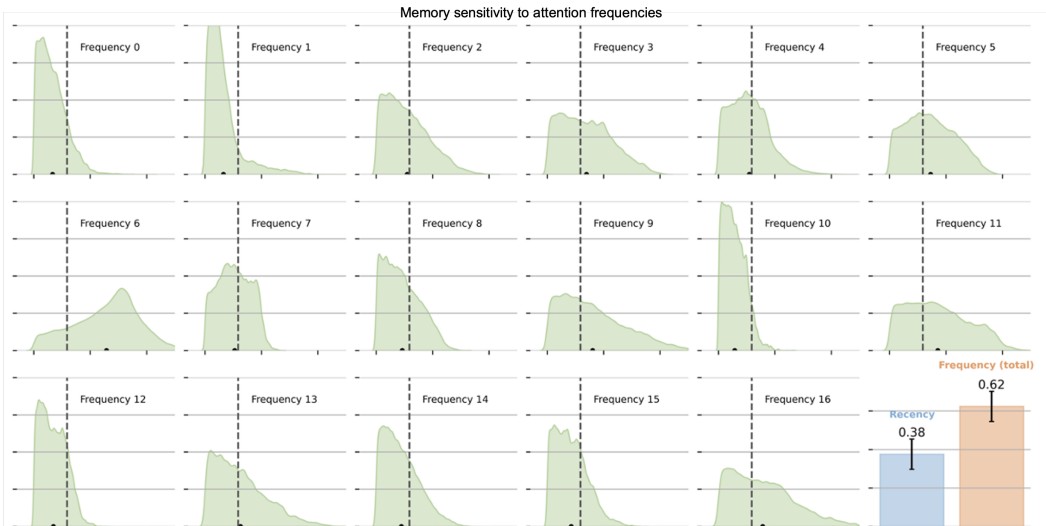

Figure 15: Distribution of gradient magnitudes for the token scores with respect to all the seventeen features in our attention spectrogram representations. In the rightmost-lower subplot, we also compare the total magnitudes of the frequency information with the recency information in the positional embeddings.

on token recency and the sum of the attention values, which has so far been considered a strong established recipe for KV cache management (Oren et al., 2024; Zhang et al., 2024c; Ge et al., 2024; Devoto et al., 2024).

### D.3 INFINITEBENCH RESULTS COMPARISON

On the InfiniteBench tasks, our NAMM achieve particularly outstanding improvements over the base model and other baselines, with an over ten-fold score increase (from 1.05% to 11%). However, we note that even with NAMMs, the performance of Llama 3 8B still lags considerably behind the performance of powerful LMs designed specifically for long-context problems, as reported in Zhang et al. (2024a). Nonetheless, on the En.Sum task, concerned with the summarization of fictitious novels, we find our main NAMM brings the performance of the context-extended Llama 3 from 7.73 to 14.91 even slightly beyond GPT4's (14.73). While this performance is still low in absolute terms[12], such a result appears quite notable and suggests that improvements from NAMMs are orthogonal in nature to the ones brought by architectural improvements and scaling, which, by themselves, might be insufficient to address the challenges brought by long and noisy contexts.

We qualitatively inspect the effects of NAMMs on En.Sum by comparing example answers generated by Llama 3 with and without our memory models, together with examples generated by GPT4. As illustrated in Figure 16, we find both the Llama and GPT models to incur several failure modes, producing answers that entirely miss the objective of the original task. For instance, the context-extended Llama 3 often gets stuck in generation loops continuously repeating part of sentences without coherent structure. Instead, the GPT answers appear to forego summarizing the text and rather attempt to continue the provided passage, by generating end-of-text tokens or even roleplaying some of the characters. However, while introducing NAMMs appears to avoid many instances of these failure modes, we find the summarization of the memory-augmented Llama 3 still displays many imperfections such as misspelling character names (left) or lacking much depth by being extremely concise (right).

---

[12]InfiniteBench tasks are scored in a range between 0 and 100.

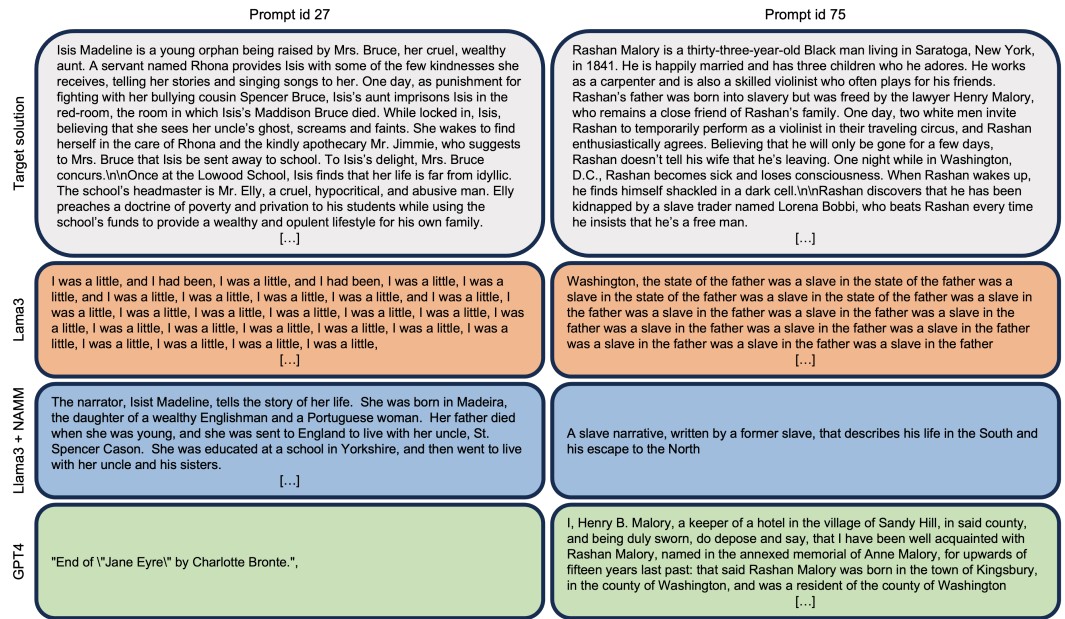

Figure 16: Qualitative examples comparing the ground produced responses by Llama3 with and without our NAMM memory, together with GPT4, on two prompts from the En.Sum task part of InfiniteBench.

# E    EXTENDED RELATED WORKS

Similar to our NAMMs implementation, memory management through token eviction has been explored mostly to reduce memory constraints and enable querying LMs with longer contexts (Luohe et al., 2024). Commonly, strategies entail simply cropping input prompts to a shorter length, often more effective when done from the middle rather than the ends (Xiao et al., 2023; Jin et al., 2024). More advanced, several heuristic strategies have been proposed to identify and evict the least important tokens in the KV cache, selectively pruning it to a fixed size for each layer. These strategies assess token relevance using metrics like L2 magnitude (Devoto et al., 2024) or entropy (Yao et al., 2024), or analyze statistics from the attention matrix, such as value magnitude or cumulative sums (Liu et al., 2024b; Oren et al., 2024; Zhang et al., 2024c). Building on these ideas, Ge et al. (2024) and Li et al. (2024b) apply multiple strategies simultaneously, choosing the best fit for each layer by matching them with specific attention patterns. Similar ideas where also explored in older work targeting encoder-decoder models, for instance, Huang et al. (2022) proposed a more complex strategy for token selection based on solving the *core-set* problem with a parallelized greedy approach. However, unlike previous work, our approach uniquely employs a black-box model to *learn* KV cache management in order to boost the base model's performance with improved efficiency coming as a free side benefit.

Many other methods to reduce memory consumption, affecting the KV cache, are mostly orthogonal and likely complementary to our approach. For instance, MQA (Shazeer, 2019) and GQA (Ainslie et al., 2023) propose merging different attention heads during the training of LLMs, either fully or partially, to improve deployment-time throughput. Brandon et al. (2024), pushed these strategies further, attempting to merge heads even across different layers. GQA is commonly employed in many modern LMs, including the LLama 3 family of models which we use to train and evaluate NAMMs on language tasks (Dubey et al., 2024). Furthermore, several methods have looked at KV cache compression through either quantization of the keys and values (Hooper et al., 2024; Dong et al., 2024a;b) or even the whole hidden states (DeepSeek-AI et al., 2024). Similarly to the aforementioned prior work concerning KV cache pruning, these methods considered mainly hand-designed strategies, such as employing different quantization rates based on heuristically recognizing important tokens. We note that using evolution to optimize for which channels to merge or compress could

also yield new interesting unexplored approaches, combining these orthogonal directions with some of the principles introduced by NAMMs.

There has also been much research interest in exploring new architectures to explicitly model components of a memory system or to address key challenges of reasoning over longer contexts. For instance, past work has looked at incorporating neural models of memory within neural networks by implementing different reading and writing operations - either directly replacing their layers (Weston et al., 2014; Sukhbaatar et al., 2015), or introducing new auxiliary components (Rae et al., 2016; Lample et al., 2019). In relation to transformers, more recent works have been proposed rethinking the ingredients of the self-attention operation, mostly in the context of LMs. These works looked at either efficient linear approximation to self-attention to overcome quadratic costs (Beltagy et al., 2020; Katharopoulos et al., 2020; Wang et al., 2020; Peng et al., 2021), or introducing new kinds of persistent tokens and storage to extend information propagation (Dai et al., 2019; Munkhdalai et al., 2024; Hwang et al., 2024). However, as also noted by Dao et al. (2022), none of these methods and approximations have managed to replace standard approaches so far. We take a different approach that can be integrated in a zero-shot manner even without any fine-tuning.

Lastly, methodologically related to NAMMs, there have been other prior methods making use of evolution for or with transformer models. For example, Tang & Ha (2021) also trained a small attention-based model through evolution, exploiting the inherent parameter efficiency behind these operations. Furthermore, So et al. (2019) proposed using evolution to meta-optimize the basic building of transformers via neural architecture search, while Akiba et al. (2024) focused on evolving different merging strategies across layers belonging to LMs with different capabilities. As for these works, we note that evolution plays a critical role for NAMMs, allowing us to directly optimize for target performance and overcome the inherent non-differentiability underlying our new framework.

Table 24: NAMMs evaluation on the canonical *Needle In A Haystack* task (Kamradt, 2024). The normalized performance (in brackets) is calculated using the base model with full cache.

| Model/Task name | Needle prompt length | | | Overall | |
|---|---|---|---|---|---|
| | *0-10000* | *10001-20000* | *20001+* | **All prompt lengths** | **Cache size** |
| Base model (full cache) | 8.87 (1.00) | **7.53** (1.00) | **3.50** (1.00) | **6.32** (1.00) | 32768 |
| NAMM (BAM) | **9.00** (1.02) | 4.80 (0.64) | 3.05 (0.87) | 5.36 (0.85) | 10208 |
| NAMM (BAM), $\gamma = 0.9999^{s_w}$ | **9.00** (1.02) | 5.33 (0.71) | 3.45 (0.99) | 5.68 (0.90) | 10347 |

# F LIMITATIONS AND FUTURE EXTENSIONS

## F.1 EXPLORING THE DESIGN SPACE OF NEURAL ATTENTION MEMORY MODELS

In this work, we introduced Neural Attention Memory Models and showed their efficacy and potential to improve the performance and efficiency of transformers, even when evaluated zero-shot for unseen architectures and domains. However, given the novelty of our framework, we note that our design choices were mostly motivated by simplicity and practicality rather than quantitative empirical evidence. Thus, there is an extremely large design space in terms of the implementation, training, and deployment of these models that should be explored beyond this work, which is likely to yield further improvements.

For instance, while our current feature extraction, based on computing the spectrogram of the attention matrix, enables capturing global frequency information about the attention values of each token, it might fall short of modeling local information with enough granularity. This hypothesized limitation inherently comes from a few design choices we made with the purpose of limiting the input size and corresponding parameter count of our memory models. In particular, our spectrogram features only consider the real components of a short-time Fourier transform with a small Hann window of size thirty-two. Thus, we only provide NAMMs information about a relatively limited number of thirty-two frequencies, losing any notion of the phase of the attention matrix that would be captured by the full complex-valued Fourier coefficients. Consequently, the representations of tokens with high attention values for entirely non-overlapping queries occurring with the same frequency would be indistinguishable to our models. Moreover, our exponentially moving average reduction over the time dimension of the spectrograms provides an additional layer of heavy compression inevitably trading off expressivity for simplicity.

To partially address these concerns, an alternative design we explored entailed delaying the initial element-wise exponentially moving average reduction. Concretely, this involved computing $T$ different scores, feeding $m_\phi$ all feature vectors $\omega_i^t$ for $t = 1, 2, \ldots, T$, across the attention spectrogram's compressed time axis, only then reducing the resulting scores $s_i^{1:T}$ via EMA. While, in principle, this alternative ordering would allow for additional expressivity without adding to the parameter count, in practice, when evaluated with an initial version of the simple 2-layer MLP model, we found no significant performance difference and opted for the former lighter option. However, introducing cross-token interactions with the improved BAM design and further scaling is likely to introduce a need of re-evaluating this choice.

One further limitation comes from the current reliance on the exact values of the attention matrix. This reliance precludes NAMMs training from making use of fast kernel algorithms developed to accelerate inference by foregoing materializing attention values (Dao et al., 2022). While the main focus of this work has been to introduce NAMMs and display its potential to improve transformers across different domains, more scalable parameterizations and efficient backend integrations remain exciting open challenges for future research.

## F.2 IMPROVING LONG-CONTEXT SPARSE RETRIEVALS

One notable example exemplifying some of the aforementioned limitations, comes from the canonical *Needle In A Haystack* task (Kamradt, 2024), which has been used to qualitatively evaluate LLMs for their ability to remember sparse information over long *noisy* horizons. We provide results on this task using the best-performing NAMM after three stages of incremental training with the BAM

architecture, averaging evaluation scores provided by a GPT-4 model (Achiam et al., 2023) across different prompt ranges, consistently with Bai et al. (2024). As shown in Table 24, while NAMMs do not manage to exceed the overall performance of the base model, they still provide some notable efficiency gains. However, looking more closely at the score distribution across different prompt length ranges we observe an unexpected trend that is in contrast with the rest of our results on other benchmarks. In particular, while our NAMM obtains slightly higher than the base model for prompts with a size less than 10000, it seems to increasingly struggle with longer prompts.

After comparing the spectrogram features extracted for the different prompts, our explanation for these results highlights one current failure mode of the current implementation. In particular, the Needle In a Haystack task is constructed such that the model is tasked to remember some important information introduced at the beginning of the prompt, and later followed by completely unrelated 'filler' text. Hence, the attention scores and the corresponding spectrogram features for the tokens containing the relevant information are forcibly sparse, being high only at the very beginning of the prompt. Yet, since the evaluated NAMM reduces these features over the time axis of the spectrogram with an EMA coefficient of $\gamma = 0.99^{s_w}$, all the frequency information regarding these tokens will be inevitably overwritten. To empirically validate our theory we provide results simply raising the EMA coefficient from $\gamma = 0.99^{s_w}$ to $\gamma = 0.9999^{s_w}$. Since our NAMMs was never actually trained with this higher coefficient, we note that this change effectively brings the input features out-of-distribution. Nonetheless, as shown in the final row of Table 24, the larger coefficient still manages to improve performance on the longer prompts by enabling the preservation of the frequency components from the target 'needle' over a longer horizon. These findings suggest that future NAMM designs should consider higher EMA reduction coefficients or, potentially, even directly *learning* this parameter with evolution in addition to the NAMM's network weights.

