# OpenReview forum: "An Evolved Universal Transformer Memory"
_ICLR.cc/2025/Conference — ICLR 2025 Poster_

### Official Review · Reviewer_845X · 2024-10-31

**Soundness:** 2
**Presentation:** 3
**Contribution:** 3
**Rating:** 6
**Confidence:** 4

**Summary:**

This paper focuses on optimizing the memory management of Transformers. The authors introduce the NAMMs (Neural Attention Memory models) method, which employs trainable parameters to learn the importance of token column vectors. By removing less important tokens, this approach reduces resource requirements and enhances efficiency. The effectiveness of the proposed method is validated through experiments conducted on multiple datasets.

**Strengths:**

1. The paper is well-structured with a clear logical flow, making it easy for readers to follow. Figures are appropriately placed alongside the relevant text, contributing to a harmonious overall layout. The experimental results are effectively highlighted and processed, allowing readers to easily grasp the key points. This is a significant strength of the paper.

2. The NAMMs method proposed in the paper achieves a reduction in model parameter size, which facilitates training.

3. The architecture introduced by the authors is straightforward and easily reproducible. Extensive experiments have been conducted to substantiate its effectiveness, further validating the claims made in the paper.

**Weaknesses:**

1 My understanding is that the method involves the fusion of column vectors for all tokens (3.1 169）. However, if the dimensionality is too high, could this lead to substantial additional computational time? Please provide some computational complexity analysis or empirical runtime comparisons.

2 Removing tokens with importance scores <0 implies that the number of removed tokens may vary each time. How can the consistency of dimensions be guaranteed? Without consistent dimensions, subsequent training would be impossible. Does this require an additional constraint? I feel this point was not clearly explained in the paper. (How is the following issue implemented？）
 a.How they handle varying numbers of removed tokens in practice
 b.If there are any mechanisms to ensure dimension consistency
 c.Whether any additional constraints are used

3 In my understanding, compression is performed before assessing the importance of tokens. Would it be possible to compare the effects without compression? It seems that compression also has its advantages. Please conduct an ablation study specifically comparing performance with and without the compression step.

4.Could you list specific learning rates, parameter settings, and other details? Publishing the experimental code would also be beneficial. (Please provide the following materials.)
 a.A table of hyperparameters used for each experiment
 b.Any details on hyperparameter tuning processes
 c.Information on where to access their code or if they plan to release it This would give readers a clearer picture of how to reproduce the results.

**Questions:**

1 Removing tokens with importance scores <0 implies that the number of removed tokens may vary each time.

 2.How can the consistency of dimensions be guaranteed? Without consistent dimensions, subsequent training would be impossible.

3.Does this require an additional constraint? I feel this point was not clearly explained in the paper.

4.In my understanding, compression is performed before assessing the importance of tokens. Would it be possible to compare the effects without compression? It seems that compression also has its advantages.

5.Could you list specific learning rates, parameter settings, and other details? Publishing the experimental code would also be beneficial.

---

> ### Author Response · Authors · 2024-11-23
> **Response to Reviewer 4 (845X) - Part 1**
>
> We thank the reviewer for taking the time to provide thoughtful comments and suggestions. The following are our responses to the comments and questions, please see the details in the revised PDF.
>
> > 1 My understanding is that the method involves the fusion of column vectors for all tokens (3.1 169）. However, if the dimensionality is too high, could this lead to substantial additional computational time? Please provide some computational complexity analysis or empirical runtime comparisons.
>
> The reviewer is correct that processing attention sequences for all tokens in the KV cache may introduce computational costs. However, we find these costs are minimal compared to the inference costs of running LLMs, thanks to NAMMs being very parameter-efficient modules, and the speedups from NAMMs’ ability to evict tokens from the KV cache far outweigh any overhead.
>
> **Following the reviewer's feedback we added Section C.3 to the Appendix, where we provide runtime comparisons** across two settings: 1) samples from the full LongBench benchmark (average length: 12,099 tokens), and 2) longer samples exceeding the transformer’s max length (average length: 32,641 tokens).
>
> Finally, we also record the running time of an ablation study with a version of our NAMM run on top of the base transformer that does not modify its KV cache (named “NAMM (full cache)” in the following table). This artificially-inefficient ablation was done to disentangle the gains from the reduced memory and better analyze the pure overheads from our model’s execution:
>
>
> | Method| Time per task sample |
> |-----------------------------|----------------------|
> | Longbench average | |
> | Base model (full cache) | 4.0|
> | NAMM (full cache) | 4.54 |
> | H2O | 3.78 |
> | L2| 3.74 |
> | NAMM (Ours) | 3.97 |
> | Selected max length samples | |
> | Base model (full cache) | 15.52|
> | NAMM (full cache) | 17.6 |
> | H2O | 11.79|
> | L2| 10.5 |
> | NAMM (Ours) | 13.61|
>
> These results show that NAMMs incur marginal overheads, as they are small and applied at fixed intervals (e.g., every 512 tokens). Finally, we also record the running time of an ablated version of our NAMM run on top of the base transformer that does not modify its KV cache (named “NAMM (full cache)” in the following table). This artificially-inefficient ablation was done to disentangle the gains from the reduced memory and better analyze the pure overheads from our model’s execution:
>
> Overall, NAMMs achieve efficiency gains comparable to H2O [1] and L2 [2] baselines but offer the critical advantage of improving model performance, unlike the baselines, which have to trade off their performance for these efficiency gains.
>
> > 2 Removing tokens with importance scores <0 implies that the number of removed tokens may vary each time. How can the consistency of dimensions be guaranteed? Without consistent dimensions, subsequent training would be impossible. Does this require an additional constraint? I feel this point was not clearly explained in the paper. (How is the following issue implemented？） a.How they handle varying numbers of removed tokens in practice b.If there are any mechanisms to ensure dimension consistency c.Whether any additional constraints are used
>
>
> As noted by the reviewer there are two dynamic dimensionalities during our training and inference procedures:
>
> 1. Sequence Length of Attention Values: the length of the sequence of attention values used to condition NAMMs inherently grows during generation and varies for each different token. However, we deal with this by compressing this sequence into a fixed-sized vector through a Short-Time Fourier Transform (STFT) and a frequency-wise exponentially-moving average (EMA), detailed in Section 3.1 and illustrated in Figure 3.
>
> Following the reviewer’s feedback, **we extended Section 3.1 in the latest revision to make the explanation clear** and explicitly state that the STFT and EMA allow us to obtain an input for each token with a consistent dimension. **We also added a new paragraph with additional implementation details and an Algorithm block in Section 3.2** to summarize the whole execution pipeline of our model.
>
> 2. Variable KV Cache Sizes: we also note that differently sized KV caches are not a problem for our training procedure as we do not employ any backpropagation. In particular, we optimize NAMMs with CMA-ES [3], an evolutionary algorithm that does not require any gradient information and can directly optimize black-box undifferentiable metrics. This property allows us to both optimize for the non-differentiable token selection task of our NAMMs and also train its parameters to maximize non-differentiable task performance metrics directly.
>
> Following the reviewer’s feedback, **we modified Section 3.3** to make this clearer as summarized above and we also **added Section A.4 to the Appendix, where we provide these exact details together with an additional high-level step-by-step description of the CMA-ES** optimization loop applied to our setting.

---

> ### Author Response · Authors · 2024-11-23
> **Response to Reviewer 4 (845X) - Part 2**
>
> > 3 In my understanding, compression is performed before assessing the importance of tokens. Would it be possible to compare the effects without compression? It seems that compression also has its advantages. Please conduct an ablation study specifically comparing performance with and without the compression step.
>
> As described in our response to Point 2, the compression of raw attention values is performed using an STFT followed by an EMA operation. This strategy was specifically designed to preserve relevant information about each token’s history within a fixed-sized representation. We would also like to note that such use of the STFT to process unbounded 1D input signals has been very well-established in domains like audio [4], medicine [5], and seismology [6], and more.
>
> **To address the reviewer’s request, we added an ablation study in Section C.10 of the Appendix, comparing NAMMs with and without the STFT compression step.** We evaluated two alternative baselines:
> 1. Raw Attention Features: Directly using the set of most recent raw attention values as input.
> 2. Recency-Attention-Diversity (RAD): A handcrafted representation combining three quantities: i) the sum of the attention values of each token, ii) the recency of each token, and iii) the diversity of each token (computed by concatenating the keys and values to represent each token and averaging the L2 distance to all other tokens).
>
>  Please refer to Table 23 of the rebuttal revision or the summarized Table provided below for our results:
>
> | Model                              | LongBench base (mean) | Normalized performance (mean) | Normalized test performance (mean) | Average Cache size |
> |------------------------------------|-----------------------|-------------------------------|------------------------------------|--------------------|
> | Base model                         | 28.86                 | 1.00                          | 1.00                               | 12099              |
> | NAMM (stage 2)                     | 29.25                 | 1.07                          | 1.04                               | 8521               |
> | NAMM (stage 3)                     | 29.33                 | 1.11                          | 1.07                               | 8409               |
> | NAMM (Recency-Attention-Diversity) | 28.71                 | 1.02                          | 1.00                               | 8000               |
> | NAMM (Raw attention features)      | 29.10                 | 1.03                          | 0.95                               | 8523               |
>
> Key findings:
> 1. Raw Attention Features: While showing large improvements on the training tasks, this baseline clearly underperforms on the set of test tasks that were not used for training our NAMM. Thus, we find this result is strongly suggestive of the occurrence of overfitting, which we believe is to be expected as our memory model now only conditions on very high-frequency information that only considers the latest attention values.
> 2. Recency-Attention-Diversity: We find that our ‘handcrafted’ baseline is instead able to improve over the original model, but its improvements are only marginal. We find these results consistent with section D.2 of the extended analysis, which suggests that the behavior of NAMMs is considerably influenced by a combination of different frequencies in the attention spectrogram which are lost by this approach.
>
> These results empirically validate that our compression step via STFT and EMA is key for our NAMMs, allowing us to learn expressive memory models conditioned on information from the full history of attention values and able to effectively generalize beyond the training tasks.

---

> ### Author Response · Authors · 2024-11-23
> **Response to Reviewer 4 (845X) - Part 3**
>
> > 4.Could you list specific learning rates, parameter settings, and other details? Publishing the experimental code would also be beneficial. (Please provide the following materials.) a.A table of hyperparameters used for each experiment b.Any details on hyperparameter tuning processes c.Information on where to access their code or if they plan to release it This would give readers a clearer picture of how to reproduce the results.
>
> **We would like to note that we did already share our full code, including our full and parallelized training pipeline,** which can be found in its anonymized version in the supplementary material link provided above with our submission.
>
> Furthermore, **across Section A of the Appendix, we provide a full list of details** used for both training and validation of our models. These include hyper-parameters, training specifications, tuning details for our method and the baselines, and more.
>
> We agree with the reviewer about the importance of open and reproducible research. We hope our code and provided details will allow the rest of the community to extend and contribute to our new line of research.

---

> ### Author Response · Authors · 2024-11-23
> **Response to Reviewer 4 (845X) - Part 4**
>
> > Q1. Removing tokens with importance scores <0 implies that the number of removed tokens may vary each time.
> > Q2.How can the consistency of dimensions be guaranteed? Without consistent dimensions, subsequent training would be impossible.
> > Q3.Does this require an additional constraint? I feel this point was not clearly explained in the paper.
>
> As these questions are constinuent parts of Point 2, please see our response to Point 2 in Part 1.
>
>
> > Q4. In my understanding, compression is performed before assessing the importance of tokens. Would it be possible to compare the effects without compression? It seems that compression also has its advantages.
>
> As this question is a replicate of Point 3, please see our reponse to Point 3 in Part 2.
>
> > Q5. Could you list specific learning rates, parameter settings, and other details? Publishing the experimental code would also be beneficial.
>
> As this question is a replicate of Point 4, please see our response to Point 4 in Part 3.
>
>
> **References**
>
> [1] Zhang, Zhenyu, et al. "H2o: Heavy-hitter oracle for efficient generative inference of large language models." Advances in Neural Information Processing Systems 36 (2023): 34661-34710.
>
> [2] Devoto, Alessio, et al. "A Simple and Effective $ L_2 $ Norm-Based Strategy for KV Cache Compression." arXiv preprint arXiv:2406.11430 (2024).
>
> [3] Hansen, Nikolaus. "The CMA evolution strategy: A tutorial." arXiv preprint arXiv:1604.00772 (2016).
>
> [4] Allen, J. "Applications of the short time Fourier transform to speech processing and spectral analysis." ICASSP'82. IEEE International Conference on Acoustics, Speech, and Signal Processing. Vol. 7. IEEE, 1982.
>
> [5] Kıymık, M. Kemal, et al. "Comparison of STFT and wavelet transform methods in determining epileptic seizure activity in EEG signals for real-time application." Computers in biology and medicine 35.7 (2005): 603-616.
>
> [6] Astuti, W., et al. "Adaptive Short Time Fourier Transform (STFT) Analysis of seismic electric signal (SES): A comparison of Hamming and rectangular window." 2012 IEEE symposium on industrial electronics and applications. IEEE, 2012.

---

> > ### Author Response · Authors · 2024-12-01
> > **Follow-up to Rebuttal**
> >
> > We would like to thank once again Reviewer 845X for their time and efforts in reviewing our work.
> >
> > With our response and rebuttal revision, we made our best efforts to try and address each one of the points raised in their constructive feedback. As the discussion period is coming to a close, we would really appreciate it if they could let us know if they found our response satisfactory and their revised opinion.
> >
> > Please, do not hesitate to raise any further questions or concerns about our work.

---

### Official Review · Reviewer_J4vq · 2024-11-03

**Soundness:** 2
**Presentation:** 2
**Contribution:** 2
**Rating:** 6
**Confidence:** 3

**Summary:**

The paper proposes Neural Attention Memory Models (NAMMs), a network learned via evolution (CMA-ES) that can be applied to Transformers, reducing the number of tokens needed (KV-cache) at inference time.
Results show that NAMMs outperform recent methods for pruning the KV-cache and occasionally improves the performance of the base model.

**Strengths:**

The performance gains using NAMMs over H2O are significant.

Improving the performance using NAMMs over the base model is impressive.

**Weaknesses:**

H2O has some advantages over NAMMs. Notably, H2O does not require any new training. In contrast, NAMMs are trained using evolution. As such, NAMMs may require significantly more compute. The paper, however, lacks information regarding the computational resources needed by the method.

Furthermore, in practice, we care about the cache size and inference computational complexity. It is unclear how (1) the performance of NAMMs fare when varying the cache size and (2) the complexity (runtime and memory) of applying NAMMs at inference time.

Several details are missing to understand if the idea will generalize to other models. In this paper, only Llama 3 8B was considered. However, in contrast, the H2O paper considered several architectures (e.g., OPT,
LLaMA, and GPT-NeoX), showing their idea generalizes across architectures. However, in this case, it's unclear how NAMMs performance will generalize across different language models.

The results on Decision Transformer (DT) are odd considering that several papers (e.g., the original DT and Online DT papers) showed results that are significantly better than what is reported in Table 7.

Benchmarks in the paper are mentioned by name (e.g., Long Bench, InfiniteBench, LongVideoBench, MLVU, D4RL etc...) without concrete details (e.g., types of tasks in the benchmarks' datasets). It is unclear to readers who are unfamiliar with the various fields.

**Questions:**

The cache size is important for complexity. However, in practice, what we actually care about is the empirical runtime and memory complexities. What is the runtime and memory needed to process new tokens compared to that of the baselines (e.g., H2O)?

Evolution is typically expensive to train using. What are the computational resources used to train NAMMs in terms of the runtime and memory?

Considering that language models get significantly larger than 8B, it is important that the idea generalizes to larger settings. How would the resources needed to train and perform inference scale with the size of the base models?

When pruning, the size of the cache size affects the performance significantly. In practice, we may want to select an optimal cache size that trades off performance and computational resources. In this paper, only 1 cache size was chosen for each dataset. How was this cache size determined for each model? And why are the results reported for only a single cache size?

Relatedly, in H2O, plots are shown with the x-axis being the cache size (ratio) and y-axis being the performance. As a result, we can see the performance trade-off with respect to the cache size. Could you elaborate on why the results were not shown the same way? And could you include these plots?

Could you include details in the Appendix regarding the various datasets (e.g., Long Bench, InfiniteBench, LongVideoBench, MLVU, D4RL etc...)?

The paper runs experiments on D4RL using Decision Transformers. However, the reported results are significantly worse than what is reported in many papers (e.g., original DT paper, Online DT, Aaren, etc..). Could you elaborate on why this is the case? Could you add details regarding the experimental setup that resulted in these numbers?

For the LongBench results, the overall numbers are reported as "All tasks". However, a subset of the tasks are used for training NAMM. Reporting "All tasks" seems unfair for evaluating generalization. Instead, could you report separate aggregate scores for (1) the tasks used in training and (2) the held-out tasks not used in training, to more clearly show generalization performance."



Interestingly, using NAMM improves over even the base model in several cases. What is your intuition as to why there's such a large improvement in some cases?

---

> ### Author Response · Authors · 2024-11-23
> **Response to Reviewer 3 (J4vq) - Part 1**
>
> We thank the reviewer and appreciate the detailed comments and concrete suggestions. The following are our responses to the comments and questions, please see the details in the revised PDF.
>
> **Feedback points**
>
> > 1 Inference and Memory with NAMMs - From Question 1/Weakness 2:
> [...] What is the runtime and memory needed to process new tokens compared to that of the baselines (e.g., H2O)?
> It is unclear [...] the complexity (runtime and memory) of applying NAMMs at inference time [...]
>
> Following the reviewer’s feedback, **we added Section C.6 to the Appendix, detailing and discussing inference time costs, GPU memory savings, and training time costs of our NAMM together with the H2O [1] and L2 [2] baselines** - using our publicly shared codebase. We also provide a full description of our hardware setup. Below, we provide a summary of this discussion with the main inference-time and memory results for easier reference (Tables 19-21 in the rebuttal revision).
>
> In terms of inference, we recorded the average running times in two settings: 1) Using samples from the full LongBench benchmark with an average length of 12099 2) Using only samples from LongBench selected to exceed the base transformer maximum length with an average length of 32641. Finally, we also record the running time of an ablated version of our NAMM run on top of the base transformer that does not modify its KV cache (named “NAMM (full cache)” in the following table). This artificially-inefficient ablation was done to disentangle the gains from the reduced memory and better analyze the pure overheads from our model’s execution:
>
> | Method                      | Time per task sample |
> |-----------------------------|----------------------|
> | Longbench average           |                      |
> | Base model (full cache)     | 4.0                  |
> | NAMM (full cache)           | 4.54                 |
> | H2O                         | 3.78                 |
> | L2                          | 3.74                 |
> | NAMM (Ours)                 | 3.97                 |
> | Selected max length samples |                      |
> | Base model (full cache)     | 15.52                |
> | NAMM (full cache)           | 17.6                 |
> | H2O                         | 11.79                |
> | L2                          | 10.5                 |
> | NAMM (Ours)                 | 13.61                |
>
> In terms of memory, we reported estimated effects in peak GPU memory consumption. These results were calculated from the peak KV cache sizes, together with the sizes of additional information (e.g., attention matrix) and models used by each method (again recorded on LongBench):
>
> | Method                  | KV cache storage (MB) | Additional storage overheads (MB) | Total savings (MB)|
> |-------------------------|------------------|------------------------------|---------------|
> | Base model (full cache) | 65282            | 0                            | 0             |
> | H2O                     | 17408            | 5659                         | 42215         |
> | L2                      | 17408            | 0                            | 47874         |
> | NAMM                    | 27236            | 5668                         | 32378         |
>
> As shown in the above summarized tables, due to NAMMs being much smaller than the base model and that they are applied at ‘constant intervals’ (every 512 tokens seen by the base transformer in our case) their overheads appear to be marginal. Moreover, as many operations with modern deep learning are memory-bound, the resulting reduction in KV cache size from using NAMMs or any of our baselines seems to offset these overheads and provide significant gains both in terms of inference-time and memory.
>
> Overall, the efficiency gains appear very close to the H2O and L2 baselines. However, we note that for both H2O and L2 these efficiency gains come at a loss of performance. In contrast, NAMMs concretely improve the base model’s behavior, with its efficiency gains being free by-products of its fully-learned memory management strategy.

---

> ### Author Response · Authors · 2024-11-23
> **Response to Reviewer 3 (J4vq) - Part 2**
>
> > 2 Training and scaling NAMMs - From Questions 2-3/Weakness 1:
> [...] What are the computational resources used to train NAMMs in terms of the runtime and memory?
> [...] How would the resources needed to train and perform inference scale with the size of the base models?
> [...] NAMMs are trained using evolution. [...] The paper lacks information regarding the computational resources needed by the method.
>
> As noted in the above response, following the reviewer’s feedback, **we added Section C.6 to the Appendix, detailing and discussing inference time costs, GPU memory savings, and training time costs of our NAMMs together with the H2O and L2 baselines** - using our publicly shared codebase. Below, we provide a summary of this discussion for training and scaling NAMMs with the main training-time results for easier reference:
>
> We optimize NAMMs with the Covariance Matrix Adaptation Evolution Strategy (CMA-ES) [3]. Being an evolutionary algorithm, CMA-ES does not require any gradient information and can directly optimize by simply running inference with the base model (i.e., without additional time or memory overheads from gradient recording, backpropagation etc.).
>
> Thus, from our use of evolution, **the training time of NAMMs are dominated, in the worst case, by the cost of running inference with the original large language model** (in our case Llama 3 8B). Moreover, **no additional overheads are introduced when scaling up the number of parameters of the base model.** In practice, training time costs are even inferior thanks to the speedups provided by our NAMM itself as reported in our inference and memory recordings above.
>
> However, we note linearly increasing training time when scaling the number of tasks. This motivated our ‘incremental’ approach (i.e., using a single task for phase 1, and adding a new task in each additional phase) and our task ordering (from faster to slowest to respond). To provide specific details using our specific hardware configuration, we provide the recorded time taken per generation in the following Table:
>
> | Incremental phase| Time per generation (s) |
> |--------------------------------|-------------------------|
> | Phase 1 (+PassageRetrieval-en) | 1597.62 |
> | Phase 2 (+DuReader)| 4027.33 |
> | Phase 3 (+NarrativeQA) | 9848.31 |
>
> As detailed in our new Section C.6, these times were collected training NAMMs on a single compute node. However, thanks to the highly parallelizable nature of evolution, training times would obtain linear speedups given multi-node configurations (for which our shared code is already fully-compatible with). Moreover, we would like to note that our NAMM was only trained once over a few-hundred generations, making the overall training costs in terms of GPU hours, orders of magnitude lower than traditional gradient-based training or finetuning of LLMs.

---

> ### Author Response · Authors · 2024-11-23
> **Response to Reviewer 3 (J4vq) - Part 3**
>
> > 3 Performance as a function of cache size - From Questions 4-5:
> [...] How was this cache size determined for each model?
> Relatedly, in H2O, plots are shown with the x-axis being the cache size and y-axis being the performance. [...] could you include these plots?
>
> Unlike prior methods, where the cache size can be seen as a hyper-parameter, NAMMs are optimized end-to-end to improve task performance by managing the memory of the base model, in order to discard unhelpful and redundant information and allow each layer to ‘focus’ on the most helpful subset of tokens. Thus, the tokens inside the KV cache and its corresponding size are not manually determined as they are chosen directly by our neural network module for each layer and prompt to yield the best performance.
>
> In contrast, both H2O and L2 do have a single main ‘maximum cache size’ hyper-parameter that determines their memory efficiency. This was chosen following preliminary tests in order to avoid losing too much performance on LongBench over the base model while still reducing the KV-cache size to a significant extent.
>
> Following the reviewer’s suggestion, **we now added results on the full set of 21 LongBench tasks varying the maximum cache size of H2O and L2 across [2048, 4096, 8192, 16384, 32768], and comparing with our NAMM**. We added these results in Figure 5 in Section 4.1 of the main text.
>
> Please, refer to the latest rebuttal revision or [**this link**](https://anonymous.4open.science/r/evomemory-rebuttal-1ED4/perf_per_cs.png) for their visualization.
>
> In line with the results in the prior literature of KV cache pruning, we find that the overall performance of H2O and L2  is always inferior to the original model with full cache and monotonically increases with larger cache sizes. Moreover, a big drop in performance can be observed when lowering the maximum cache size to 4096, which we also observed in our prior experiments and motivated our baseline cache size selection of 8192.
>
> We believe this new Figure now clearly displays the dichotomy between hand-designed strategies aimed at performance retention as opposed to our learned NAMMs, which are remarkably able to go beyond the original model’s performance - while still providing substantial efficiency gains.
>
>
> > 4 NAMMs with other language models - Weakness 3:
> [...] it's unclear how NAMMs performance will generalize across different language models.
>
> Following the reviewer’s suggestion, **we now also analyzed applying NAMMs on top of the Mistral 7b base model**. We considered 2 settings:
>
> 1) 0-shot application, taking our best NAMM model trained with the Llama 8b context-extended model.
> 2) With a small amount of additional finetuning (20 generations using CMA-ES and the same 3 training tasks used for Llama).
>
> **We provide our full results and analysis in Table 22 and the new Section C.9 of the Appendix**. We provide a summarized version below:
>
> | Model           | LongBench Mistral base (mean) | Normalized performance (mean) | Average Cache size |
> |-----------------|-------------------------------|-------------------------------|--------------------|
> | Base model      | 30.33                         | 1.00                          | 10107             |
> | H2O             | 28.27                         | 0.96                          | 6662              |
> | L2              | 26.27                             | 0.97                             | 6662              |
> | NAMM (0-shot)   | 30.12                         | 1.04                          | 8518               |
> | NAMM (Finetune) | 30.52                         | 1.08                          | 8654               |
>
> Analogously to our other 0-shot transfer results, we find that NAMMs provide both visible performance and efficiency gains also when transferred to the Mistral model, overcoming the efficiency-performance tradeoff of hand-designed baselines. Furthermore, this analysis also shows that performance could be further improved by a few finetuning generations with different base models. While we did not investigate finetuning with our other models, we believe these results highlight the potential of cheaply improving NAMMs’ 0-shot benefits, which we hope will be further explored in future work.
>
> Overall, we would like to note that we now consider 3 different language models (Llama 3 8b, Llama 3 70b, and Mistral 7b) and also 2 additional foundation models for vision and reinforcement learning (Llava Video Next, Decision transformer). Thus, we hope the reviewer will appreciate our efforts to explore a very wide range of base models and applications in the current revision.

---

> ### Author Response · Authors · 2024-11-23
> **Response to Reviewer 3 (J4vq) - Part 4**
>
> > 5 Decision transformer base model - From Question 7/Weakness 4:
> [...] on Decision Transformer (DT)  [...] several papers (e.g., the original DT and Online DT papers) showed results that are significantly better than what is reported in Table 7.
> [...] using Decision Transformers [...] the reported results are significantly worse than what is reported in many papers [...] Could you add details regarding the experimental setup that resulted in these numbers?
>
> In Appendix A.3 we describe in detail our experimental setting and the hyper-parameter used for our zero-shot transfer experiments, including reinforcement learning:
>
> In particular, rather than retraining DT from scratch, we used the open-sourced checkpoints provided by the huggingface team [4]. As noted by the reviewer, on some task-dataset combinations these checkpoints appear to obtain notably lower performance than what is reported in the original DT paper [5] (e.g., Walker pre-trained on medium-expert data). However, we do not believe that these differences should affect our conclusions as we used the same base decision transformer model to evaluate all our memory management baselines.
>
> Following the reviewer’s feedback, **we now explicitly provide this important detail before introducing the D4RL results in Section 4.2 of the main text.** Furthermore, **we also extended section A.3 of the appendix to describe and explicitly discuss the inconsistency between the shared Huggingface models and the paper’s original reported results**, as summarized above.
>
> If the reviewer deems it further necessary, we are open to training a new decision transformer from scratch on D4RL, so as to obtain a better base checkpoint for applying our NAMMs 0-shot across modalities. If this were the case, we hope they will not hesitate to let us know, and will collect them after the rebuttal period in time for a potential camera-ready revision.
>
>
> > 6 Benchmark details - From Question 7/Weakness 4:
> Benchmarks in the paper are mentioned by name [...] without concrete details (e.g., types of tasks in the benchmarks' datasets). It is unclear to readers who are unfamiliar with the various fields.
> Could you include details in the Appendix regarding the various datasets (e.g., Long Bench, InfiniteBench, LongVideoBench, MLVU, D4RL etc...)?
>
> In Appendix B, we provided a full description of our new Choubun benchmark. This includes details about how each task was constructed, specific task examples, and evaluation results for a wide range of both open and closed-sourced language models. However, for many other existing benchmarks we employed from the literature, we only provided high-level descriptions of the types of tasks in Section 4 and Appendix A.3 (e.g., long-context LLM evaluation, video language evaluation, offline reinforcement learning).
>
> Following the reviewer’s feedback, **we now added Section B.2 to the Appendix, where we provide a more in-depth page-long summary of each individual benchmark we consider**. In particular, we make sure to provide each of the benchmark’s motivations, its target application, and a full list of types of tasks included. We hope this will provide details and background for the unfamiliar reader to comprehensively understand the nature of these benchmarks without referring to their respective papers.
>
> > 7 Test task LongBench results - From Question 8:
> For the LongBench results, the overall numbers are reported as "All tasks". [...] Instead, could you report separate aggregate scores for (1) the tasks used in training and (2) the held-out tasks not used in training, to more clearly show generalization performance.
>
> In all our Tables reporting performance on the full set on LongBench (i.e., Table 2 and Table 5 in Section 4; Table 12, Table 15, Table 18 in the Appendix), on the right of the ‘All tasks’ columns we do provide a ‘Test tasks’ column. This column precisely reports the normalized average performance exclusively on the held-out tasks not used during evolution. In the first paragraph of Section 4.1, we also state “Our NAMM yields concrete improvements [...] both when considering the full set or exclusively the held-out set of test tasks [...] of 11% and 7% respectively.”
>
> Following the reviewer’s comment, we realized these results could be missed amongst the many columns of these tables. Thus, **we have now marked the titles of the ‘test-task’ columns with a blue background** to make them very clearly visible when reading the paper.
>
> We hope the notable performance improvements NAMMs provide not only on the test tasks on LongBench, but also on the entirely ‘unseen’ InfiniteBench and Choubun benchmarks validate that its performance and efficiency benefits extend beyond the tasks chosen for training.

---

> ### Author Response · Authors · 2024-11-23
> **Response to Reviewer 3 (J4vq) - Part 5**
>
> > 8 How NAMM is inherently different from prior approaches - From Question 9:
> Interestingly, using NAMM improves over even the base model in several cases. What is your intuition as to why there's such a large improvement in some cases?
>
> Prior methods for KV-cache memory management (e.g. H2O [1], L2 [2], FastGen [6], …) are all designed to drop tokens for a fixed target level of efficiency while trying to minimize the loss of information using hand-designed heuristics (e.g., preserving high attention values, high L2 norms, punctuation, …).
>
> In contrast to these approaches, our NAMMs are never forced to drop information as they are neural network modules exclusively optimized for performance. Thus, NAMMs could still potentially learn to retain the full KV-cache and not prune any token in case this would yield the best performance. However, one of the key hypotheses of our work is that there is a large amount of redundant and misplaced information that is actually hurting the base model. We tried to extensively validate this hypothesis within our results, showing that NAMMs are actually able to consistently improve the full-cache performance by learning to forget this unhelpful information allowing the base model to ‘focus’.
>
> **We provide some qualitative examples in Figure 7 of Section 4.3**, showing that the unhelpful information our NAMMs learn to prune in natural language tasks includes grammatical redundancies and repeated words, while in coding tasks includes comments and some import statements. Moreover, the discarded information seems to vary in different layers of the base model. For instance, only particular layers seem to need information about each task’s preamble.
>
> As noted by the reviewer, we also observe that in some tasks with particularly long context, such as in InfiniBench, these gains are extremely significant (going from an overall score of 1.05 for the full-cache model up to 11 with NAMMs). **In Figure 16 of Section D.3, we provide an in-depth qualitative analysis of one such task**, where NAMMs atop the base Llama 8B model seem to even outperform the performance of some of the strongest private models such as GPT4. In this example, we show that NAMMs are able to overcome several of the failure modes of both the full cache model and GPT4 such as forgetting about the original question and continuing the prompt by roleplaying, or even getting stuck in ‘generation loops.’
>
>
> **References**
>
> [1] Zhang, Zhenyu, et al. "H2o: Heavy-hitter oracle for efficient generative inference of large language models." Advances in Neural Information Processing Systems 36 (2023): 34661-34710.
>
> [2] Devoto, Alessio, et al. "A Simple and Effective $ L_2 $ Norm-Based Strategy for KV Cache Compression." arXiv preprint arXiv:2406.11430 (2024).
>
> [3] Hansen, Nikolaus. "The CMA evolution strategy: A tutorial." arXiv preprint arXiv:1604.00772 (2016).
>
> [4] Edward Beeching and Thomas Simonini. Introducing decision transformers on hugging face, Mar
> 2022. URL https://huggingface.co/blog/decision-transformers.
>
> [5] Chen, Lili, et al. "Decision transformer: Reinforcement learning via sequence modeling." Advances in neural information processing systems 34 (2021): 15084-15097.
>
> [6] Ge, Suyu, et al. "Model tells you what to discard: Adaptive kv cache compression for llms." arXiv preprint arXiv:2310.01801 (2023).

---

> > ### Comment · Reviewer_J4vq · 2024-11-26
> >
> > Thank you for your comprehensive response. I have updated my score accordingly (3 -> 6).

---

### Official Review · Reviewer_tak4 · 2024-11-04

**Soundness:** 3
**Presentation:** 3
**Contribution:** 3
**Rating:** 8
**Confidence:** 4

**Summary:**

Transformers are known to suffer from quadratic memory blow-ups. This paper attempts to make Transformers more efficient by using Neural Attention Memory Models (NAMMs). This involves pruning the KV cache memory using NAMMs.

**Strengths:**

1. The idea behind the contribution is simple and easy to understand.
2. NAMMs can be zero-shot transferred to other Transformers, which is quite interesting.
3. The benchmarks are diverse (but not comprehensive as discussed in weakness 2).

**Weaknesses:**

1. Pruning is lossy, meaning that it results in losing some of the information in the KV cache memory. For example, during the pruning process, you may very well discard information that the model could have found useful later on.
2. KV cache memory pruning (more broadly KV cache management) has been investigated in the last couple of years, and while the authors compare against some of them, like H2O and L2, they do not discuss or compare against some of the other relevant work. Some of the relevant work (in language modelling) include:
   - Model Tells You What to Discard: Adaptive KV Cache Compression for LLMs, ICLR 2024
   - CacheGen: KV Cache Compression and Streaming for Fast Large Language Model Serving, SIGCOMM 2024

I appreciate that comparing against so many related works can be cumbersome. However, it is crucial that the authors discuss the related work and either compare their work against them or discuss why they are not comparable.

**Questions:**

Please see the weaknesses.

---

> ### Author Response · Authors · 2024-11-23
> **Response to Reviewer 2 (tak4) - Part 1**
>
> We thank the reviewer for the feedback and concrete actionable suggestions. The following are our responses to the comments and questions, please see the details in the revised PDF.
>
> **Weaknesses**
>
> > 1 [...] during the [memory] pruning process, you may very well discard information that the model could have found useful later on.
>
> Prior methods for KV-cache memory management (e.g. H2O [1], L2 [2], FastGen [3], …) are all designed to drop tokens for a fixed target level of efficiency while trying to minimize the loss of information using hand-designed heuristics (e.g., preserving high attention values, high L2 norms, punctuation, …). As noted by the reviewer, all these heuristics must still discard information that the model might find useful later on in order to achieve the specified target efficiency, empirically resulting in performance degradation.
>
> In contrast to these approaches, our NAMMs are never forced to drop information as they are neural network modules exclusively optimized for performance. Thus, NAMMs could still potentially learn to retain the full KV-cache and not prune any token in case this would yield the best performance. However, one of the key hypotheses of our work is that there is a large amount of redundant and misplaced information that is actually hurting the base model. This hypothesis is extensively validated within our results, as NAMMs are actually able to consistently improve the full-cache performance by learning to forget this unhelpful information allowing the base model to ‘focus’.
>
> We provide some qualitative examples in Figure 7 of Section 4.3, showing that the unhelpful information our NAMMs learn to prune in natural language tasks includes grammatical redundancies and repeated words, while in coding tasks includes comments and some import statements. Moreover, the discarded information seems to vary in different layers of the base model. For instance, only particular layers seem to need information about each task’s preamble.
>
> Following the reviewer’s feedback, **we added new results and discussion in Section 4.1 to expand our analysis of the issues with prior hand-designed ‘lossy’ approaches and better highlight the difference to our method. In particular, we now added results on the full set of 21 LongBench tasks varying the maximum cache size of H2O and L2 across [2048, 4096, 8192, 16384, 32768], and compared them with our NAMM.**
>
> Please, refer to Figure 5 of the latest rebuttal revision or [**this link**](https://anonymous.4open.science/r/evomemory-rebuttal-1ED4/perf_per_cs.png) for their visualization.
>
> In line with the results in the prior literature of KV cache pruning, the overall performance of H2O and L2  is always inferior to the original model with full cache. Furthermore, intuitively, with smaller cache sizes and more lossy reconstructions the performance degradation monotonically worsens. In contrast, our learned NAMMs are the only models which are remarkably able to go beyond the original model’s performance.
>
> We hope these new results and discussion now clearly highlight the difference between NAMMs and prior hand-designed approaches, further validating the potential for learning an unconstrained neural network to drop the unhelpful information discussed above.

---

> ### Author Response · Authors · 2024-11-23
> **Response to Reviewer 2 (tak4) - Part 2**
>
> > 2 [...] while the authors compare against some of them, like H2O and L2, they do not discuss or compare against some of the other relevant work [...]  and either compare their work against them or discuss why they are not comparable.
>
> We thank the reviewer their suggestion and for pointing us to these recent works:
>
> While definitely relevant, the second referenced work [4] proposes CacheGen which is a framework for encoding the KV cache with a carefully-designed compression scheme that allows for low context-loading delays and is shown to only suffer small performance degradation. Thus, we believe CacheGen to be an orthogonal approach to our method, as its purpose is to compress the tokens in the KV cache rather than pruning. In fact, in its paper, CacheGen is also evaluated on top of other KV cache pruning techniques, such as H2O itself.
>
> Thus, following the reviewer’s suggestion, **we now added the CacheGen paper to our related work**, discussing its orthogonality to our approach.
>
> In contrast, the first referenced work [3] proposes FastGen, another interesting hand-designed method for KV cache pruning, which proposes a multi-step strategy combining existing heuristics from the literature about which tokens to preserve. While we already discussed this paper in our Related work section, we did not consider a comparison since the authors did not share their code.
>
> Thus, to still strive to accommodate the reviewer’s request **we re-implemented FastGen**, trying to closely follow the specifications from its paper. We will consider sharing our reimplementation to help the rest of the community also compare against this baseline (in case the FastGen authors do not decide to update their GitHub repository, which currently sits empty almost one year after publication). **We added Section A.5 to the Appendix, where we provide a description of FastGen together with all re-implementation and hyper-parameter details.**
>
> In practice, we found FastGen’s performance to be quite sensitive to hyper-parameters. As we are dealing with tasks with much longer prompts than what was originally considered in their paper, we found we had to retune its hyper-parameters, considerably raising its main ‘threshold’ to avoid losing too much performance over the base model (see Figure 9 of the rebuttal revision or [**this link**](https://anonymous.4open.science/r/evomemory-rebuttal-1ED4/perf_per_th_fastgen.png) for our ablation). **We added results** for the best parameter that retained over 95% normalized performance while still discarding a non-trivial portion of tokens on all LongBench tasks together with our main results **in Section 5**. We summarize these results also in the table below:
>
>
> | Model | Performance (mean) | Normalized performance (mean) | Average Cache size |
> |-------------|--------------------|-------------------------------|--------------------|
> | Base model| 28.86| 1.00| 10107|
> | H2O | 28.37| 0.99| 6662 |
> | L2| 27.42| 1.00| 6662 |
> | FastGen | 27.88| 0.95| 9538 |
> | NAMM (Ours) | 29.33| 1.11| 8409 |
>
>
> In line with the results in their original paper, our re-implementation is also able to obtain efficiency gains without major compromises to performance. However, the trade-off seems to be worse than the H2O and L2 baselines, which we hypothesize generalize better to tasks with much longer contexts due to their simplicity and their much fewer number of hyper-parameters and implementation details than FastGen (which we note, once again, were conceived for LLMs and tasks with much shorter contexts).
>
> In our experiment Section, we also collected and included results for the InfiniteBench and Choubun benchmarks, confirming the same trends, as summarized in the tables below:
>
> | Model | InfiniteBench (mean) | Normalized performance (mean) | Average Cache size |
> |-------------|----------------------|-------------------------------|--------------------|
> | Base model| 1.05 | 1.00| 32747|
> | H2O | 1.05 | 1.00| 8193 |
> | L2| 1.63 | 1.55| 8193 |
> | FastGen | 1.42 | 1.49| 23016|
> | NAMM (Ours) | 11.00| 10.45 | 13192|
>
>
> | Model | ChouBun (mean) | Normalized performance (mean) | Average Cache size |
> |-------------|----------------|-------------------------------|--------------------|
> | Base model| 21.21| 1.00| 12099|
> | H2O | 19.86| 0.94| 8292 |
> | L2| 18.93| 0.89| 8292 |
> | FastGen | 18.93| 0.89| 8616 |
> | NAMM (Ours) | 24.22| 1.15| 9895 |
>
> However, we note that FastGen is only compatible with language modeling tasks. In particular, one of the main ways it differs from H2O [1] is by preserving particular grammar tokens in some of its strategies (e.g., punctuation, special words, etc).
> Thus, as this baseline was specifically designed for LMs rather than arbitrary transformers, we did not consider applying it to the other 0-shot transfer settings (computer vision, RL).

---

> > ### Author Response · Authors · 2024-11-23
> > **Response to Reviewer 2 (tak4) - Part 3**
> >
> > **References**
> >
> > [1] Zhang, Zhenyu, et al. "H2o: Heavy-hitter oracle for efficient generative inference of large language models." Advances in Neural Information Processing Systems 36 (2023): 34661-34710.
> >
> > [2] Devoto, Alessio, et al. "A Simple and Effective $ L_2 $ Norm-Based Strategy for KV Cache Compression." arXiv preprint arXiv:2406.11430 (2024).
> >
> > [3] Ge, Suyu, et al. "Model tells you what to discard: Adaptive kv cache compression for llms." arXiv preprint arXiv:2310.01801 (2023).
> >
> > [4] Liu, Yuhan, et al. "Cachegen: Kv cache compression and streaming for fast large language model serving." Proceedings of the ACM SIGCOMM 2024 Conference. 2024.

---

> > > ### Author Response · Authors · 2024-12-01
> > > **Follow-up to Rebuttal**
> > >
> > > We would like to thank reviewer tak4 once again for their time and efforts in reviewing our work.
> > >
> > > With our response and rebuttal revision, we made our best efforts to try and address the points raised in their constructive feedback. As the discussion period is coming to a close, we would really appreciate it if they could let us know if they found our response satisfactory and their revised opinion.
> > >
> > > Please, do not hesitate to raise any further questions or concerns about our work.

---

> ### Comment · Reviewer_tak4 · 2024-12-02
> **Response to Authors' Rebuttals**
>
> Sorry for my late response. Thank you for your comprehensive rebuttals which clarify most of my concerns and adding further experiments/discussion about other related works, especially adding the  implementation of FastGen.  In light of your responses and improvements, I am increasing my score (6-->8).

---

### Official Review · Reviewer_PVCR · 2024-11-09

**Soundness:** 3
**Presentation:** 2
**Contribution:** 2
**Rating:** 8
**Confidence:** 4

**Summary:**

They propose a learned strategy for maintaining the size of your KV cache. An auxiliary model utilizes the attention map to produce a score of how important each key-value pair is for predicting future tokens. This model is small (1-layer transformer). They train this auxiliary model using the attention maps of Llama 3 8B for long-context language tasks, but show that this auxiliary model is compatible with other models (Llama 3 70B) and can transfer to other modalities (video). The KV cache management reduces the memory of the cache, but performance does not decrease by much, sometimes improving upon using the full cache.

**Strengths:**

The algorithmic design seems sound and seems to perform well based on their results. It’s interesting that the auxiliary model can then transfer to other models and modalities. I liked some of their ablation studies, such as showing that NAMM drops more tokens from later layers than earlier layers. This seems to imply that there is much more to learn from attention scores than what manually-designed algorithms do.

**Weaknesses:**

There are a couple problems with the paper presentation (missing citations) + rigor of their experiments. I highly recommend that these problems are addressed for the score to be raised.


There are some problems with the flow of the paper due to many of the details being relegated to the Appendix.
- The related works paragraph in the main body is a bit too short, and poses NAMM to be the first method to propose a learned black-box strategy. However, previous works already do this. For example, Pyramid-BERT [1] does exactly that, and DMC [2] proposes a continual pre-training objective.
- Please include limitations of the method in the main body or refer to the Appendix section where you do somewhere.
- Limitations should also address the computational complexity of this method. How slow is NAMM compared to other methods?
- There should be an algorithm block in the main body detailing exactly how NAMM is used during inference. The written description is too vague. Especially include details such as the 512-chunking that is detailed in the Appendix.
- There are no details of exactly how the auxiliary model is trained in the main body or the appendix, except for a sentence mentioning that the auxiliary model optimizes the performance of 3 benchmarks. Please detail the exact objective somewhere.


Weaknesses in experiments:
- They should report the computational complexity of NAMM, how much longer is it to do inference using NAMM versus other baselines? In other words, how expensive is it to generate scores from the auxiliary model for each attention layer?
- A result that they particularly emphasize in this paper is that models do better on long-context benchmarks with NAMM cache management over utilizing the full KV cache. That makes sense, the auxiliary model seems to pick out the most important tokens as the aux model is _trained on long-context benchmarks itself_. However, I worry that the significance of these results is not rigorously tested and may simply be because they do not use any fine-tuning strategy to adapt the  base model to long-context tasks, as it is standard to do in practice.
  - All the reported results apply NAMM on top of a Llama-3-8B model that they adapt to use long-context using “NTK-aware positional interpolation,” a zero-shot method that can be found in a Reddit post. I am not aware of the validity or the limitations of this approach. They should also apply NAMM on top of standard long-context models already out there, such as Mistral 7b with 32k context. Does NAMM still do significantly better?
   - NAMM is only compared against L2 and H2O, but it should also compare against other finetuning strategies e.g. DMC, and strategies that also utilize the attention scores explicitly such as FastGen [3].
- The degree to which each baseline prunes the KV cache varies significantly, so it is hard to tell whether NAMM has better performance on benchmarks simply because it does not prune the cache to the degree the other baselines do. It would be good to actually control the hyperparameters of each method, such as the threshold at which a key-value pair is pruned, such that performance between methods is compared for some fixed cache size. Ideally, there should be a plot depicting this with x axis as cache size and y axis as performance.


1. Pyramid-BERT: Reducing Complexity via Successive Core-set based Token Selection https://arxiv.org/pdf/2203.14380
2. Dynamic Memory Compression: Retrofitting LLMs for Accelerated Inference
3. Suyu Ge,  et al.   Model tells you what to discard:  Adaptive KV cache compression for LLMs.

**Questions:**

1. Paper seems to imply they transform the attention map after applying the causal mask. Is this true? If so, what is STFT doing exactly? Is it doing anything meaningful?
2. How important is it to use the  fourier transformation to the signals instead of the original attention map?
3. I found the intuition for the gradient analysis section fairly confusing, and I am not sure what the takeaway is or why the quantities they measure matter. I wish this was better motivated, or it could be moved to the Appendix.

---

> ### Author Response · Authors · 2024-11-23
> **Response to Reviewer 1 (PVCR) - Part 1**
>
> We thank the reviewer for the constructive feedback. The following are our responses to the comments and questions, please see the details in the revised PDF.
>
> **Weaknesses**
>
> > 1 The related works paragraph in the main body is a bit too short, and poses NAMM to be the first method to propose a learned black-box strategy. [...] previous works already do this (Pyramid-BERT [1] and DMC [2])
>
> Regarding our Related works section, for space constraints, we opted to only describe what we believed to be the most directly connected papers in the main text. However, at the end of the section, we also referred to Appendix E, where we tried to give a quite extensive overview (almost a full page), better placing our work in the wider literature, including papers about efficient architectures, memory, and evolution, following the reviewer’s feedback, we moved some of this material back to the main text.
>
> We also thank the reviewer for making us aware of the Pyramid-BERT [1] and DMC [2] works. Pyramid-BERT definitely proposes an interesting and relevant hand-designed token selection strategy based on a parallelized greedy approach of solving the core-set problem. Furthermore, the very recent DMC work appears even more relevant for our paper, as they finetune the original base model to also select and compress specific tokens in the KV-cache in order to minimally deviate from the base model’s original behavior.
>
> Following the reviewer’s feedback, **we modified and extended the related works section** with some additional content from the Appendix and also to try and give proper attributions to these papers, as summarized above. We also rephrased the sentence about the novelty of NAMMs which, to the best of our knowledge, are the first learned ‘memory modules’ dealing with KV cache management that are optimized for improving performance (rather than to retain the base model’s behavior) and show potential of improving both the effectiveness and efficiency of transformers. Please, we hope you will not hesitate to let us know in case this characterization still seems inappropriate in any way.
>
> > 2 Please include limitations of the method in the main body or refer to the Appendix section where you do somewhere.
>
> In Appendix F, we provide an extensive limitation section, providing additional experiments, analyzing some specific failure cases of NAMM potential with practical solutions, and highlighting directions and suggestions for future work.
>
> Following the reviewer’s feedback, we realized this Section was not properly referenced within the main text, **and we have now tried to explicitly mention and highlight it in Section 6.**

---

> > ### Author Response · Authors · 2024-11-23
> > **Response to Reviewer 1 (PVCR) - Part 2**
> >
> > > 3 Limitations should also address the computational complexity of this method. How slow is NAMM compared to other methods?
> >
> > Following the reviewer’s feedback, **we added Section C.3 to the Appendix where we analyzed the training and inference times of NAMMs and the baselines** on our hardware. In particular, we recorded the average running times in two settings: 1) Using samples from the full LongBench benchmark with an average length of 12099, and 2) Using only samples from LongBench selected to exceed the base transformer maximum length with an average length of 32641.
> >
> > Finally, we also record the running time of an ablated version of our NAMM run on top of the base transformer that does not modify its KV cache (named “NAMM (full cache)” in the following table). This artificially-inefficient ablation was done to disentangle the gains from the reduced memory and better analyze the pure overheads from our model’s execution:
> >
> > | Method                      | Time per task sample |
> > |-----------------------------|----------------------|
> > | Longbench average           |                      |
> > | Base model (full cache)     | 4.0                  |
> > | NAMM (full cache)           | 4.54                 |
> > | H2O                         | 3.78                 |
> > | L2                          | 3.74                 |
> > | NAMM (Ours)                 | 3.97                 |
> > | Selected max length samples |                      |
> > | Base model (full cache)     | 15.52                |
> > | NAMM (full cache)           | 17.6                 |
> > | H2O                         | 11.79                |
> > | L2                          | 10.5                 |
> > | NAMM (Ours)                 | 13.61                |
> >
> >
> > As shown in the summarized table above, due to NAMMs being much smaller than the base model and that they are applied at ‘constant intervals’ (every 512 tokens seen by the base transformer in our case) their overheads appear to be marginal. Moreover, as many operations with modern deep learning are memory-bound, the resulting reduction in KV cache size from using NAMMs or any of our baselines seems to offset these overheads and actually provide significant gains in terms of inference time.
> >
> > Overall, the efficiency gains appear very close to the H2O and L2 baselines. However, we note that for both H2O and L2 these efficiency gains come at a loss of performance. In contrast, NAMMs concretely improve the base model’s behavior, with its efficiency gains being free by-products of its fully-learned memory management strategy.
> >
> > > 4 There should be an algorithm block in the main body detailing exactly how NAMM is used during inference.[...] Especially include details such as the 512-chunking that is detailed in the Appendix.
> >
> > Following the reviewer’s feedback, **we added a new paragraph at the end of Section 3.1** providing additional implementation details from Appendix A about how the model is used during inference, including the input chunking and NAMM execution frequency.
> >
> > As suggested, **we accompany this paragraph with an algorithm block** summarizing the full execution pipeline of NAMMs. We also added a second algorithm block to Appendix A, providing an extended and more verbose description.
> >
> > > 5 There are no details of exactly how the auxiliary model is trained in the main body or the appendix, except for a sentence mentioning that the auxiliary model optimizes the performance of 3 benchmarks
> >
> > As described in Section 3, we optimize NAMMs with the Covariance Matrix Adaptation Evolution Strategy (CMA-ES) [3]. Being an evolutionary algorithm, CMA-ES does not require any gradient information and can directly optimize black-box undifferentiable metrics. This property allows us to both optimize for the non-differentiable token selection task of our NAMMs and also train its parameters to maximize non-differentiable task performance metrics directly. In the case of our training procedure on 3 LongBench tasks, these metrics correspond to exact match accuracy (PassageRetrieval-en), ROUGE-L score (DuReader), and F1 score (NarrativeQA).
> >
> > Following the reviewer’s feedback, **we modified Section 3 to make this clearer and we also added Section A.4 to the Appendix, where we provide these exact details together with an additional step-by-step high-level summary of the CMA-ES** algorithm and optimization loop applied to our setting.

---

> ### Author Response · Authors · 2024-11-23
> **Response to Reviewer 1 (PVCR) - Part 3**
>
> **Experiments**
>
> > 1 Report the computational complexity of NAMM, how much longer is it to do inference using NAMM versus other baselines?
>
> We address this feedback in our response to Weakness 3.
>
> > 2 They should also apply NAMM on top of standard long-context models already out there, such as Mistral 7b with 32k context. Does NAMM still do significantly better?
>
> Our choice of using a 0-shot context-extended model was motivated by the potential of learning between the pre-training and finetuning stages, which could enable drastically cutting down the currently prohibitive long-context finetuning costs due to the quadratic dependencies of transformers at training time. Moreover, while NTK-aware interpolation was first introduced in a Reddit post, we believe it is still considered one of the most established and effective methods for 0-shot context adaption in the literature, e.g. see [4].
>
> Following the reviewer’s suggestion, **we now also analyzed applying NAMMs on top of the Mistral 7b base model**. We considered 2 settings:
>
> 1 -  0-shot application, taking our best NAMM model trained with the Llama 8b context-extended model.
> 2 -  With a small amount of additional finetuning (20 generations using CMA-ES and the same 3 training tasks used for Llama).
>
> **We provide our full results and analysis in Table 22 and the new Section C.9 of the Appendix**. We provide a summarized version below:
>
> | Model           | LongBench Mistral base (mean) | Normalized performance (mean) | Average Cache size |
> |-----------------|-------------------------------|-------------------------------|--------------------|
> | Base model      | 30.33                         | 1.00                          | 10107             |
> | H2O             | 28.27                         | 0.96                          | 6662              |
> | L2              | 26.27                             | 0.97                             | 6662              |
> | NAMM (0-shot)   | 30.12                         | 1.04                          | 8518               |
> | NAMM (Finetune) | 30.52                         | 1.08                          | 8654               |
>
> Analogously to our other 0-shot transfer results, we find that NAMMs provide both visible performance and efficiency gains when transferred to the Mistral model, overcoming the efficiency-performance tradeoff of hand-designed baselines. Furthermore, this analysis also shows that performance could be further improved by a few finetuning generations with different base models. While we did not investigate finetuning with our other models, we believe these results highlight the potential of cheaply improving NAMMs’ 0-shot benefits, which we hope will be further explored in future work.

---

> ### Author Response · Authors · 2024-11-23
> **Response to Reviewer 1 (PVCR) - Part 4**
>
> > 3 NAMM should also compare against other finetuning strategies e.g. DMC, and strategies that also utilize the attention scores explicitly such as FastGen [3].
>
> Unlike our work, both DMC [2] and FastGen [3] do not share their code. Moreover, instead of training a small module, DMC performs full gradient fine-tuning of the base model. Given our current computation availability, this would be prohibitively expensive, especially for our considered long-context models (as training transformers has a quadratic dependence on context size, which in our case is 16 times larger than the Llama 2 model considered in the DMC work).
>
> Thus, to still strive to accommodate the reviewer’s request **we re-implemented FastGen**, trying to closely follow the specifications from its paper. We will consider sharing our reimplementation to help the rest of the community also compare against this baseline (in case the FastGen authors do not decide to update their GitHub repository, which currently sits empty almost one year after publication). **We added Section A.5 to the Appendix, where we provide a description of FastGen together with all re-implementation and hyper-parameter details.**
>
> In practice, we found FastGen’s performance to be quite sensitive to hyper-parameters. As we are dealing with tasks with much longer prompts than what was originally considered in their paper, we found we had to retune its hyper-parameters, considerably raising its main ‘threshold’ to avoid losing too much performance over the base model (see Figure 9 of the rebuttal revision or [**this link**](https://anonymous.4open.science/r/evomemory-rebuttal-1ED4/perf_per_th_fastgen.png) for our ablation). **We added results** for the best parameter that retained over 95% normalized performance while still discarding a non-trivial portion of tokens on all LongBench tasks together with our main results **in Section 5**. We summarize these results also in the table below:
>
> | Model       | LongBench (mean) | Normalized performance (mean) | Average Cache size |
> |-------------|--------------------|-------------------------------|--------------------|
> | Base model  | 28.86              | 1.00                          | 10107              |
> | H2O         | 28.37              | 0.99                          | 6662               |
> | L2          | 27.42              | 1.00                          | 6662               |
> | FastGen     | 27.88              | 0.95                          | 9538               |
> | NAMM (Ours) | 29.33              | 1.11                          | 8409               |
>
>
> In line with the results in their original paper, our re-implementation is also able to obtain efficiency gains without major compromises to performance. However, the trade-off seems to be worse than the H2O and L2 baselines, which we hypothesize generalize better to tasks with much longer contexts due to their simplicity and their much fewer number of hyper-parameters and implementation details than FastGen (which we note, once again, were conceived for LMs and tasks with much shorter contexts).
>
> In our experiment Section, we also collected and included results for the InfiniteBench and Choubun benchmarks, confirming the same trends, as summarized in the tables below:
>
> | Model       | InfiniteBench (mean) | Normalized performance (mean) | Average Cache size |
> |-------------|----------------------|-------------------------------|--------------------|
> | Base model  | 1.05                 | 1.00                          | 32747              |
> | H2O         | 1.05                 | 1.00                          | 8193               |
> | L2          | 1.63                 | 1.55                          | 8193               |
> | FastGen     | 1.42                 | 1.49                          | 23016              |
> | NAMM (Ours) | 11.00                | 10.45                         | 13192              |
>
>
> | Model       | ChouBun (mean) | Normalized performance (mean) | Average Cache size |
> |-------------|----------------|-------------------------------|--------------------|
> | Base model  | 21.21          | 1.00                          | 12099              |
> | H2O         | 19.86          | 0.94                          | 8292               |
> | L2          | 18.93          | 0.89                          | 8292               |
> | FastGen     | 18.93          | 0.89                          | 8616               |
> | NAMM (Ours) | 24.22          | 1.15                          | 9895               |
>
>
> However, we note that FastGen is only compatible with language modeling tasks. In particular, one of the main ways it differs from H2O [5] is by preserving particular grammar-based tokens in some of its strategies (e.g., punctuation, special words, etc).
> Thus, as this baseline was specifically designed for LMs rather than arbitrary transformers, we did not consider applying it to the other 0-shot transfer settings (computer vision, RL).

---

> ### Author Response · Authors · 2024-11-23
> **Response to Reviewer 1 (PVCR) - Part 5**
>
> > 4 It would be good to actually control the hyperparameters of each method, such as the threshold at which a key-value pair is pruned, such that performance between methods is compared for some fixed cache size. Ideally, there should be a plot depicting this with x axis as cache size and y axis as performance.
>
> Unlike prior methods, where the cache size can be seen as a hyper-parameter, NAMMs are optimized end-to-end to improve task performance by managing the memory of the base model, in order to discard unhelpful and redundant information and allow each layer to ‘focus’ on the most helpful subset of tokens. Thus, the tokens inside the KV cache and its corresponding size are not manually determined as they are chosen directly by our neural network module for each layer and prompt to yield the best performance.
>
> In contrast, both H2O and L2 do have a single main ‘maximum cache size’ hyper-parameter that determines their memory efficiency. This was chosen following preliminary tests in order to avoid losing too much performance on LongBench over the base model while still reducing the KV-cache size to a significant extent.
>
> Following the reviewer’s suggestion, **we now added results on the full set of 21 LongBench tasks varying the maximum cache size of H2O and L2 across [2048, 4096, 8192, 16384, 32768], and compared them with our NAMM**. We added these results in Figure 5 in Section 4.1 of the main text.
>
> Please, refer to the latest rebuttal revision or [**this link**](https://anonymous.4open.science/r/evomemory-rebuttal-1ED4/perf_per_cs.png) for their visualization.
>
> In line with the results in the prior literature of KV cache pruning, we find that the overall performance of H2O and L2  is always inferior to the original model with full cache and monotonically increases with larger cache sizes. Moreover, a big drop in performance can be observed when lowering the maximum cache size to 4096, which we also observed in our prior experiments and motivated our baseline cache size selection of 8192.
>
> We believe this new Figure now clearly displays the dichotomy between hand-designed strategies aimed at performance retention as opposed to our learned NAMMs, which are remarkably able to go beyond the original model’s performance - while still providing substantial efficiency gains.
>
> **Questions**
>
> > 1 Paper seems to imply they transform the attention map after applying the causal mask. Is this true? If so, what is STFT doing exactly?
>
> We do not think what the reviewer is describing to be the case. The input of our NAMMs for each token in the KV cache is constructed from the raw sequence of attention values, i.e., the vector of all attention scores given from the history of past queries. This unmodified vector is used as input to the STFT, which serves to produce a frequency-based representation of the original signal, with no other transformation happening in between.
>
> Following the reviewer’s question, **we reworded the first paragraph of Section 3.2** to make this explicitly clear.

---

> ### Author Response · Authors · 2024-11-23
> **Response to Reviewer 1 (PVCR) - Part 6**
>
> > 2 How important is it to use the fourier transformation to the signals instead of the original attention map?
>
> As described in the above response to Question 1, NAMMs are conditioned exclusively on the history of attention values for each token. Moreover, the length of this history inherently grows during generation and varies for each different token (since each token was first stored at different times). Thus, the STFT processing step allows us to meaningfully represent the original (potentially unbounded) list of attention values in frequency space and convert it to a fixed-sized feature vector (required for neural network processing), and at the same time, preserving relevant information about the token’s full history. We note that this method of extracting information from unbounded 1D modalities has been very well established across fields such as audio [6], medicine [7], seismology [8], and many more.
>
> Following the reviewer’s question, **we modified Section 3 of the main text to emphasize more clearly the importance of the STFT** to produce meaningful fixed-sized inputs for our NAMMs.
> Moreover, **we also added Section C.10 to the Appendix, where we ablate the STFT procedure and re-train our NAMMs with two different alternatives**, in particular:
> 1) First, we considered the ‘naive’ approach of using the raw attention values directly (cropped to a fixed length) as input to NAMMs. We denote this as “NAMM (Raw attention features).
> 2) Second, we considered substituting the STFT features by constructing a ‘handcrafted’ feature representation that simply includes three values: i) the sum of the attention values of each token, ii) the recency of each token, and iii) the diversity of each token (computed by concatenating the keys and values to represent each token and averaging the L2 distance to all other tokens). We denote this as “NAMM (Recency-Attention-Diversity)”.
>
> Due to time and resource constraints, we trained these baselines only for two incremental phases on the PassageRetrieval-en and Dureader tasks (thus, we also compared them with our original NAMM model after phase 2). Please refer to Table 23 or the summarized Table provided below for our results:
>
> | Model                              | LongBench base (mean) | Normalized performance (mean) | Normalized test performance (mean) | Average Cache size |
> |------------------------------------|-----------------------|-------------------------------|------------------------------------|--------------------|
> | Base model                         | 28.86                 | 1.00                          | 1.00                               | 12099              |
> | NAMM (stage 2)                     | 29.25                 | 1.07                          | 1.04                               | 8521               |
> | NAMM (stage 3)                     | 29.33                 | 1.11                          | 1.07                               | 8409               |
> | NAMM (Recency-Attention-Diversity) | 28.71                 | 1.02                          | 1.00                               | 8000               |
> | NAMM (Raw attention features)      | 29.10                 | 1.03                          | 0.95                               | 8523               |
>
> We find our baselines yield quite different behaviors, both underperforming our original NAMM design:
>
> 1) Our ‘naive’ baseline, taking as input the cropped attention value, is not able to improve over the full cache model when evaluated on the unseen test tasks of LongBench. However, we note that its performance on the training task is significantly beyond the base model. Thus, we find this is strongly suggestive of the occurrence of overfitting, which we believe is to be expected as our memory model now only conditions on very high-frequency information that only considers the latest attention values.
> 2) We find that our ‘handcrafted’ Recency-Attention-Diversity baseline is instead able to improve over the original model, but its improvements are only marginal. We find these results consistent with section D.2 of the extended analysis, which suggests that the behavior of NAMMs is considerably influenced by a combination of different frequencies in the attention spectrogram which are lost by this approach.

---

> > ### Author Response · Authors · 2024-11-23
> > **Response to Reviewer 1 (PVCR) - Part 7**
> >
> > > 3 I found the intuition for the gradient analysis section fairly confusing, and I am not sure what the takeaway is or why the quantities they measure matter. I wish this was better motivated, or it could be moved to the Appendix.
> >
> > At the end of Section 4.3, we analyze how the presence of each token in memory affects the scores of the other tokens. In particular, we do this by studying and visualizing the gradients of the scores with respect to the features of the tokens in memory. The main takeaway of this analysis is that our model learns a mechanism for ‘cross-token competition’, with our model providing higher scores for the tokens with unique ‘attention-spectrogram’ features which are the result of ‘affecting’ the attention matrix in unique ways.
> >
> > Following the reviewer’s feedback and suggestion, **we moved this part of the analysis together with the rest of the extended analysis in Appendix D**, which provided the space for addressing the reviewers' other suggestions. Furthermore, as there is no space limitation for the Appendix, **we also took the occasion to expand this part to provide a more explicit and clearer motivation with additional mathematical details** about the quantities considered.
> >
> >
> > **References**
> >
> > [1] Huang, Xin, et al. "Pyramid-BERT: Reducing complexity via successive core-set based token selection." arXiv preprint arXiv:2203.14380 (2022).
> >
> > [2] Nawrot, Piotr, et al. "Dynamic memory compression: Retrofitting llms for accelerated inference." arXiv preprint arXiv:2403.09636 (2024).
> >
> > [3] Hansen, Nikolaus. "The CMA evolution strategy: A tutorial." arXiv preprint arXiv:1604.00772 (2016).
> >
> > [4] Peng, Bowen, et al. "Yarn: Efficient context window extension of large language models." arXiv preprint arXiv:2309.00071 (2023).
> >
> > [5] Zhang, Zhenyu, et al. "H2o: Heavy-hitter oracle for efficient generative inference of large language models." Advances in Neural Information Processing Systems 36 (2023): 34661-34710.
> >
> > [6] Allen, J. "Applications of the short time Fourier transform to speech processing and spectral analysis." ICASSP'82. IEEE International Conference on Acoustics, Speech, and Signal Processing. Vol. 7. IEEE, 1982.
> >
> > [7] Kıymık, M. Kemal, et al. "Comparison of STFT and wavelet transform methods in determining epileptic seizure activity in EEG signals for real-time application." Computers in biology and medicine 35.7 (2005): 603-616.
> >
> > [8] Astuti, W., et al. "Adaptive Short Time Fourier Transform (STFT) Analysis of seismic electric signal (SES): A comparison of Hamming and rectangular window." 2012 IEEE symposium on industrial electronics and applications. IEEE, 2012.

---

### Author Response · Authors · 2024-11-23
**General rebuttal response**

We thank the reviewers for the time dedicated to our work, and for providing constructive feedback. We tried to comprehensively address each of the raised points in our individual responses below, where we **emphasized** the main additions and modifications now in the rebuttal revision (where we marked new text in the red color). We provide a summarized list with the main changes below:

- We added Section C.9 where we analyzed applying NAMMs on top of a Mistral 7b model, both 0-shot and after some finetuning from the best NAMM trained on top Llama 3 8b.
- We implemented and added FastGen as an additional main baseline in the main text.
- We added results to Section 5 comparing NAMM against the L2 and H2O methods over a broad sweep of cache sizes.
- We added Section C.10 where we ablate the STFT procedure and re-train and test our NAMMs with two different alternatives.
- We modified and extended our related works section with the suggested additional references.
- We added Section C.3 to the Appendix with the training times, inference times, and memory savings of NAMMs and the baselines on our hardware.
- We added Section A.4 to the Appendix with a step-by-step summary of CMA-ES applied to our setting
- We added Section B.2  where we provide a comprehensive high-level description of each individual benchmark we consider.
- We extended Section 4.2 to provide more details about our experimental setting with Decision Transformer.

We hope to have addressed all the raised concerns and would be happy to respond to further questions and suggestions.

---

### Meta-Review · Area_Chair_2eDg · 2024-12-20

**Metareview:**

Post rebuttal, all of reviewers vote for acceptance. The AC checked all the materials and concurs that the paper has done a valuable exploration of using evolution algorithms to compress the KV cache based Transformer memory for efficient decoding. The paper received initial mixed reviews, but the concerns have been thoroughly addressed during the rebuttal period, with authors paying great efforts providing new results and re-writing the draft. With these efforts, the paper can be accepted. Please incorporate necessary changes in the final version.

**Additional Comments On Reviewer Discussion:**

Reviewers had initial concerns including but not limited to the computation complexity of the method (NAMM), generalization to other models, and connection to related works etc. Such concerns have already been effectively addressed to the point where all reviewers vote for acceptance post-rebuttal. Plus the paper also shows interesting behaviors of the model (e.g., pruning unhelpful information shows improvement over the base model). These are all positive signs pointing to acceptance.

---

### Decision · Program_Chairs · 2025-01-22

Accept (Poster)